# POLY-ATTENTION: A GENERAL SCHEME FOR HIGHER-ORDER SELF-ATTENTION

**Sayak Chakrabarti**
Computer Science,
Columbia University,
New York, NY 10027.
sayaksc@gmail.com

**Toniann Pitassi**
Computer Science,
Columbia University,
New York, NY 10027.
toni@cs.columbia.edu

**Josh Alman**
Computer Science,
Columbia University,
New York, NY 10027.
josh@cs.columbia.edu

## ABSTRACT

The self-attention mechanism, at the heart of the Transformer model, is able to effectively model pairwise interactions between tokens. However, numerous recent works have shown that it is unable to perform basic tasks involving detecting triples of correlated tokens, or compositional tasks where multiple input tokens need to be referenced to generate a result. Some higher-dimensional alternatives to self-attention have been proposed to address this, including higher-order attention (Sanford et al., 2023) and Strassen attention (Kozachinskiy et al., 2025), which can perform some of these polyadic tasks in exchange for slower, superquadratic running times.

In this work, we define a vast class of generalizations of self-attention, which we call poly-attention mechanisms. Our mechanisms can incorporate arbitrary higher-order (tensor) computations as well as arbitrary relationship structures between the input tokens, and they include the aforementioned alternatives as special cases. We then systematically study their computational complexity and representational strength, including giving new algorithms and matching complexity-theoretic lower bounds on the time complexity of computing the attention matrix exactly as well as approximately, and tightly determining which polyadic tasks they can each perform. Our results give interesting trade-offs between different desiderata for these mechanisms, including a tight relationship between how expressive a mechanism is, and how large the coefficients in the model may be so that the mechanism can be approximated in almost-linear time.

Notably, we give a new attention mechanism which can be computed exactly in quadratic time, and which can perform function composition for any fixed number of functions. Prior mechanisms, even for just composing two functions, could only be computed in superquadratic time, and our new lower bounds show that faster algorithms for them are not possible.

## 1 INTRODUCTION

The transformer architecture, introduced by Vaswani et al. (2017), has the *self-attention* mechanism at its heart, which is used to capture pair-wise correlations in large language models. Since its inception, it has been used in a variety of large language model (LLM) architectures, including BERT (Devlin et al., 2019), GPT series (Radford et al., 2018; Brown et al., 2020; OpenAI, 2023), Claude (Anthropic, 2024), Llama (Grattafiori et al., 2024), and o1 (OpenAI, 2024). Its success has led to its prominent use in nearly every area of modern deep learning.

Transformers consist of three main components within each block: an input Multilayer Perceptron (MLP) layer, followed by a self-attention mechanism, then finally an output MLP layer Vaswani et al. (2017). The self-attention mechanism is a function from $\mathbb{R}^{n \times d} \to \mathbb{R}^{n \times d}$ which computes and combines weighted pairwise correlations between tokens in its input, and is key to the success of the Transformer model.

**Self-attention (Vaswani et al., 2017).** For a matrix $M$ and index $i$, we write $M_i$ to denote the $i$th row of $M$. Given a query matrix $Q \in \mathbb{R}^{n \times d}$, key matrix $K \in \mathbb{R}^{n \times d}$ and value matrix $V \in \mathbb{R}^{n \times d}$ for a specific input, the output of the self-attention function is given by the matrix $Att \in \mathbb{R}^{n \times d}$, whose $i^{th}$ row is:

$$Att_i = \frac{\sum_{j \in [n]} \exp(\frac{1}{d}\langle Q_i, K_j \rangle) V_j}{\sum_{j \in [n]} \exp(\langle Q_i, K_j \rangle)}.$$

Despite the widespread use of self-attention in Transformers, there are limits to its expressive power, which is intuitively limited to capturing pairwise correlations between tokens. In particular, researchers have defined a number of basic tasks such as iterated function composition, Match3, Parity, Majority, and Dyck-1 which require higher order relationships than pairwise correlations and provably cannot be solved by simple self-attention networks (Sanford et al., 2024b; Peng et al., 2024; Hahn, 2020). Empirical studies have also confirmed this intuition, showing poor performance by simple Transformers on benchmark datasets like multiplication, logical puzzles and dynamic programming Dziri et al. (2023), memorized mappings (Zhang et al., 2025) and other datasets like SCAN (Lake & Baroni, 2018), PCFG (Hupkes et al., 2020), CLUTRR (Sinha et al., 2019), CoGS (Kim & Linzen, 2020), GFQ (Keysers et al., 2020), and CREPE (Ma et al., 2023).

In this paper, we focus especially on a type of task called function composition. As a simple example, the language model may be given the query "If Sam lives in Toronto, Peter lives in Paris, Toronto is in Canada, and Paris is in France, which country does Sam live in?", and the model is expected to reply "Canada". This is a composition of two functions: the first maps people to cities, and the second maps cities to countries. Several works including (Peng et al., 2024; Dziri et al., 2023; Lu et al., 2023) have shown, both theoretically and experimentally, that simple language models are unable to perform these tasks. In order to overcome these representational limitations, several stronger attention mechanisms have been proposed, notably *higher-order tensor attention* and *Strassen attention* which we define next.

**Tensor-attention.** Clift et al. (2020) came up with a tensor generalization of self-attention, called 2-simplicial attention, which Sanford et al. (2024b) also studied as the *higher-order tensor attention* (that we will call 3-tensor attention) for a query matrix $Q^{(1)} \in \mathbb{R}^{n \times d}$, key matrices $Q^{(2)}, Q^{(3)} \in \mathbb{R}^{n \times d}$ and value matrices $V^{(2)}, V^{(3)} \in \mathbb{R}^{n \times d}$. The output is given by the matrix $Att^{(T)} \in \mathbb{R}^{n \times d}$, whose $i^{th}$ row is given by:

$$Att_i^{(T)} = \frac{\sum_{\ell_1, \ell_2 \in [n]} \exp(\frac{1}{d}\langle Q_i^{(1)}, Q_{\ell_2}^{(2)}, Q_{\ell_3}^{(3)} \rangle) V_{\ell_2}^{(2)} \odot V_{\ell_3}^{(3)}}{\sum_{\ell_1, \ell_2 \in [n]} \exp(\frac{1}{d}\langle Q_i^{(1)}, Q_{\ell_2}^{(2)}, Q_{\ell_3}^{(3)} \rangle)}.$$

Here $\odot$ denotes the element-wise product (also called Hadamard product), and for three vectors $a, b, c \in \mathbb{R}^d$, we define $\langle a, b, c \rangle = \sum_{\ell=1}^d a[\ell]b[\ell]c[\ell]$.

Sanford et al. (2024b) showed that one 3-tensor attention head can solve more complicated tasks like Match3, which requires finding a triple of correlated tokens. They also defined a natural generalization to $t$-tensor attention, which can solve Match-$t$ for $t \geq 3$.

**Strassen-attention.** Later, Kozachinskiy et al. (2025) gave a more efficient attention mechanism that can also perform Match3 and several other tasks difficult for self-attention. (As we will discuss shortly, 3-tensor attention can have prohibitive computational complexity, and Strassen-attention was defined as a step toward addressing this.) This attention mechanism is again defined over a query matrix $Q^{(1)} \in \mathbb{R}^{n \times d}$, key matrices $Q^{(2)}, Q^{(3)} \in \mathbb{R}^{n \times d}$ and value matrices $V^{(2)}, V^{(3)} \in \mathbb{R}^{n \times d}$. The output matrix is $Att^{(S)} \in \mathbb{R}^{n \times d}$, where the $i^{th}$ row, for $i \in [n]$, is given by:

$$Att_i^{(T)} = \frac{\sum_{\ell_2, \ell_3 \in [n]} \exp(\frac{1}{d}(\langle Q_i^{(1)}, Q_{\ell_2}^{(2)} \rangle + \langle Q_{\ell_2}^{(2)}, Q_{\ell_3}^{(3)} \rangle + \langle Q_{\ell_3}^{(3)}, Q_i^{(1)} \rangle)) V_{\ell_2}^{(2)} \odot V_{\ell_3}^{(3)}}{\sum_{\ell_2, \ell_3 \in [n]} \exp(\frac{1}{d}(\langle Q_i^{(1)}, Q_{\ell_2}^{(2)} \rangle + \langle Q_{\ell_2}^{(2)}, Q_{\ell_3}^{(3)} \rangle + \langle Q_{\ell_3}^{(3)}, Q_i^{(1)} \rangle))}.$$

Quite recently, 3-tensor attention has been implemented and performances studied by Roy et al. (2025). We refer the reader to Section B in which we survey other attention mechanisms and the landscape of results known about them in more detail.

## 1.1 RUNNING TIME CONSIDERATIONS

A natural trade-off arises in these proposed attention mechanisms: as the attention mechanism becomes more general to give more representational power, the required running time increases too. This can often be prohibitive: the quadratic running time of self-attention is already a computational bottleneck which is mitigated in practice only by extensive hardware; a *superquadratic* running time may not be practical even with such hardware speedups.

We compare here the running times of various attention mechanisms as a function of $n$, the number of input tokens, where the embedding dimension is $d = O(\log n)$; see running times in Table 1 below.

**Exact Algorithms.** The best algorithms for self-attention take time $n^{2+o(1)}$, matching the straightforward algorithm. For tensor attention, the best algorithm is also the straightforward algorithm, which for $t$-tensor attention ($t \geq 3$) runs in superquadratic time $n^{t+o(1)}$.

The straightforward algorithm for Strassen attention, just following its definition, takes time $n^{3+o(1)}$. However, Kozachinskiy et al. (2025) give a faster algorithm for Strassen attention with running time $O(n^\omega)$, where $\omega \leq 2.3714$ is the exponent of matrix multiplication (Alman et al., 2025), i.e., the constant such that $n \times n$ matrices can be multiplied in time $O(n^\omega)$. This faster algorithm is still truly supercubic, and moreover, we note that the aforementioned bound on $\omega$ comes from a highly theoretical algorithm, and typically either $\omega \approx 2.81$ from Strassen's algorithm (Strassen, 1969), or even $\omega = 3$ from the straightforward matrix multiplication algorithm, are used in practice. (Kozachinskiy et al. (2025) named it after Strassen's matrix multiplication algorithm to emphasize this faster algorithm.)

It is natural to wonder whether even faster algorithms are possible, and particularly whether tensor attention or Strassen attention could be computed in quadratic time. In fact, these known running times are known to be optimal under standard complexity-theoretic assumptions, so these algorithms cannot be improved. For self-attention and tensor attention, this was shown in prior work (Alman & Song, 2023; 2024); for Strassen attention, we prove this here in Theorem 3.6 below.

**Approximation Algorithms.** In most cases, a sufficiently accurate *approximation* of self-attention suffices, and this can sometimes be computed much faster. Alman & Song (2023) shows that as long as the entries of the query and key matrices are bounded (and all have magnitude at most $B = o(\sqrt{\log n})$) we can compute an entry-wise approximation of the self-attention matrix in almost linear time, $n^{1+o(1)}$. [1] Alman & Song (2024) similarly showed how to compute an entry-wise approximation of tensor attention $Att^{(T)}$ in $n^{1+o(1)}$ time, with a smaller bound on $B$. These prior works have also shown matching lower bounds, showing that these bounds $B$ are tight: if the weights are even slightly larger, than the straightforward exact running times discussed above are unavoidable. (These lower bounds use standard assumptions from fine-grained complexity theory; see Section 4 for more details.) Many different lines of experimental work studied Transformers with reasonable precision guarantees (Zafrir et al., 2019; Sun et al., 2019; Katharopoulos et al., 2020; Dettmers et al., 2022; Xiao et al., 2023; Dettmers et al., 2022; Perez et al., 2023; Roy et al., 2021; Han et al., 2024).

In this paper, we build on this line of work and give the first fast approximation algorithm for Strassen attention. We show that, if all the weights are bounded by $B = o(\sqrt{\log n})$, then one can approximate Strassen attention in almost linear time $n^{1+o(1)}$, and if the weights are larger, then the exact running time of $n^{\omega-o(1)}$ cannot be avoided (again using fine-grained complexity assumptions). This lower bound fits within a new, much more general lower bound on different generalizations of attention which we will state in Theorem 3.6 later. In partic-

| Mechanism | Exact cc | Apx cc | Bound |
|---|---|---|---|
| Self-attention | $n^{2+o(1)}$ | $n^{1+o(1)}$ | $o(\sqrt{\log n})$ |
| $t$-Tensor | $n^{3+o(1)}$ | $n^{1+o(1)}$ | $o((\log n)^{1/t})$ |
| Strassen | $n^{\omega+o(1)}$ | $n^{1+o(1)}$ | $o(\sqrt{\log n})$ |
| Tree (new) | $n^{2+o(1)}$ | $n^{1+o(1)}$ | $o(\sqrt{\log n})$ |
| Poly (new) | $n^{t+o(1)}$ | $n^{1+o(1)}$ | $o((\log n)^{1/k})$ |

Table 1: This summarizes the running times of both exact and approximate algorithms for these attention variants. For entry-wise approximation (Apx cc), the bound $B$ is the maximum absolute value of the matrix entries such that we can entry-wise approximate the output matrix in near-linear time; the attention polynomial is in $t$ variables and has degree $k$. Alman & Song (2023; 2024) proved bounds for self-attention and tensor-attention, while we prove the rest.

---

[1] An entry-wise approximation outputs a matrix where each entry is at most $\frac{1}{\text{poly}(n)}$ far from the exact value.

ular, although the statement appears
similar to prior work, proving this
requires substantial new techniques,
since prior techniques focused on proving *cubic* lower bounds, but Strassen attention actually has a subcubic (but superquadratic) time algorithm based on matrix multiplication; see Section 4 for more details.

## 1.2 POLY-ATTENTION IS ALL YOU NEED

In this work we introduce a more general class of attention mechanisms called *poly-attention* that generalizes and improves upon these previous attention mechanisms. An instantiation of poly-attention is given by a *base* polynomial, $h$, over $t$ variables, degree $k$ and sparsity $s$. We will precisely define poly-attention shortly, and show that it includes self-attention, tensor attention, and Strassen attention as special cases.

Our main results include complete and exhaustive analyses of the running times one can achieve to compute or approximate different poly-attentions, as well as the expressive power of each one. Using these, we identify new, specific instantiations of poly-attention which are simultaneously more expressive than self-attention, and easier to compute than prior replacements to self-attention. One may also use our results to identify attention mechanisms of interest which achieve a desired trade-off between expressiveness and computational complexity.

**Tree-attention.** We particularly highlight a subclass of our poly-attention mechanisms that we call *tree-attention*, which loosely speaking is characterized by a subclass of degree-2 base polynomials $h$ that possesses a tree-like property. We find that all tree-attention mechanisms can be computed in *quadratic* time, matching the running time of standard self-attention. Furthermore, we show that tree-attention can solve $r$-fold function composition for *any* constant $r$.

This is a substantial improvement on prior attention mechanisms. Self-attention cannot even solve 2-fold function composition. Meanwhile, 3-tensor attention and Strassen attention, which can solve 2-fold function composition, require superquadratic time, and furthermore, they cannot solve 3-fold function composition. Our new tree-attention can solve $r$-fold function composition for all $r$ and can be computed in quadratic time (Theorem 3.4).

We give a more detailed analysis of tree-attention, including tight exact and approximation algorithms, in Section 3.2, (Theorem 3.5). We posit tree-attention as the best of all worlds in terms of representational strength and time complexity. In addition to strictly improving the expressive power of the self-attention mechanism, we will see that the runtime of tree-attention matches the best possible runtimes in both the exact and approximate versions. We envision two types of users/applications:

- if quadratic running time can be tolerated then use the exact algorithm for tree-attention
- if a faster, almost linear running time is needed, then the user should find the largest bound $B$ on the weights which can be tolerated by their hardware and architecture, and then apply the most expressive tree-attention which can be approximated quickly for that $B$ (we will explore the trade-off in Section 3.2).

We emphasize that our exact and approximate algorithms for tree-attention only use straightforward matrix multiplication algorithms, and do not rely on bounds on $\omega$ or other impractical fast matrix multiplication algorithms. See Section 5 for an experimental validation.

**Full characterization of poly-attention.** Beyond tree-attention, we give a full characterization of the running time needed to compute poly-attention as a function of the underlying properties of the base polynomial, $h$. We find that these mechanisms often require cubic or more time to compute exactly, but nonetheless have fast approximation algorithms when $B$ (the bound on the weights) is small enough, and meanwhile can perform *very* complex tasks.

## 2 THE POLY-ATTENTION MECHANISM

In this section, we define the general class of poly-attention mechanisms. They will be described by a special class of multi-linear polynomials, which we will call *attention polynomials*.

**Definition 2.1** (Attention polynomial). *We call a polynomial $h(x_1, \ldots, x_t)$ an* attention polynomial *of degree $k$ if it is multi-linear, it has coefficients only in $\{0, 1\}$, and all its monomials have degree at least 2 and at most $k$.*

Attention polynomials will be a central concept in this article. We will use them to concisely denote combinations of inner products of vectors. Given vectors $Y_1, \ldots, Y_t \in \mathbb{R}^d$, consider a multi-linear monomial of an attention polynomial, $m$, of degree $k$ containing variables $x_{j_1}, \ldots, x_{j_k}$, where $1 \leq j_1 < \ldots < j_k \leq t$. We denote $m(Y_1, \ldots, Y_t) := \langle Y_{j_1}, Y_{j_2}, \ldots, Y_{j_k} \rangle$, which is an inner product of order $k$. Then, given an attention polynomial $h(x_1, \ldots, x_t)$ containing $s$ monomials $m_1, \ldots, m_s$, we define $h(Y_1, \ldots, Y_t) := \sum_{i \in [s]} m_i(Y_1, \ldots, Y_t)$.

Now, we describe our new class of *poly-attention* mechanisms, of order $t$, using an attention polynomial $h(x_1, \ldots, x_t)$ of degree $k$ having $s$ monomials (typically think of $t, k, s$ as small constants).

**Definition 2.2** (Poly-attention). *For an attention polynomial $h(x_1, \ldots, x_t)$ having $s$ monomials of degree at most $k$, we define the* poly-attention function *from $\mathbb{R}^{n \times d}$ to $\mathbb{R}^{n \times d}$, which depends on $h$ and has, as its parameters, query-key weights $W_{Q^{(1)}}, \ldots, W_{Q^{(t)}} \in \mathbb{R}^{d \times d}$ and value weights $W_{V^{(2)}}, \ldots, W_{V^{(t)}} \in \mathbb{R}^{d \times d}$.*

*For an input $X \in \mathbb{R}^{n \times d}$, the query-key matrices are denoted as $Q^{(1)} := XW_{Q^{(1)}}, \ldots, Q^{(t)} := XW_{Q^{(t)}}$ and the value matrices as $V^{(2)} := XW_{V^{(2)}}, \ldots, V^{(t)} := XW_{V^{(t)}}$.*

*The output of the poly-attention function will be given by the matrix*

$$Att^{(h)}(Q^{(1)}, \ldots, Q^{(t)}, V^{(1)}, \ldots, V^{(t)}) \in \mathbb{R}^{n \times d},$$

*where the $\ell_1$-th row is defined as:*

$$Att^{(h)}_{\ell_1} = \frac{\sum_{\ell_2, \ldots, \ell_t \in [n]} \exp\left(\frac{1}{d} h(Q^{(1)}_{\ell_1}, \ldots, Q^{(t)}_{\ell_t})\right) V^{(2)}_{\ell_2} \odot V^{(3)}_{\ell_3} \odot \ldots \odot V^{(t)}_{\ell_t}}{\sum_{\ell_2, \ldots, \ell_t \in [n]} \exp\left(\frac{1}{d} h(Q^{(1)}_{\ell_1}, \ldots, Q^{(t)}_{\ell_t})\right)}. \tag{1}$$

We will often drop the $Q^{(i)}$'s and $V^{(j)}$'s from the notation $Att^{(h)}$ when it doesn't lead to ambiguity.

Here, $Q^{(1)}$ will be the query matrix as used in the usual self-attention mechanisms, and $Q^{(2)}, \ldots, Q^{(t)}$ will be the key matrices, as the index of the row of $Q^{(1)}$ corresponds to the row of the output of poly-attention, and correlations are considered with respect to that. However, since we use all the variables (and hence, the matrices) in a symmetric sense, we denote both the query and the key matrices using $Q^{(j)}$ for ease of notation.

**Lemma 2.3.** *Poly-attention captures all the previous higher-order self-attention techniques. In particular, (i) self-attention is poly-attention with the base polynomial $h(x_1, x_2) = x_1 x_2$; (ii) $t$-tensor attention is poly-attention with $h(x_1, \ldots, x_t) = x_1 \ldots x_t$; and (iii) Strassen-attention is poly-attention with $h(x_1, x_2, x_3) = x_1 x_2 + x_2 x_3 + x_3 x_1$.*

## 3 BEYOND SELF-ATTENTION: THE POWER OF POLY-ATTENTION

In this section, we study the strength and limitations of the poly-attention scheme. We begin in Section 3.1 by studying an illustrative example. Thereafter, we will consider tree-attention and poly-attention in full generality.

### 3.1 AN EXAMPLE: FUNCTION COMPOSITION

To demonstrate the power of poly-attention, we analyze a special case when $h(x_1, x_2, x_3) = x_1 x_2 + x_2 x_3$. We show that this specific poly-attention can efficiently solve important tasks faster than any other previous attention mechanisms.

To demonstrate the strength of this polynomial $h$, we define the function composition problem demonstrated earlier. Mathematically, the 2-fold function composition problem is: given two functions $f_1, f_2 : [n] \to [n]$ and $x \in [n]$,

| Mechanism | 2-fold | 3-fold |
|---|---|---|
| Self-attention | No | No |
| 3-Tensor | Yes | No |
| Strassen | Yes | No |
| Tree (new) | Yes | Yes |
| Poly (new) | Yes | Yes |

Table 2: Compositionality results showing support for function composition. Peng et al. (2024) prove impossibility bounds for self-attention, Kozachinskiy et al. (2025) simulate 2-fold with Strassen-attention, while we prove the rest.

output $f_2(f_1(x))$. To express this problem for an attention mechanism, the input is $X \in \mathbb{R}^{(2n+1) \times d}$, where $X_i$ for $i \in [n]$ contains an encoding of $f_1(i)$, $X_j$ for $j \in [n+1, 2n]$ contains an encoding $f_2(j-n)$ and $X_{2n+1}$ contains an encoding of $x$; and our goal is to output the value of $f_2(f_1(x))$ in the $(2n+1)$-th entry of the output.

Peng et al. (2024) proved that self-attention cannot simulate 2-fold function composition, and even that almost $n$ self-attention heads are needed in order to solve it. Since self-attention needs quadratic time to compute, it would take cubic time to compute $n$ heads. All prior mechanisms that solve this, including 3-tensor attention and Strassen-attention, require superquadratic time. This leads to our punchline: poly-attention for this very simple polynomial $h_2$ can simulate 2-fold function composition in just quadratic time!

**Theorem 3.1.** *Let $h_2(x_1, x_2, x_3) = x_1 x_2 + x_2 x_3$. Poly-attention for $h_2$ can simulate function composition using only one head. Furthermore, $Att^{(h_2)}$ can be computed in $O(n^2)$ time.*

We will tightly characterize what weights are needed for efficient approximation of all poly-attentions; in the case of $Att^{(h_2)}$, we find:

**Theorem 3.2.** *Given the polynomial $h_2(x_1, x_2, x_3) = x_1 x_2 + x_2 x_3$, where the entries of the query-key matrices are in $[-B, B]$:*

1. *If $B = o(\sqrt{\log n})$, we can compute an entry-wise $(1/\text{poly}(n))$-approximation of $Att^{(h_2)}$ in time $n^{1+o(1)}$.*

2. *If $B = \Omega(\sqrt{\log n})$, then every algorithm for computing an entry-wise $(1/\text{poly}(n))$-approximation of $Att^{(h_2)}$ requires time $\Omega(n^2)$, unless SETH is false.*

We consider next 3-fold function composition, in which the input is three functions, $f_1, f_2, f_3 : [n] \to [n]$ and $x \in [n]$, and we want to compute $f_3(f_2(f_1(x)))$. To our knowledge, no prior attention mechanisms could perform 3-fold function composition. In particular, although Strassen-attention and 3-tensor attention were designed to solve problems like 2-fold function composition, we prove that they *cannot* compute 3-fold function composition when the precision is bounded:

**Theorem 3.3.** *Strassen-attention and 3-tensor attention, require at least $H > n^{1-o(1)}$ heads to simulate 3-fold function composition when the precision is bounded.*

However, we prove that poly-attention can indeed simulate 3-fold composition, and even more generally $r$-*fold composition* for any constant $r$, and still be evaluated in quadratic time!

**Theorem 3.4.** *For any integer $r \geq 2$, define the polynomial $h_r(x_1, \ldots, x_r) = x_1 x_2 + x_2 x_3 + \ldots + x_r x_{r+1}$. Then, poly-attention for $h_r$ can simulate $r$-fold function composition, and $Att^{(h_r)}$ can be computed exactly in time $O(r^3 n^2)$ (input dimension here is $O(rn)$, not $n$).*

In fact, we give a general characterization of which polynomials $h$ can be used in $Att^{(h)}$ to perform $r$-fold function composition. For example, we will also prove that poly-attention for $h_{r-1}$ can not simulate $r$-fold function composition.

## 3.2 TREE-ATTENTION: POLYNOMIALS LEADING TO EFFICIENT POLY-ATTENTION

We saw in the previous section that instances of poly-attention which can be computed in quadratic time can have great representational strength. A natural question arises: what is the class of attention polynomials that can be exactly computed in only $n^{2+o(1)}$ time? Could there be even stronger ones? We answer this by giving a *complete characterization*. We first define a few notations to describe them.

For an attention polynomial $h(x_1, \ldots, x_t)$ of degree 2, we say that a simple graph $G$ is the *graphical representation* of $h$, if $G$ contains $t$ vertices $v_1, \ldots, v_t$, where vertex $v_i$ corresponds to the variable $x_i$, and

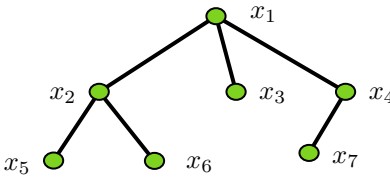

Figure 1: Graphical representation for the tree polynomial $h(x_1, \ldots, x_7) = x_1 x_2 + x_1 x_3 + x_1 x_4 + x_2 x_5 + x_2 x_6 + x_4 x_7$

there exists an edge between $v_i$ and $v_j$ if and only if $x_i x_j$ is a monomial present in $h$. If the graphical representation of $h$ is a tree or a forest, we say that $h$ is a *tree polynomial*, and poly-attention for a tree polynomial will be called *tree-attention*.

Our main result about tree-attention shows that it can be computed just as efficiently as self-attention, both for exact algorithms (where it can be computed in quadratic time) and approximate algorithms (which has the same bound $B = o(\sqrt{\log n})$ as in self-attention, which is also the largest bound for any poly-attention):

**Theorem 3.5.** *Given a tree polynomial $h$, where the entries of the query-key matrices are in $[-B, B]$:*

1. *The output of tree-attention, $Att^{(h)}$, can be exactly computed in $n^{2+o(1)}$ time.*

2. *If $B = o(\sqrt{\log n})$, entry-wise approximation of $Att^{(h)}$ can be computed in $n^{1+o(1)}$ time.*

3. *If $B = \Omega(\sqrt{\log n})$, under standard complexity assumptions, entry-wise approximation of $Att^{(h)}$ requires $\Omega(n^2)$ time.*

Tree polynomials include the polynomials $h_r$ from Theorem 3.4 which can compute function composition. More generally, the poly-attention for a tree polynomial, where the tree has depth $q$, can simulate $(q-1)$-fold function composition, as well as a variety of tree generalizations. (Function composition can be naturally seen as corresponding to the path graph, which is the graphical representation of $h_r$.)

We show next that, for any attention polynomial which is not a tree polynomial (either because it has degree more than 2, or because its graphical representation contains a cycle), its poly-attention requires superquadratic time to compute. Thus, as promised, tree-attentions form a complete characterization of quadratic-time poly-attentions.

## 3.3 COMPUTATIONAL COMPLEXITY OF NON-TREE POLY-ATTENTION

Next, we give a complete characterization of the computational complexity (both exact and approximate) for poly-attention for all attention polynomials $h$.

**Theorem 3.6.** *Given poly-attention for an attention polynomial $h(x_1, \ldots, x_t)$ of degree $k$ and sparsity $s$ which is not a tree polynomial, where the query-key matrices have entries in $[-B, B]$:*

1. *If $B = o((\log n)^{1/k})$, entry-wise $\frac{1}{\text{poly}(n)}$-approximation of $Att^{(h)}$ can be computed in almost-linear time.*

2. *If $B = \Omega((\log n)^{1/k})$, entry-wise $\frac{1}{\text{poly}(n)}$-approximation of $Att^{(h)}$ requires superquadratic time, assuming standard complexity assumptions.*

Prior work gave this characterization for specific polynomials $h$ (Alman & Song (2023) for the usual self-attention (i.e., $h(x_1, x_2) = x_1 x_2$), followed by Alman & Song (2024) for $t$-tensor attention i.e., $h(x_1, \ldots, x_t) = x_1 \cdots x_t$). We discuss in Section 4 below a number of technical hurdles which we overcome to generalize their results to all attention polynomials and prove Theorem 3.6.

Notably, for many polynomials such as $h(x_1, x_2, x_3) = x_1 x_2 + x_2 x_3 + x_1 x_3$ (corresponding to Strassen attention), there is a subcubic algorithm which uses fast matrix multiplication, so prior approaches, which can only prove cubic (or above) lower bounds, cannot apply. In fact, we generalize the Strassen attention algorithm (Kozachinskiy et al., 2025), and prove that for *any* degree-2 attention polynomial $h$ whose graphical representation contains exactly one cycle, there is an exact algorithm for $Att^{(h)}$ running in subcubic time $O(n^\omega)$, and that this cannot be improved.

## 3.4 REPRESENTATIONAL STRENGTH OF POLY-ATTENTION

We have discussed function composition at length, but poly-attention is also able to perform a variety of other basic expressive problems. As an example, Match3 has been highlighted by prior work (Sanford et al., 2024a; Kozachinskiy et al., 2025) as a problem which requires detecting correlated triples of tokens. We define here a generalization called *polynomial root-finding* which can be solved by poly-attention.

The problem is defined in terms of a fixed polynomial $p(x_1, \ldots, x_n)$ (which, unlike an attention polynomial, may have degree 1 monomials, and may not be multi-linear). In the problem, given as input a set $S$ containing $n$ integers, and the goal is to find $y_1, \ldots, y_t \in S$ such that $p(y_1, \ldots, y_t) = 0$.

Match3 is a special case of root-finding, corresponding to the simple polynomial $p(x_1, x_2, x_3) = x_1 + x_2 + x_3$. Circuit evaluation for constant sized circuits, and other related problems can also be captured by polynomial root-finding by using arithmetization. We prove that for *any* polynomial $p$, one can solve polynomial root-finding using poly-attention:

**Theorem 3.7.** *For every polynomial $p(x_1, \ldots, x_t)$, there is an attention polynomial $h(x_1, \ldots, x_t)$ such that a Transformer using two heads of poly-attention for $h$ can solve polynomial root-finding.*

Finding the attention polynomial $h$ for a given polynomial $p$ using our approach is straightforward but requires some details; it could be performed by a user who would like to answer query patterns corresponding to polynomial root-finding for a particular $p$.

### 3.5 IMPLICATIONS OF POLY-ATTENTION

As we have seen, tree-attention can solve many problems which self-attention cannot, and still it can be computed in quadratic time. We show that this quadratic time is indeed practicable by showing in Figure 5 that the time-complexity does not hide large constants.

We further show in experiments in Section H that tree-attention is indeed more expressive than self-attention. This seems to be a promising area of research, and it will be interesting to study the large scale deployment of tree-attention instead of self-attention in follow-up work. One can select an appropriate tree-polynomial to use depending on the relationships between the data that the model intends to process.

When we move to more general poly-attention, for any attention polynomial $h$ which is not a tree polynomial, we have shown in Theorem 3.6 that (without a small bound on the model weights) poly-attention provably requires superquadratic time. Thus, there is a trade-off between expressiveness (most straightforwardly represented by the degree and order of the polynomial $h$, although it could also take into account which tasks like polynomial root-finding can be performed), and running time (depending on how bounded the entries must be). Model designers therefore have a choice, potentially depending on the hardware available to them, the desired running time, and the logical structures they expect to see in their data and queries.

It would be exciting, in future work, to further study the expressive power of tree-attentions other than the ones studied here, and find more examples of complicated tasks with tree-like logical structures that it can solve. As an example, Peng et al. (2024) proposed some more problems like relationship composition, spatial composition and temporal composition which current language models cannot solve; it would be interesting to see how well tree-attention performs on these problems.

## 4 TECHNIQUE OVERVIEW

**Representational strength.** Our representational strength results include both constructions (e.g., showing that tree-attention can perform $r$-fold function composition) and lower bounds (e.g., showing that Strassen-attention and 3-tensor attention cannot perform 3-fold function composition).

Our constructions use a generalization of the "sum of squares" approach of Kozachinskiy et al. (2025): If one can design a simple polynomial $c$ which checks possible outputs of function composition, so that it outputs 0 on correct outputs and large values on incorrect values, then the softmax underlying attention can detect 0s and thus solve the problem. An interesting algebraic challenge arises of expressing $c$ in terms of the monomials available in an attention polynomial $h$.

Our lower bounds make use of communication complexity theory, similar to many other representational lower bounds in the literature. We show that if function-composition can be simulated by these mechanisms, then there is a resulting, very efficient communication protocol for a problem called *myopic pointer jumping*. Results from Chakrabarti (2007); Kozachinskiy et al. (2025) showing that myopic pointer jumping cannot be solved with small communication can then be applied.

**Fast approximation algorithms.** For obtaining entry-wise approximation algorithms for poly-attention, we use low-rank decomposition methods based on the *polynomial method*, which were first applied in the context of Gaussian kernel density estimation (see Aggarwal & Alman (2022); Alman & Guan (2024)). In this approach, one critically approximates the exponential function (part

of softmax) with a low-degree single-variable polynomial. The bound $B$ on the weights then naturally comes into play: the smaller the interval one must approximate the exponential on, the lower degree polynomial one may use.

A similar approach has been used to design approximation algorithms for other variants on attention Alman & Song (2023; 2024; 2025), although a number of intricacies arise in this general setting. For instance (recalling that $t$ is the number of variables in the attention polynomial $h$, and $k$ is the degree), directly applying the approach of Alman & Song (2024) would yield an approximation algorithm whenever $B = o((\log n)^{1/t})$, but our algorithm works even for the much larger bound $o((\log n)^{1/k})$. This is a significant improvement for $t > k$– in tree-attention, one could choose $t = 20$ but $k = 2$.

**Lower bounds.** Our running time lower bounds, where we show that different poly-attention mechanisms cannot be computed in quadratic time (for big enough bounds $B$ on the weights), make use of tools from fine-grained complexity theory. In particular, as in the previous works of Alman & Song (2023; 2024; 2025) on the fine-grained complexity of attention mechanisms, we use a popular conjecture called the Strong Exponential Time Hypothesis (SETH) to obtain conditional hardness results. First introduced in Impagliazzo & Paturi (2001), SETH is a strengthening of the P $\neq$ NP conjecture (so, proving SETH would imply P $\neq$ NP), and is perhaps the most widely used conjecture in fine-grained complexity.

Notably, the way SETH has been used in prior work results in *cubic* (or higher) lower bounds, and makes it difficult to prove lower bounds for running time $\Omega(n^\omega)$ from the matrix multiplication exponent $\omega < 3$. Indeed, for such a lower bound, our starting assumption must itself use matrix multiplication in some way!

In order to prove our lower bound against Strassen attention and other poly-attention mechanisms with $O(n^\omega)$ running times, we therefore use a different conjecture, the Max-2SAT conjecture (see Alman & Vassilevska Williams (2020) and its uses in El Halaby (2016); Jansen & Włodarczyk (2024); Bringmann & Slusallek (2021); Lincoln et al. (2018)), which roughly asserts that our current best algorithm for the Max-2SAT problem cannot be substantially improved. We ultimately show that a faster algorithm for Strassen attention could be used to design a faster algorithm for Max-2SAT, refuting the conjecture. Our proof of this makes use of the *distributed PCP framework* (Abboud et al., 2017) for reducing variants of SAT to other problems through *multi-party interactive communication protocols* (Aaronson & Wigderson, 2009; Rubinstein, 2018).

## 5 EXPERIMENTAL VALIDATION

We have proved that tree-attention can be computed in the same $O(n^2)$ time as self-attention, and can simulate function composition (whereas self-attention cannot). We complement this with a simple experiment to demonstrate empirical learnability and efficiency. We compare the following models: (i) a model with one head and one layer of tree-attention; (ii) a model with one head and two layers of self-attention; and (iii) a model with one head and one layer of self-attention. We train all three in the same way to solve function composition. As expected (proved by Peng et al. (2024)), one head and one layer of self-attention is not able to learn function composition, but we find that the other two are. Furthermore, we find that our tree-attention model learns function composition in many fewer training epochs. Lastly, our empirical evaluation of inference time validates that tree-attention takes roughly similar time as self-attention.[2] See Figure 2 for a summary, and Section H for further details and quantitative results.

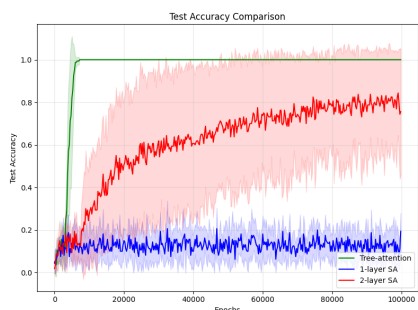

Figure 2: Accuracy per epoch for learning $f_1(f_2(x))$ for sequence length 51, on a single layer of tree-attention, one layer self-attention and two layer self-attention.

We also perform experiments comparing simple networks with self-attention and tree-attention on the COGS NLP dataset Kim & Linzen (2020). This is a dataset which tests whether a model can perform simple compositional tasks when processing language. We find that networks with tree-attention learn to higher accuracy in the same number of epochs. See Section H for more details and quantitative results.

---

[2]As shown in Figure 5, tree-attention takes around 1.3x time as that of self-attention.

## 6 ACKNOWLEDGEMENTS

We especially thank Todd Morrill for valuable discussions that greatly improved the experiments in this paper. We also thank Richard Zemel for helpful suggestions that strengthened both the paper and the experimental evaluation. SC thanks Hantao Yu for pointing out Kozachinskiy et al. (2025) and Roy et al. (2025). Finally, we thank the anonymous reviewers for comments that improved the presentation.

## 7 ETHICS STATEMENT

We affirm that all aspects of this research comply with the ICLR Code of Ethics. This paper does not involve human subjects, personally identifiable data, or sensitive applications, and we do not foresee direct ethical risks.

## 8 REPRODUCIBILITY STATEMENT

The paper contains theoretical results to categorize higher-order self-attention mechanism, and provide a fundamental framework for future work. All these results, including theorems and algorithms, have complete proofs, presented in the appendix. A roadmap to the proofs has been provided in Section A.1 for the reader.

The code which produces the experimental results described in Sections 5 and H can be found in the supplementary materials.

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

## CONTENTS

## A   PRELIMINARIES

### A.1   ROADMAP

In the rest of this paper, we prove all the results that we have stated in the main version. After describing some relevant notations and conjectures that we will use, we prove the results in two parts. First we prove the computational complexities of the poly-attention mechanism, followed by proofs of representational strengths. The proofs of computational complexities use two subdivisions– an upper bound where we show that if the entries of the query-key matrices are bounded, then we can compute an entry-wise approximation in near-linear time, and a lower bound where we show that if the entries of the query-key matrices are large, then assuming certain fine-grained complexity conjectures, computing entry-wise approximations are difficult. As a warm-up, we start with upper and lower bounds for Strassen-attention (Section C), based on which, we proceed to prove the same for general poly-attention in Section E. In order to completely characterize time complexities for poly-attention, we also give quadratic time algorithms for tree-attentions in Section D.

The proofs of the results stated in the main paper are given as follows:

- For Theorem 3.1, poly-attention can simulate function-composition has been proved in Theorem F.6 for $t_0 = 2$, and the time complexity of $O(n^2)$ has been proved in Theorem D.2.

- Theorem 3.2 Part 1 has been proved in Theorem E.2, and Theorem 3.2 Part 2 has been proved in Theorem E.3 Part 1.

- Theorem 3.3 has been proved in Theorem F.3 and Corollary F.4.

- Theorem 3.4 has been proved in Theorem F.6.

- Theorem 3.5 Part 1 has been proved in Theorem D.2, Theorem 3.5 Part 2 has been proved in Theorem E.2, and Theorem 3.5 Part 3 has been proved in Theorem E.3.

- Theorem 3.6 Part 1 has been proved in Theorem E.2 and Theorem 3.6 Part 2 has been proved in Theorem E.3.

- Theorem 3.7 has been proved in Theorem G.1.

### A.2   NOTATION AND BACKGROUND

Throughout this article, for a natural number $n$ we denote $[n]$ as the set $\{1, 2, \ldots, n\}$, $[i : j]$ as the set of integers $\{i, i+1, \ldots, j\}$ for $i < j$, and $[i, j]$ as the set of real numbers between $i$ and $j$. Given a matrix $M \in \mathbb{R}^{n \times m}$, for $i \in [n], j \in [m]$, we denote $[M]_{i,j}$, and more loosely $M_{i,j}$, as the $(i, j)$-th entry of the matrix, $M_i$ as the $i$-th row as a $m$-dimensional vector, and $M_{.,j}$ as the $j$-th column as the transpose of a $n$-dimensional vector. $M_{(i_1:j_1, i_2:j_2)}$ will also denote the submatrix of $M$ having rows $[i_1 : j_1]$ and columns $[i_2 : j_2]$.

For two matrices $A, B \in \mathbb{R}^{n \times m}$, we define $\frac{A}{B}$ as the entry-wise division, i.e., $[\frac{A}{B}]_{i,j} = \frac{A_{i,j}}{B_{i,j}}$. Given a vector $X \in \mathbb{R}^{n \times 1}$, by $diag(X)$, we denote the $n \times n$ diagonal matrix such that $[diag(X)]_{i,i} = X[i]$ for all $i \in [n]$. Some other operators on matrices are defined as follows.

**Definition A.1** (Hadamard product $\odot$)**.** *Given to matrices $A, B \in \mathbb{R}^{n \times m}$, we denote the Hadamard product, denoted by $A \odot B \in \mathbb{R}^{n \times m}$, as the entrywise product*

$$[A \odot B]_{i,j} = A_{i,j} \cdot B_{i,j},$$

*for $i \in [n], j \in m$.*

**Definition A.2** (Row-wise Kronecker product $\oslash$). *For matrices $A \in \mathbb{R}^{n \times d}, B \in \mathbb{R}^{m \times d}$, we denote the row-wise Kronecker product as $A \oslash B \in \mathbb{R}^{nm \times d}$, where*

$$[A \oslash B]_{(i-1)m+j} = A_i \odot B_j,$$

*for $i \in [n], j \in [m]$.*

**Definition A.3** (Entry-wise approximation). *Given a matrix $M \in \mathbb{R}^{n \times d}$, we say that $\widehat{M}$ is an entry-wise $\gamma$-approximation of $M$ if for all $i \in [n], j \in [d]$, we have*

$$|\widehat{M}_{i,j} - M_{i,j}| < \gamma.$$

Throughout this paper, we will choose $\gamma = 1/\text{poly}(n)$.

**Definition A.4** (Entry-wise function). *Given a function $f : \mathbb{R} \to \mathbb{R}$ and a matrix $M \in \mathbb{R}^{n \times m}$, we define the matrix $[M]^f$ as the $n \times n$ matrix such that the the $(i, j)$-th element is*

$$[M]^f_{i,j} = f(M_{i,j}).$$

We will use $[M]^e$ as the entrywise exponentiation function, i.e., $[M]^e_{i,j} = e^{M_{i,j}}$. For a real number $c$, $M/c$ will also refer to the matrix obtained by dividing each entry of $M$ by $c$.

The *coefficient of matrix multiplication*, $\omega$, roughly refers to the exponent of $n$ such that two $n \times n$ matrices can be multiplied in time $O(n^\omega)$ for large enough $n$. There is a series of works trying to improve this coefficient Alman & Williams (2024); Duan et al. (2023); Williams et al. (2024); Fawzi et al. (2022); Alman et al. (2025), with the fastest being Alman et al. (2025) that achieves $\omega = 2.371339$. However, these matrix multiplications require $n$ to be quite large and the hidden constants are enormous, which does not make implementations feasible. There is an algorithm by Strassen Strassen (1969) which is more practicable and achieves $\omega \approx 2.8$, but in most cases, only the naive matrix multiplication algorithm is used as GPUs work better on them.

We will use some more concepts to define the ideas in this article. Given an integer $t$, we define the *symmetric group* of order $r$, $\binom{[t]}{r}$, as the set of tuples:

$$\binom{[t]}{r} = \{(j_1, j_2, \ldots, j_r) \mid 1 \le j_1 < j_2 < \ldots < j_r \le t\}.$$

Note that $\left|\binom{[t]}{r}\right| = \binom{t}{r}$. Based on this, an *elementary symmetric polynomial* of degree $r$ having $t$ variables is defined as

$$\sum_{1 \le j_1 < j_2 < \ldots < j_r \le t} x_{j_1} x_{j_2} \ldots x_{j_r}.$$

**Definition A.5** (Variable separability). *We say that a polynomial $h(x_1, \ldots, x_t)$ is variable separable if there exists a maximum integer $r$ and non-zero attention polynomials $g_1(x_1, \bar{x}^1), \ldots, g_r(x_1, \bar{x}^r)$, where $\bar{x}^1, \ldots, \bar{x}^r$ are disjoint subsets of the variables, such that $h(x_1, x_2, \ldots, x_n) = g_1(x_1, \bar{x}^1) + \ldots + g_r(x_1, \bar{x}^r)$.*

*We denote each of the polynomials $g_i(x_1, \bar{x}^i)$ as* branches.

Note that this definition of variable separability differs slightly from the folklore usage as here we allow $f$ and $g$ to share at most one variable, $x_1$.

In this paper, for a given polynomial $h$, we are interested in computing the entry-wise approximation of $Att^{(h)}$. For this, we define the following version of computing poly-attention approximately.

**Definition A.6** (Entry-wise Approximate Poly-Attention Computation $\mathsf{APAC}^{(h)}(n, d, \Gamma, \gamma)$). *Let $h$ be an attention polynomial in $t$ variables of degree $k$ having sparsity $s$. Given query-key matrices $Q^{(1)}, \ldots, Q^{(t)} \in [-\Gamma, \Gamma]^{n \times d}$ and value matrices $V^{(2)}, \ldots, V^{(t)} \in \mathbb{R}^{n \times d}$, we want to output a matrix $\widehat{Att^{(h)}} \in \mathbb{R}^{n \times d}$ such that for all $i \in [n], j \in [d]$,*

$$|[\widehat{Att^{(h)}}]_{i,j} - [Att^{(h)}(Q^{(1)}, \ldots, Q^{(t)}, V^{(2)}, \ldots, V^{(t)})]_{i,j}| \le \gamma.$$

### A.3 Conjectured hard problems

We define some commonly known problems in fine-grained complexity and state conjectures which will be used to show conditional hardness of generalized attention computations. First, we start by defining a few central problems in fine-grained complexity.

**Definition A.7** ($k$IP problem). *Given sets of vectors $A^1, \ldots, A^k \subseteq \{0,1\}^d$, each of size $n$, and a target inner product $m \in [d]$, the problem of $k$IP asks if there exists $a^1 \in A^1, a^2 \in A^2, \ldots, a^k \in A^k$ such that $\langle a^1, a^2, \ldots, a^k \rangle = m$.*

For $k = 2$ and $m = 0$, it is the famous orthogonal vectors problem, which we will abbreviate 2OV or just the OV problem, and for $k = 2$ and arbitrary $m$, we will abbreviate the problem as IP.

**Definition A.8** ($k$SAT). *In the $k$SAT problem for $k \geq 2$, given as input a $k$-CNF formula $\phi$, determine whether or not $\phi$ has a satisfying assignment.*

**Definition A.9** (Max-kSAT). *In the Max-kSAT$_{n,m}$ problem for $k \geq 2$, given as input a $k$-CNF formula $\phi$ in $n$ variables and $m$ clauses, determine the maximum number of clauses in $\phi$ that can be simultaneously satisfied by a Boolean assignment to the underlying variables.*

Based on these definitions, we are now ready to describe some popular conjectures in fine-grained complexity that we will use to prove our (conditional) hardness results.

**Hypothesis 1** (SETH Impagliazzo & Paturi (2001)). *For every $\delta > 0$, there exists $k \geq 3$ such that $k$SAT can not be solved in time $O(2^{(1-\delta)n})$.*

The current fastest known algorithm for $k$SAT uses the reduction to OV with dimension $d = c \log n$. The best known time complexity of OV is $n^{2-1/O(\log c)}$ given by Abboud et al. (2014a); Chan & Williams (2016).

Since $k$SAT is a special case of Max-kSAT, SETH implies that Max-kSAT also cannot be solved in time $\Omega(2^{(1-\delta)n})$ for every $\delta > 0$. The next hypothesis Alman & Vassilevska Williams (2020) strengthens this further to sparse instances of Max-kSAT.

**Hypothesis 2** (Sparse Max-kSAT Hypothesis). *For every $k \geq 3$ and every $\delta > 0$, there exists $c > 0$ such that Max-kSAT$_{n,cn}$ cannot be solved in time $O(2^{(1-\delta)n})$.*

The fastest known algorithm for sparse instances of Max-kSAT$_{n,cn}$ for $k \geq 3$ takes time $2^{n(1-1/\tilde{O}(c^{1/3}))}$ Alman et al. (2016); therefore the above hypothesis is consistent with the state-of-the-art algorithms. In contrast to the special case of Max-kSAT for $k = 2$, the hypothesis is false. The best algorithm for Max-2SAT Williams (2005; 2007) runs in time $2^{\omega n/3}\text{poly}(n)$, where $\omega$ is the matrix multiplication exponent. The following Max-2SAT hypothesis states that William's algorithm Williams (2005) is essentially optimal when $k = 2$.

**Hypothesis 3** (Max2SAT hypothesis). *For every $\delta > 0$, there exists a $c > 0$ such that Max-2SAT$_{n,cn}$ cannot be solved in time $O(2^{n(\omega/3-\delta)})$, where $\omega$ is the matrix multiplication exponent.*

The following theorem gives a reduction from $k$SAT to $k$IP, thus proving the hardness of $k$IP under SETH.

**Theorem A.10** (Williams (2005); Abboud et al. (2014b); Backurs & Indyk (2015); Abboud et al. (2015)). *Assuming SETH, for every $k$ and $\delta > 0$, there exists $c > 0$ such that $k$IP$_{n,c\log n}$ cannot be solved in time $O(n^{(1-\delta)k})$,*

## B  Related works

The most similar prior works on attention mechanisms which are more expressive than self-attention are Sanford et al. (2024b) and Kozachinskiy et al. (2025), which we have already discussed in detail. There is another attention mechanism, *triangular attention*, introduced by Bergen et al. (2021), whose design was inspired by logic programming, and which was shown to perform better than self-attention on certain compositional tasks. However, Kozachinskiy et al. (2025) proved that it cannot perform function composition.

As we have discussed, the self-attention mechanism (Vaswani et al., 2017) is at the center of all large language models because of its expressivity in real-life applications, but the quadratic time complexity for computing its output is sometimes already prohibitively expensive. One extensive line of work has introduced faster *heuristic* algorithms, which work well on many inputs. These have used different approximation techniques, including sparsity assumptions, norm bounds, and kernel density estimation (Zandieh et al., 2023; Han et al., 2024; Kitaev et al., 2020; Choromanski et al., 2021; Pagliardini et al., 2023; Child et al., 2019; Wang et al., 2020; Daras et al., 2020; Katharopoulos et al., 2020; Chen et al., 2021; 2022; Qin et al., 2022; Liu et al., 2023b; He et al., 2021; Kacham et al., 2024; Dao et al., 2022; Dao, 2024; Roy et al., 2021; Sun et al., 2021; Ding et al., 2023; Han et al., 2023; Zaheer et al., 2020; Dass et al., 2023).

Other alternatives have been considered which completely replace attention with different mechanisms. A simple example is Hardmax attention, in which the softmax is replaced by a (hard) max, but training Hardmax attention Transformer models appears difficult as we do not know an efficient way to perform gradient descent. The power of hardmax has been explored in Alcalde et al. (2024); Pérez et al. (2021); Kajitsuka & Sato (2024). Instead of computing the output of self-attention faster, some other alternatives to Transformers have been proposed that completely replace attention with other mechanisms; examples include Synthesizer (Tay et al., 2021), routing Transformers (Roy et al., 2021), and Mamba (Gu & Dao, 2024). These alternatives can typically be computed much faster than attention (often in almost linear time by definition), but in exchange appear to have weaker expressive power (Alman & Yu, 2025). This paper continues a long line of work on understanding the power and limitations of Transformers, and finding more expressive alternative models.

Some papers have studied the circuit complexity of Transformers (Chiang, 2024; Merrill & Sabharwal, 2023a; Merrill et al., 2022b; Chen et al., 2024; Merrill & Sabharwal, 2023b; Merrill et al., 2022a; Chiang et al., 2023). Other works on the representational strength of Transformers focus on their relationship with other models of computation. For example, a line of work has studied the ability of Transformers to approximate other models of computation (Pérez et al., 2021; Wei et al., 2022a; Malach, 2023; Liu et al., 2023a; Hao et al., 2022). On the other hand, there are many more tasks, beyond those discussed here, which are difficult to solve by a Transformer, including compositional reasoning (Dekker et al., 2022; Zerroug et al., 2022; Marcus, 2018; Kozachinskiy, 2024; Sanford et al., 2024a; Peng et al., 2024).

Another approach to overcoming the limitations of Transformers is to augment them in other ways. An important example is chain-of-thought (Wei et al., 2022b). Merrill & Sabharwal (2024) studied the space and time complexity of chain-of-thought, and Peng et al. (2024) studied how this relates to function composition.

## C  WARM-UP: STRASSEN-ATTENTION UPPER AND LOWER BOUNDS

As a warm-up, we describe the polynomial method and show Max-2SAT-based hardness results on Strassen-attention. Since Strassen-attention is only a special case of poly-attention, we will later move on to show similar algorithms and lower bounds on poly-attention in Section E.

### C.1  ALGORITHM FOR STRASSEN-ATTENTION

In this section, we give a near-linear algorithm for computing an entry-wise approximation of the output matrix of Strassen-attention, when the entries of the query-key matrices are bounded. We will use the polynomial method, which has been used in entry-wise approximations of other attention mechanisms as well, like in self-attention Alman & Song (2023), tensor-attention Alman & Song (2024), RoPE based attention Alman & Song (2025).

Our goal is to compute the $n \times d$ matrix $Att^{(S)}$, the output of Strassen-attention, for query-key matrices $Q^{(1)}, Q^{(2)}, Q^{(3)} \in [-\Gamma, \Gamma]^{n \times d}$ and value matrices $V^{(1)}, V^{(2)} \in \mathbb{R}^{n \times d}$. Using the expression of Strassen-attention Kozachinskiy et al. (2025), it can also be written as

$$Att_{i,\ell}^{(S)} = \frac{[\frac{1}{d}Q^{(1)}(Q^{(2)})^T]_{(i,1:n)}^e D^{1,\ell}[\frac{1}{d}Q^{(2)}(Q^{(3)})^T]^e D^{2,\ell}[\frac{1}{d}Q^{(3)}(Q^{(1)})^T]_{(1:n,i)}^e}{[\frac{1}{d}Q^{(1)}(Q^{(2)})^T]_{(i,1:n)}^e[\frac{1}{d}Q^{(2)}(Q^{(3)})^T]^e[\frac{1}{d}Q^{(3)}(Q^{(1)})^T]_{(1:n,i)}^e \mathbf{1}_n}, \tag{2}$$

for all $i \in [n], \ell \in [d]$, where $D^{1,\ell} = diag(V_{(1:n,\ell)}^{(1)})$ and $D^{2,\ell} = diag(V_{(1:n,\ell)}^{(2)})$.

We will compute the entry-wise approximations of the numerator and the denominator terms of Equation 2 separately. The main idea is to use a low rank entry-wise approximations for each of $[\frac{1}{d}Q^{(1)}(Q^{(2)})^T]^e, [\frac{1}{d}Q^{(2)}(Q^{(3)})^T]^e, [\frac{1}{d}Q^{(3)}(Q^{(1)})^T]^e$, and multiply the low rank matrices together–something that can be done more efficiently. In order to obtain the low rank approximations, we will use the following lemma from Aggarwal & Alman (2022).

**Lemma C.1** (Aggarwal & Alman (2022)). *Let $\Gamma > 1$, $\varepsilon \in (0, 0.1)$. There exists a polynomial $P(x) \in \mathbb{R}[x]$ of degree $t := \Theta\left(\max\left\{\frac{\log(1/\varepsilon)}{\log(\log(1/\varepsilon)/\Gamma)}, \Gamma\right\}\right)$ such that for all $a \in [-\Gamma, \Gamma]$, we have $|P(a) - e^a| < \varepsilon e^a$. Furthermore, $P$ can be computed in $\mathrm{poly}(t)$ time and its coefficients are rational numbers.*

Using the previous lemma, we obtain the low rank matrix approximations as a corollary.

**Lemma C.2** (Low rank approximation Alman & Song (2023; 2024)). *Let $\varepsilon = 1/\mathrm{poly}(n)$, $d = O(\log n)$, $r = n^{o(1)}$, and $B = o(\log n)$. Given matrices $P, Q \in [-\Gamma, \Gamma]^{n \times d}$, we can compute matrices $U, W \in \mathbb{R}^{n \times r}$ in time $O(n^{1+o(1)})$ such that $UW^T$ entry-wise $\varepsilon$-approximates $PQ^T$; that is: $|[UW^T]_{i,j} - [PQ^T]_{i,j}^e| < \varepsilon[PQ^T]_{i,j}^e$.*

This is an instance of the Gaussian KDE which has widely been used in LLMs and machine learning algorithms Zandieh et al. (2023); Backurs et al. (2018); Katharopoulos et al. (2020); Alman et al. (2020); Aggarwal & Alman (2022); Alman & Song (2023; 2024).

We will show that we can compute $\forall i \in [n]$, $\gamma$-approximations of denominator in Equation 2 in time $O(n^{1+o(1)})$, and fixing any $\ell \in [d]$, we can compute $\gamma$-approximations of the numerator in time $O(n^{1+o(1)})$, $\forall i \in [n]$, where $\gamma = 1/\mathrm{poly}(n)$. Once we find the values of the denominator and the numerator, we perform a division, to compute the $\gamma$-approximation $\widehat{Att^{(S)}}_{i,\ell}$, which takes a total time of $O(n^{1+o(1)} + d.n^{1+o(1)} + nd) = O(n^{1+o(1)})$. Using this as the central idea, we prove the following result. Since Strassen-attention is a special case of poly-attention with the polynomial $h_S(x_1, x_2, x_3) = x_1x_2 + x_2x_3 + x_3x_1$, we state the following result:

**Theorem C.3.** *There is an algorithm that solves $\mathsf{APAC}^{(h_S)}(n, d = O(\log n), \Gamma = o(\sqrt{\log n}), \gamma = 1/\mathrm{poly}(n))$ with query-key matrices $Q^{(1)}, Q^{(2)}, Q^{(3)} \in [-\Gamma, \Gamma]^{n \times d}$, and value matrices $V^{(2)}, V^{(t)} \in \mathbb{R}^{n \times d}$ in time $O(n^{1+o(1)})$.*

The algorithm is summed up as follows.

---

**Algorithm 1** Algorithm to compute entry-wise approximation of $Att^{(S)}$

---

**Input:** A number $\Gamma = o(\sqrt{\log n})$, query-key matrices $Q^{(1)}, Q^{(2)}, Q^{(3)} \in [-\Gamma, \Gamma]^{n \times d}$, value matrices $V^{(1)}, V^{(2)} \in \mathbb{R}^{n \times d}$, an approximation parameter $\gamma = 1/\text{poly}(n)$.

**Output:** Entry-wise $\gamma$-approximation $\widehat{Att^{(S)}}$ of $Att^{(S)}$.

1: Initialize $\widehat{Att^{(S)}} := \mathbf{0}_{n \times d}$.
2: Compute the low-rank $\gamma$-approximations $U^1(W^1)^T$ of $[\frac{1}{d}Q^{(1)}(Q^{(2)})^T]^e$, $U^2(W^2)^T$ of $[\frac{1}{d}Q^{(2)}(Q^{(3)})^T]^e$ and $U^3(W^3)^T$ of $[\frac{1}{d}Q^{(3)}(Q^{(1)})^T]^e$ using Lemma C.2, where $U^1, W^1, U^2, W^2, U^3, W^3 \in \mathbb{R}^{n \times r}$ for $r = n^{o(1)}$. ▷ $O(n^{1+o(1)}r)$ time.
3: $D^{1,\ell} := diag(V^{(1)}_{(1:n,\ell)})$, $D^{2,\ell} := diag(V^{(2)}_{(1:n,\ell)})$. ▷ $O(n)$ time.
4: Compute $\tilde{U}^2 := D^{1,\ell}U^2$, $\tilde{W}^2 := D^{2,\ell}W^2 \in \mathbb{R}^{n \times r}$. ▷ $O(nr)$ time.
5: Compute $A := \underbrace{(W^1)^T U^2}_{r \times r} \underbrace{(W^2)^T U^3}_{r \times r}$ and $B := \underbrace{(W^1)^T \tilde{U}^2}_{r \times r} \underbrace{(\tilde{W}^2)^T U^3}_{r \times r}$. ▷ $O(nr^2)$ times.
6: **for** $i \in [n], \ell \in [d]$ **do**
7:     Compute the $\Theta(\gamma)$-approximation of the denominator (Equation 2) as

$$R_i := U^1_{(i,1:r)} A (W^3_{(i,1:r)})^T \in \mathbb{R}.$$

▷ $O(r^2)$ time.

8:     Compute the $\Theta(\gamma)$-approximation of the numerator (Equation 2) as

$$P_i^\ell := U^1_{(i,1:r)} B (W^3_{(i,1:r)})^T \in \mathbb{R}.$$

▷ $O(r^2)$ time.

9:     Compute the $\ell$-th row of the entry-wise $\Theta(\gamma)_a$-approximation of $Att^{(S)}$ as

$$\widehat{Att^{(S)}}[i, \ell] := \frac{P_i^\ell}{Q_i}.$$

▷ $O(1)$ time.

10: **end for**
11: **Return** $\widehat{Att^{(S)}}$.

---

Before proving the correctness of this algorithm, we first show that the entries of the exponentiated matrices are bounded, which is necessary for applying Lemma C.2.

**Lemma C.4** (Bounded entries). *The entries of $[\frac{1}{d}Q^{(1)}(Q^{(2)})^T]^e, [\frac{1}{d}Q^{(2)}(Q^{(3)})^T]^e, [\frac{1}{d}Q^{(3)}(Q^{(1)})^T]^e$ are bounded as*

$$e^{-\Gamma^2} \leq [\frac{1}{d}Q^{(1)}(Q^{(2)})^T]^e_{i,j}, [\frac{1}{d}Q^{(2)}(Q^{(3)})^T]^e_{i,j}, [\frac{1}{d}Q^{(3)}(Q^{(1)})^T]^e_{i,j} \leq e^{\Gamma^2},$$

*for all $i, j \in [n]$.*

*Proof.* Without loss of generality, we prove the upper bound only for $X$ and the rest follows similarly. Since each entry of $Q^{(1)}, Q^{(2)}, Q^{(3)}$ are in $[-\Gamma, \Gamma]$, the value of $[Q^{(1)}(Q^{(2)})^T]_{i,j}$ is

$$[Q^{(1)}(Q^{(2)})^T]_{i,j} = \langle Q^{(1)}_i, Q^{(2)}_j \rangle = \sum_{\ell \in [d]} Q^{(1)}_{i,\ell} Q^{(2)}_{j,\ell},$$

$$\implies -\Gamma^2 \leq \langle Q^{(1)}_i, Q^{(2)}_j \rangle / d \leq \Gamma^2 \quad \text{(Since } -\Gamma \leq Q^{(1)}_{i,\ell}, Q^{(2)}_{j,\ell} \leq \Gamma\text{)}.$$

Therefore, $e^{-\Gamma^2} \leq [\frac{1}{d}Q^{(1)}(Q^{(2)})^T]^e_{i,j} \leq e^{\Gamma^2}$ for all $i, j \in [n]$, and we can similarly bound $[\frac{1}{d}Q^{(2)}(Q^{(3)})^T]^e, [\frac{1}{d}Q^{(3)}(Q^{(1)})^T]^e$. □

Now, we prove Theorem C.3, which is also the correctness of Algorithm 1.

*Proof of Theorem C.3.* First, we compute the low-rank approximations of $[\frac{1}{d}Q^{(1)}(Q^{(2)})^T]^e, [\frac{1}{d}Q^{(2)}(Q^{(3)})^T]^e, [\frac{1}{d}Q^{(3)}(Q^{(1)})^T]^e$ using Lemma C.2 (Step 2 of Algorithm 1). However, in order for Lemma C.2 to succeed in Step 2 of Algorithm 1, we need the entries of the exponentiated matrices to be bounded, which is true due to Lemma C.4.

We compute the Strassen-attention matrix in two steps, first computing the denominator, and then the numerator in Equation 2 to compute the entire self-attention matrix.

**Computing the denominator.** This has been described in Step 7 of Algorithm 1, and we now prove its correctness. Since the entries of $[\frac{1}{d}Q^{(1)}(Q^{(2)})^T]^e, [\frac{1}{d}Q^{(2)}(Q^{(3)})^T]^e, [\frac{1}{d}Q^{(3)}(Q^{(1)})^T]^e$ are bounded, we can apply Lemma C.2 to find their low rank approximations. Let the low-rank approximations of $[\frac{1}{d}Q^{(1)}(Q^{(2)})^T]^e, [\frac{1}{d}Q^{(2)}(Q^{(3)})^T]^e, [\frac{1}{d}Q^{(3)}(Q^{(1)})^T]^e$ be $U^1(W^1)^T$, $U^2(W^2)^T$ and $U^3(W^3)^T$ respectively, with entry-wise error $\varepsilon$ for $\varepsilon = 1/\text{poly}(n)$, where each of $U^i, W^i \in \mathbb{R}^{n \times r}$. Namely, for all $i, j, k \in [n]$,

$$
\left| [\frac{1}{d}Q^{(1)}(Q^{(2)})^T]^e_{i,j} - [U^1(W^1)^T]_{i,j} \right| < \varepsilon[\frac{1}{d}Q^{(1)}(Q^{(2)})^T]^e_{i,j} < \gamma,
$$

$$
\left| [\frac{1}{d}Q^{(2)}(Q^{(3)})^T]^e_{j,k} - [U^2(W^2)^T]_{j,k} \right| < \varepsilon[\frac{1}{d}Q^{(2)}(Q^{(3)})^T]^e_{j,k} < \gamma, \tag{3}
$$

$$
\left| [\frac{1}{d}Q^{(3)}(Q^{(1)})^T]^e_{k,i} - [U^3(W^3)^T]_{k,i} \right| < \varepsilon[\frac{1}{d}Q^{(3)}(Q^{(1)})^T]^e_{k,i} < \gamma.
$$

where $\gamma = \varepsilon e^{\Gamma^2}$. When we choose $\varepsilon$ as the inverse of a large enough polynomial such that $\varepsilon e^{\Gamma^2} = \frac{1}{\text{poly}(n)}$, we have $\gamma = 1/\text{poly}(n)$ (note that $\Gamma = O(\sqrt{\log n})$). Now, we claim that

$$
[U^1(W^1)^T U^2(W^2)^T U^3(W^3)^T]_{i,i}
$$

is an approximation of $[[\frac{1}{d}Q^{(1)}(Q^{(2)})^T]^e[\frac{1}{d}Q^{(2)}(Q^{(3)})^T]^e[\frac{1}{d}Q^{(3)}(Q^{(1)})^T]^e]_{i,i}$. For ease of notation, let us denote $X = [\frac{1}{d}Q^{(1)}(Q^{(2)})^T]^e, Y = [\frac{1}{d}Q^{(2)}(Q^{(3)})^T]^e, Z = [\frac{1}{d}Q^{(3)}(Q^{(1)})^T]^e$. Now,

$$
\begin{aligned}
&|[XYZ]_{i,i} - [U^1(W^1)^T U^2(W^2)^T U^3(W^3)^T]_{i,i}| \\
&= \left| \left([XYZ]_{i,i} - [U^1(W^1)^T YZ]_{i,i}\right) + \left([U^1(W^1)^T YZ]_{i,i} - [U^1(W^1)^T U^2(W^2)^T Z]_{i,i}\right) \right. \\
&\quad \left. + \left([U^1(W^1)^T U^2(W^2)^T Z]_{i,i} - [U^1(W^1)^T U^2(W^2)^T U^3(W^3)^T]_{i,i}\right) \right| \\
&\leq \left| [XYZ]_{i,i} - [U^1(W^1)^T YZ]_{i,i} \right| + \left| [U^1(W^1)^T YZ]_{i,i} - [U^1(W^1)^T U^2(W^2)^T Z]_{i,i} \right| \\
&\quad + \left| [U^1(W^1)^T U^2(W^2)^T Z]_{i,i} - [U^1(W^1)^T U^2(W^2)^T U^3(W^3)^T]_{i,i} \right|,
\end{aligned} \tag{4}
$$

where the last inequality follows from triangle inequality.

Now, using Equation 3 in each of the three terms, we can show that this is bounded above by $O(\gamma)$.

The computation of $[U^1(W^1)^T U^2(W^2)^T U^3(W^3)^T]_{i,i}$ for $i \in [n]$ from Algorithm 1, takes $O(n^{1+o(1)})$ time for $r = n^{o(1)}$ (which is true for the choice of $d = O(\log n), B = o(\sqrt{\log n}), \gamma = 1/\text{poly}(n)$ using the parameters of Lemma C.2).

**Computing the numerator.** An entry-wise $\gamma$-approximation of the numerator of the Strassen-attention matrix $Att^{(S)} \in \mathbb{R}^{n \times d}$ (Equation 2) has been computed in Step 8 of Algorithm 1. Here, essentially, we compute each entry $[XD^{1,\ell}YD^{2,\ell}Z]_{i,i}$, for all $i \in [n]$ by fixing $\ell \in [d]$ at a time.

We again make use of the low rank decompositions of $X, Y, Z$ as above (Equation 3). Note that the value of each element of $Att^{(S)}$ is given as

$$
Att^{(S)}_{i,\ell} = [XD^{1,\ell}YD^{2,\ell}Z]_{i,i}.
$$

We claim that $[U^1(W^1)^T D^{1,\ell} U^2(W^2)^T D^{2,\ell} U^3(W^3)^T]_{i,i}$ is an $O(\gamma)$-approximation of $[XD^{1,\ell}YD^{2,\ell}Z]_{i,i}$. Indeed, we have

$$
|[XD^{1,\ell}YD^{2,\ell}Z]_{i,i} - [U^1(W^1)^T D^{1,\ell} U^2(W^2)^T D^{2,\ell} U^3(W^3)^T]_{i,i}|
$$

$$
= \left| \left( [XD^{1,\ell}YD^{2,\ell}Z]_{i,i} - [U^1(W^1)^T D^{1,\ell} Y D^{2,\ell} Z]_{i,i} \right) \right.
$$

$$
+ \left( [U^1(W^1)^T D^{1,\ell} Y D^{2,\ell} Z]_{i,i} - [U^1(W^1)^T D^{1,\ell} U^2(W^2)^T D^{2,\ell} Z]_{i,i} \right)
$$

$$
+ \left. \left( [U^1(W^1)^T D^{1,\ell} U^2(W^2)^T D^{2,\ell} Z]_{i,i} - [U^1(W^1)^T D^{1,\ell} U^2(W^2)^T D^{2,\ell} U^3(W^3)^T]_{i,i} \right) \right| \quad (5)
$$

$$
\leq \left| [XD^{1,\ell}YD^{2,\ell}Z]_{i,i} - [U^1(W^1)^T D^{1,\ell} Y D^{2,\ell} Z]_{i,i} \right|
$$

$$
+ \left| [U^1(W^1)^T D^{1,\ell} Y D^{2,\ell} Z]_{i,i} - [U^1(W^1)^T D^{1,\ell} U^2(W^2)^T D^{2,\ell} Z]_{i,i} \right|
$$

$$
+ \left| [U^1(W^1)^T D^{1,\ell} U^2(W^2)^T D^{2,\ell} Z]_{i,i} - [U^1(W^1)^T D^{1,\ell} U^2(W^2)^T D^{2,\ell} U^3(W^3)^T]_{i,i} \right|,
$$

which again follows from the triangle inequality, and each term can be shown to be upper bounded by $O(\gamma)$ using Equation 3.

**Wrapping up.** An approximation of the $(i,\ell)$-th element, $Att_{i,\ell}^{(S)}$, is obtained by approximating the value of $[XD^{1,\ell}YD^{2,\ell}Z]_{i,i}$ and then dividing by the approximate value of $[XYZ]_{i,i}$. Using

$$
P_i^\ell = U^1(W^1)^T D^{1,\ell} U^2(W^2)^T D^{2,\ell} U^3(W^3)^T,
$$

$$
R_i = U_i^1 \left( (W^1)^T U^2 (W^2)^T U^3 \right) (W_i^3)^T,
$$

we have

$$
|[XD^{1,\ell}YD^{2,\ell}Z]_{i,i} - P_i^\ell| \leq O(\gamma),
$$

and,

$$
|[XYZ]_{i,i} - R_i|_\infty \leq O(\gamma),
$$

for $i \in [n], \ell \in [d]$.

Therefore, the error is given by

$$
|[XYZ]_{i,i}^{-1}[XD^{1,\ell}YD^{2,\ell}Z]_{i,i} - R_i^{-1}P_i^\ell| \leq |[XYZ]_{i,i}^{-1}[XD^{1,\ell}YD^{2,\ell}Z]_{i,i} - [XYZ]_{i,i}^{-1}P_i^\ell|
$$

$$
+ |[XYZ]_{i,i}^{-1}P_i^\ell - R_i^{-1}P_i^\ell| \text{ (Triangle inequality)}
$$

$$
\leq O(\gamma),
$$

which follows from Equations 4, 5, repeated applications of triangle inequalities, and the fact that $\varepsilon$ is an inverse polynomial in $n$, and,

$$
|[XD^{1,\ell}YD^{2,\ell}Z]_{i,i}| = \left| \sum_{j,k \in [n]} X_{i,j} V_{j,\ell}^1 Y_{j,k} V_{k,\ell}^2 Z_{k,i} \right| < e^{3\Gamma^2} ||V^1||_\infty ||V^2||_\infty,
$$

$$
\left| \frac{1}{[XYZ]_{i,i}} \right| = \left| \frac{1}{\sum_{j,k \in [n]} X_{i,j} Y_{j,k} Z_{k,i}} \right| < e^{3\Gamma^2},
$$

for all $i \in [n], \ell \in [d]$ since the entries of $Q^{(1)}, Q^{(2)}, Q^{(3)}$ are in $[-\Gamma, \Gamma]$ (Lemma C.4). For $d = O(\log n)$, $\Gamma = o(\sqrt{\log n})$ and $||V^{(1)}||_\infty, ||V^{(2)}||_\infty = \text{poly}(n)$, we can choose $\gamma_0 = 1/\text{poly}(n)$ for a large enough polynomial such that

$$
\left| \frac{P_i^\ell}{R_i} - Att_{i,\ell}^{(S)} \right| < \gamma_0,
$$

where $\gamma_0 = O(\gamma) = 1/\text{poly}(n)$, which is our required approximation parameter.

As described in Algorithm 1 we compute this $\gamma$-approximation for all $i \in [n]$ in time $O(n^{1+o(1)})$, and repeating this over all $\ell \in [d]$ requires $O(n^{1+o(1)}d) = O(n^{1+o(1)})$ time since $d = O(\log n)$. This proves Theorem C.3. $\qquad\square$

## C.2 HARDNESS OF STRASSEN-ATTENTION

Now, we introduce the techniques that will be used to prove lower bounds in this paper. We establish the hardness of Strassen-attention in the high weight case, assuming the Max-2SAT conjecture (Hypothesis 3). Our reduction will proceed in three steps. First, we use a reduction from Alman & Vassilevska Williams (2020) that establishes the hardness of IP$\Delta$ (Definition C.5) assuming Hypothesis 3 (hardness of Max-2SAT). Second, we prove the hardness of $\varepsilon$-Gap-IP$\Delta$ (an approximate version of IP$\Delta$ defined in Definition C.6) from the hardness of IP$\Delta$, in Section C.2.1. Lastly, we prove the hardness of Strassen-attention from the hardness of $\varepsilon$-Gap-IP$\Delta$ in Section C.2.2.

We begin by defining the problems IP$\Delta$ and $\varepsilon$-Gap-IP$\Delta$.

**Definition C.5** (IP$\Delta$). *Given three sets of vectors $A^1, A^2, A^3 \subseteq \{0,1\}^d$, $|A^1| = |A^2| = |A^3| = n$, and target inner products, $m_{12}, m_{23}, m_{31} \in \{0, \ldots, d\}$, the problem $\mathsf{IP}\Delta_{n,d}(A^1, A^2, A^3, m_{12}, m_{23}, m_{31})$ asks whether there exist vectors $a_1 \in A^1, a_2 \in A^2, a_3 \in A^3$ such that, simultaneously, $\langle a_1, a_2 \rangle = m_{12}, \langle a_2, a_3 \rangle = m_{23}, \langle a_3, a_1 \rangle = m_{31}$.*

**Definition C.6** ($\varepsilon$-Gap-IP$\Delta$). *Let $\varepsilon > 0$. Given three sets of vectors $A^1, A^2, A^3 \subseteq \{0,1\}^d$, with $|A^1| = |A^2| = |A^3| = n$, a target inner product $m \in \{0, \ldots, d\}$, and the promise that for every $a_1 \in A^1, a_2 \in A^2, a_3 \in A^3$,*

- *either $\langle a_1, a_2 \rangle \leq (1 - \varepsilon)m$ or $\langle a_1, a_2 \rangle = m$,*

- *and, either $\langle a_2, a_3 \rangle \leq (1 - \varepsilon)m$ or $\langle a_2, a_3 \rangle = m$,*

- *and, either $\langle a_3, a_1 \rangle \leq (1 - \varepsilon)m$ or $\langle a_3, a_1 \rangle = m$,*

*the problem $\varepsilon$-**Gap-IP**$\Delta_{n,d}(A^1, A^2, A^3, m)$ is to decide if there exist vectors $a_1 \in A^1, a_2 \in A^2, a_3 \in A^3$ such that: $\langle a_1, a_2 \rangle = \langle a_2, a_3 \rangle = \langle a_3, a_1 \rangle = m$.*

For IP$\Delta$ and $\varepsilon$-Gap-IP$\Delta$, we will drop the parameters $m, d$ when they are clear from context. Note that even though IP$\Delta$ might have different inner products for all three pairs, for $\varepsilon$-Gap-IP$\Delta$, the three inner products being equal suffices as the reduction for proving its hardness accommodates this property, and for proving hardness of approximating the output of Strassen-attention, we need them to be equal.

As mentioned above, the first step uses a result due to Alman & Vassilevska Williams (2020) which proved that IP$\Delta$ is at least as hard as Max-2SAT:

**Lemma C.7** (Alman & Vassilevska Williams (2020)). *Assuming the Max-2SAT conjecture (Hypothesis 3), for every $\delta > 0$ there exists $c > 0$ such that $\mathsf{IP}\Delta_{n,c\log n}$ cannot be solved in time $O(n^{\omega - \delta})$.*

### C.2.1 CONDITIONAL HARDNESS OF $\varepsilon$-GAP-IP$\Delta$

In this subsection we prove the following theorem, establishing hardness of $\varepsilon$-Gap-IP$\Delta$ assuming hardness of IP$\Delta$.

**Theorem C.8.** *For every $\delta, \varepsilon > 0$, there exists $c, c' > 0$ such that if $\varepsilon$-**Gap-IP**$\Delta_{n,c\log n}$ can be solved in time $\tilde{O}(n^{(\omega - \delta)})$, then $\mathsf{IP}\Delta_{n,c'\log n}$ can be solved in time $\tilde{O}(n^{(\omega - \delta/2)})$.*

Building on Aaronson & Wigderson (2009), Rubinstein Rubinstein (2018) gave a reduction from the IP problem to the gap version, $\varepsilon$-Gap-IP. That is, they proved a similar reduction to what we want, but where IP and $\varepsilon$-Gap-IP take as input two sets $A^1, A^2$ instead of three sets. Chen & Williams (2019); Abboud & Ron-Zewi (2025) further improved their reduction; for our reductions, we will use and build upon the proof given by Abboud & Ron-Zewi (2025).

The following lemma was proven in Abboud & Ron-Zewi (2025) (see the proofs of Lemma 4.1 and Claim 4.3 in their paper).

**Lemma C.9** (Abboud & Ron-Zewi (2025)). *For all $n, d = O(\log n)$, there exists $d' = O(d)$, $q = n^{o(1)}$, $m' = O(\log n)$, such that for every instance of $\mathsf{IP}_{n,d}$ given by sets of vectors $A, B$ and a target inner product $m \in \{0, \ldots, d\}$, there is a set of $q$ instances $\{(\tilde{A}^i, \tilde{B}^i, m') \mid i \in [q]\}$ of $\varepsilon$-**Gap-IP**$_{n,d'}$ computable in $O(n^{1+o(1)})$ time, where $\varepsilon \in (0, 1)$ is a constant such that:*

1. *(Yes case) If there exists $(a, b) \in A \times B$ such that $\langle a, b \rangle = m$, then there exists $i \in [q]$ such that $(\tilde{A}^i, \tilde{B}^i, m')$ is a yes instance of $\varepsilon$-Gap-IP$_{n,d'}$.*

2. *(No case) If for every pair $(a, b) \in A \times B$, $\langle a, b \rangle \neq m$, then for all $i \in [q]$, $(\tilde{A}^i, \tilde{B}^i, m')$ is a no instance of $\varepsilon$-Gap-IP$_{n,d'}$.*

*Proof of Theorem C.8.* We start with an instance of IP$\Delta_{n,d=O(\log n)}$, given by a target inner product $m$, and matrices $A, B, C$ each of dimension $n \times d$, where the rows of $A$ correspond to a set of $n$ vectors, and similarly for $B$ and $C$.

For the pair $(A, B)$, we apply Lemma C.9 to create a set of $q$ instances of $\varepsilon$-Gap-IP$_{n,d'}$, each with target inner product $m'$:

$$(\tilde{A}_{AB}, \tilde{B}_{AB}) = \{(\tilde{A}^i_{AB}, \tilde{B}^i_{AB}, ) \mid i \in [q]\}.$$

Similarly we apply the Lemma to the pair $(B, C)$ to get $\varepsilon$-Gap-IP instances

$$(\tilde{B}_{BC}, \tilde{C}_{BC}) = \{(\tilde{B}^i_{BC}, \tilde{C}^i_{BC}) \mid i \in [q]\}$$

and to the pair $(A, C)$ to get instances

$$(\tilde{A}_{AC}, \tilde{C}_{AC}) = \{(\tilde{A}^i_{AC}, \tilde{C}^i_{AC}) \mid i \in [q]\}.$$

By Lemma C.9, the following properties are satisfied by $(\tilde{A}_{AB}, \tilde{B}_{AB})$:

(1) For all $i \in [q]$, the instance $(\tilde{A}^i_{AB}, \tilde{B}^i_{AB}, m')$ satisfies the gap property. That is, for every $a^i_{AB} \in \tilde{A}^i_{AB}, b^i_{AB} \in \tilde{B}^i_{AB}$, $\langle a^i_{AB}, b^i_{AB} \rangle$ is either equal to $m'$ or is at most $(1 - \varepsilon)m'$.

(2) Correctness of the reduction:

    (2b) If there exists $(a, b) \in A \times B$ such that $\langle a, b \rangle = m$, then there exists $i \in [q]$, and vectors $a^i_{AB} \in \tilde{A}^i_{AB}, b^i_{AB} \in \tilde{B}^i_{AB}$ such that $\langle a^i_{AB}, b^i_{AB} \rangle = m'$.

    (2c) If for every $(a, b) \in A \times B$, $\langle a, b \rangle \neq m$, then for all $i \in [q]$, and for all vectors $a^i_{AB} \in \tilde{A}^i_{AB}, b^i_{AB} \in \tilde{B}^i_{AB}$, we have $\langle a^i_{AB}, b^i_{AB} \rangle \leq (1 - \varepsilon)m'$.

By the same argument, the above two properties are also satisfied by $(\tilde{B}_{BC}, \tilde{C}_{BC})$ and $(\tilde{A}_{AC}, \tilde{C}_{AC})$.

Equipped with the above pairs of 3-dimensional tensors, we are now ready to describe our reduction from the instance $(A, B, C, m)$ of IP$\Delta_{n,d}$ to a set of $q^3$ instances of $\varepsilon$-Gap-IP$\Delta_{n,O(\log n)}$, denoted by:

$$(\mathcal{A}, \mathcal{B}, \mathcal{C}) = \{(\mathcal{A}^{i,j,k}, \mathcal{B}^{i,j,k}, \mathcal{C}^{i,j,k}) \mid i, j, k \in [q]\}$$

For each $i, j, k \in [q]$, we define $\mathcal{A}^{i,j,k}$ to consist of the following set of length $3d'$ vectors:

$$\mathcal{A}^{i,j,k} = \{a^i_{AB} \, 0^d \, a^k_{AC} \mid a^i_{AB} \in \tilde{A}^i_{AB}, \, a^k_{AC} \in \tilde{A}^k_{AC}\}$$

Similarly we define $\mathcal{B}^{i,j,k}$ and $\mathcal{C}^{i,j,k}$:

$$\mathcal{B}^{i,j,k} = \{b^i_{AB} \, b^j_{BC} \, 0^d \mid b^i_{AB} \in \tilde{B}^i_{AB}, \, b^j_{BC} \in \tilde{B}^j_{BC}\}$$

$$\mathcal{C}^{i,j,k} = \{0^d \, c^j_{BC} \, c^k_{AC} \mid c^j_{BC} \in \tilde{C}^j_{BC}, \, c^k_{AC} \in \tilde{C}^k_{AC}\}$$

**Gap Property** First, we prove that every instance $(\mathcal{A}^{i,j,k}, \mathcal{B}^{i,j,k}, \mathcal{C}^{i,j,k})$ satisfies the gap property. Consider a generic triple $(a^{i,j,k}, b^{i,j,k}, c^{i,j,k}) \in \mathcal{A}^{i,j,k} \times \mathcal{B}^{i,j,k} \times \mathcal{C}^{i,j,k}$, where

$$a^{i,j,k} = a^i_{AB} 0^{d'} a^k_{AC},$$

$$b^{i,j,k} = b^i_{AB} b^j_{BC} 0^{d'},$$

$$c^{i,j,k} = 0^{d'} c^j_{BC} c^k_{AC}.$$

Since $\langle a^{i,j,k}, b^{i,j,k} \rangle = \langle a^i_{AB}, b^i_{AB} \rangle$, we can apply property (1) to $(\tilde{A}_{AB}, \tilde{B}_{AB})$ to infer that this inner product is either $m'$ or at most $(1-\varepsilon)m'$. By a similar argument we can show that $\langle b^{i,j,k}, c^{i,j,k} \rangle$ and $\langle a^{i,j,k}, c^{i,j,k} \rangle$ are either $m'$ or at most $(1-\varepsilon)m'$. This completes the proof of the gap property.

**Proof of Correctness.** We first consider the yes case, when there exists $(a, b, c) \in A \times B \times C$ such that $\langle a, b \rangle = \langle b, c \rangle = \langle a, c \rangle = m$. Applying property (2a) above, we have:

1. There exists $i \in [q]$, $a^i_{AB} \in \tilde{A}^i_{AB}$, $b^i_{AB} \in \tilde{B}^i_{AB}$ such that $\langle a^i_{AB}, b^i_{AB} \rangle = m'$.

2. There exists $j \in [q]$, $b^j_{BC} \in \tilde{B}^j_{BC}$, $c^k_{BC} \in \tilde{C}^j_{BC}$ such that $\langle b^j_{BC}, c^j_{BC} \rangle = m'$.

3. There exists $k \in [q]$, $a^k_{AC} \in \tilde{A}^k_{AC}$, $c^k_{AC} \in \tilde{C}^k_{AC}$ such that $\langle a^k_{AC}, c^k_{AC} \rangle = m'$.

Now consider the corresponding vectors $a^{i,j,k} \in \mathcal{A}^{i,j,k}$, $b^{i,j,k} \in \mathcal{B}^{i,j,k}$, and $c^{i,j,k}$ in $\mathcal{C}^{i,j,k}$ defined as:

$$a^{i,j,k} = a^i_{AB} 0^{d'} a^k_{AC}$$

$$b^{i,j,k} = b^i_{AB} b^j_{BC} 0^{d'}$$

$$c^{i,j,k} = 0^{d'} c^j_{BC} c^k_{AC}$$

By inspection together with the above three properties (1, 2, 3), we have

$$\langle a^{i,j,k}, b^{i,j,k} \rangle = \langle b^{i,j,k}, c^{i,j,k} \rangle = \langle a^{i,j,k}, c^{i,j,k} \rangle = m',$$

thus completing the "yes" case of correctness.

In the no case, suppose for all $(a, b, c) \in A \times B \times C$, either $\langle a, b \rangle \neq m$ or $\langle b, c \rangle \neq m$ or $\langle a, c \rangle \neq m$. We want to show that for all $i, j, k \in [q]$ and for all $(a^{i,j,k}, b^{i,j,k}, c^{i,j,k}) \in \mathcal{A}^{i,j,k} \times \mathcal{B}^{i,j,k} \times \mathcal{C}^{i,j,k}$, at least one of the following holds: (i) $\langle a^{i,j,k}, b^{i,j,k} \rangle \leq (1-\varepsilon)m'$ or (ii) $\langle b^{i,j,k}, c^{i,j,k} \rangle \leq (1-\varepsilon)m'$ or (iii) $\langle a^{i,j,k}, c^{i,j,k} \rangle \leq (1-\varepsilon)m'$.

Fix $i, j, k \in [q]$ and consider a generic triple $(a^{i,j,k}, b^{i,j,k}, c^{i,j,k})$ in $\mathcal{A}^{i,j,k} \times \mathcal{B}^{i,j,k} \times \mathcal{C}^{i,j,k}$, where

$$a^{i,j,k} = a^i_{AB} 0^{d'} a^k_{AB},$$

$$b^{i,j,k} = b^i_{AB} b^j_{BC} 0^{d'},$$

$$c^{i,j,k} = 0^d c^j_{BC} c^k_{BC}$$

Consider first the case where $\langle a, b \rangle \neq m$. Then by applying property (2b) to $(\tilde{A}_{AB}, \tilde{B}_{AB})$ we have $\langle a^i_{AB}, b^i_{AB} \rangle \leq (1-\varepsilon)m'$, and therefore $\langle a^{i,j,k}, b^{i,j,k} \rangle \leq (1-\varepsilon)m'$, so case (i) above holds.

Similarly in the second case where $\langle b, c \rangle \neq m$, applying property (2b) $(\tilde{B}_{BC}, \tilde{C}_{BC})$ it follows that $\langle b^{i,j,k}, c^{i,j,k} \rangle \leq (1-\varepsilon)m'$, so case (ii) holds. For the last case where $\langle a, c \rangle \neq m$ we can similarly use (2b) to show that (iii) holds.

This completes the proof of correctness of the reduction.

**Time complexity.** Assume we are able to solve $\varepsilon$-Gap-IP$\Delta$ in time $n^{\omega-\delta}$ for a constant $\delta > 0$. Then we can solve all $q^3$ instances $(\mathcal{A}^{i,j,k}, \mathcal{B}^{i,j,k}, \mathcal{C}^{i,j,k})$ of $\varepsilon$-Gap-IP$\Delta$ in time $q^3 n^{\omega-\delta}$. Since $q = n^{o(1)}$, $q^3 < n^{\delta/2}$ for $n$ sufficiently large, and thus the runtime of the (Turing) reduction from IP$\Delta$ to Gap-IP$\Delta$ is at most $n^{\omega-\delta/2}$. This completes the proof of Theorem C.8. $\qquad\square$

### C.2.2 Hardness of approximating Strassen-attention

In this subsection, we prove the following theorem which is the last step of our reduction for proving the lower bound. The following theorem gives an efficient reduction from $\varepsilon$-Gap-IP$\Delta$ to Strassen-attention when the weights are large. We again use the fact that Strassen-attention is poly-attention for the polynomial $h_S(x_1, x_2, x_3) = x_1 x_2 + x_2 x_3 + x_3 x_1$.

**Theorem C.10** (Hardness of Strassen-attention). *For every constant $\varepsilon > 0$, every $\delta \in (0, 0.01)$, every $c, M > 0$, there exist constants $C_a > 0$ and $C_b > 0$ such that if $\mathsf{APAC}^{(h_S)}(2n, 2c \log n, \Gamma = C_b \sqrt{\log n}, \gamma = n^{-C_a})$ (Definition A.6) with query-key matrices $Q^{(1)}, \ldots, Q^{(t)} \in [-\Gamma, \Gamma]^{2n \times 2c \log n}$, value matrices $V^{(2)}, \ldots, V^{(t)} \in \mathbb{R}^{2n \times 2c \log n}$ can be solved in time $O(n^{\omega-\delta})$, then $\varepsilon$-Gap-IP$\Delta_{n, c \log n}$ (Definition C.6) with target inner product $m = M \log n$ can also be solved in $O(n^{\omega-\delta})$ time.*

*Proof.* We start with an instance of $\varepsilon$-Gap-IP$\Delta_{n, d = c \log n}$, defined by sets $A, B, C \subseteq \{0, 1\}^d$, and target inner product $m = M \log n$ for a constant $M$, satisfying the promise given by the definition of $\varepsilon$-Gap-IP$\Delta$ (e.g., for every pair of vectors from different sets, their inner product is either equal to $m$ or at most $(1 - \varepsilon)m$). From this instance we now want to create an instance of Strassen attention, given by matrices $Q^{(1)}, Q^{(2)}.Q^{(3)}, V^{(1)}, V^{(2)}$.

Now, for a positive real number $B = \omega(1)$ that we will fix later, similar to Alman & Song (2023; 2024), we construct the matrices $Q^{(1)}, Q^{(2)}, Q^{(3)} \in \mathbb{R}^{\tilde{n} \times \tilde{d}}$ for $\tilde{n} = 2n, \tilde{d} = 2d$ as:

$$
Q^{(1)} = B \begin{bmatrix} a_1 & 1_d \\ \vdots & \vdots \\ a_n & 1_d \\ 0_d & 1_d \\ \vdots & \vdots \\ 0_d & 1_d \end{bmatrix}_{2n \times 2d}, \quad
Q^{(2)} = B \begin{bmatrix} b_1 & 0_d \\ \vdots & \vdots \\ b_n & 0_d \\ 0_d & 1_d \\ \vdots & \vdots \\ 0_d & 1_d \end{bmatrix}_{2n \times 2d}, \quad
Q^{(3)} = B \begin{bmatrix} c_1 & 0_d \\ \vdots & \vdots \\ c_n & 0_d \\ 0_d & 1_d \\ \vdots & \vdots \\ 0_d & 1_d \end{bmatrix}_{2n \times 2d}.
$$

We also define $V^{(1)}, V^{(2)} \in \mathbb{R}^{\tilde{n} \times \tilde{d}}$ whose first columns are

$$
V^{(1)}_{(1:2n,1)} = \begin{bmatrix} 1_n^T \\ 0_n^T \end{bmatrix}, \quad V^{(1)}_{(1:2n,1)} = \begin{bmatrix} 1_n^T \\ 0_n^T \end{bmatrix},
$$

and the remaining entries are zeros.

**Correctness of the construction.** We have defined the matrices $Q^{(1)}, Q^{(2)}, Q^{(3)}$ underlying Strassen-attention so that, for any $i, j, k \in [n]$ we will have $\langle Q_i^{(1)}, Q_j^{(2)} \rangle + \langle Q_j^{(2)}, Q_k^{(3)} \rangle + \langle Q_k^{(3)}, Q_i^{(1)} \rangle = B^2(\langle a_i, b_j \rangle + \langle b_j, c_k \rangle + \langle c_k, a_i \rangle)$, and the bottom half of the matrices, $Q_{(n+1:2n)}^{(1)}, Q_{(n+1:2n)}^{(2)}, Q_{(n+1:2n)}^{(3)}$, will act as a normalizing terms when we compute the softmax.

As before, computing the output of the Strassen-attention works in two steps: for all $i \in [\tilde{n}]$, we first calculate the value of the denominator $[XYZ]_{i,i}$, where $X = [\frac{1}{\tilde{d}} Q^{(1)}(Q^{(2)})^T]^e$, $Y = [\frac{1}{\tilde{d}} Q^{(2)}(Q^{(3)})^T]^e$ and $Z = [\frac{1}{\tilde{d}} Q^{(3)}(Q^{(1)})^T]^e$. The normalizing term will allow us to give similar upper and lower bounds on this. Next, we will compute the numerator, $[XD^{1,\ell}YD^{2,\ell}Z]_{i,i}$, for all $\ell \in [\tilde{d}]$, where $D^{1,\ell} = diag(V^{(1)}_{1:2n,\ell})$ and $D^{2,\ell} = diag(V^{(2)}_{1:2n,\ell})$. Our approach is to show that if there exists some $i \in [n]$ such that for some $j, k \in [n]$, we have $\langle a_i, b_j \rangle = M \log n$, $\langle b_j, c_k \rangle = M \log n$ and $\langle c_k, a_i \rangle = M \log n$, then we will be able to find such an $i$ using the entry-wise approximation of one Strassen-attention head. Thus, further improvements to the entry-wise approximation algorithm would imply an algorithm for solving $\varepsilon$-Gap-IP$\Delta$ in time $n^{\omega-\Omega(1)}$ time.

**Bounds on the denominator.** We analyze the denominator term and give upper and lower bounds on $[XYZ]_{i,i}$. For computing this value, we find the value of $\sum_{j,k \in [\tilde{n}]} \exp\left(\frac{1}{\tilde{d}}(\langle Q_i^{(1)}, Q_j^{(2)} \rangle + \langle Q_j^{(2)}, Q_k^{(3)} \rangle + \langle Q_k^{(3)}, Q_i^{(1)} \rangle)\right)$. We only care about the first

$n$ rows of the attention matrix as this is where the existence of an $\mathsf{IP}\Delta$ will be noticed. For $i \in [n]$, this is equivalent to computing

$$[XYZ]_{i,i} = \sum_{j,k\in[n]} e^{(\langle a_i,b_j\rangle+\langle b_j,c_k\rangle+\langle c_k,a_i\rangle)B^2/\tilde{d}} + \sum_{j\in[n+1:2n],k\in[n]} e^{(d+0+\langle c_k,a_i\rangle)B^2/\tilde{d}}$$
$$+ \sum_{j\in[n],k\in[n+1:2n]} e^{(\langle a_i,b_j\rangle+0+d)B^2/\tilde{d}} + \sum_{j,k\in[n+1:2n]} e^{(d+d+d)B^2/\tilde{d}}. \tag{6}$$

Using the gap property that the inner products of any pairs of $a_i, b_j, c_k$ are either less than $(1-\varepsilon)M\log n$ or exactly equal to $M\log n$, and denoting $\lambda := \frac{M\log n}{\tilde{d}}$ where $\tilde{d} = 2c\log n$, from the previous equation, we get

$$[XYZ]_{i,i} \geq \sum_{j,k\in[n]} e^{3(1-\varepsilon)\lambda B^2} + \sum_{j\in[n+1:2n],k\in[n]} e^{(1+(1-\varepsilon)\lambda)B^2}$$
$$+ \sum_{j\in[n],k\in[n+1:2n]} e^{((1-\varepsilon)\lambda+1)B^2} + \sum_{j,k\in[n+1:2n]} e^{3B^2/2}$$
$$\geq n^2 e^{3(1-\varepsilon)\lambda B^2} + 2n^2 e^{(1+(1-\varepsilon)\lambda)B^2} + n^2 e^{3B^2/2} \geq n^2 e^{3B^2/2}$$

We also have $\lambda < 1/2$ since $M < c$. Now, an upper bound of $[XYZ]_{i,i}$ can also be computed using $\langle a_i,b_j\rangle, \langle b_j,c_k\rangle, \langle c_k,a_i\rangle \leq M\log n$ and Equation 6 as,

$$[XYZ]_{i,i} \leq \sum_{j,k\in[n]} e^{3\lambda B^2} + \sum_{j\in[n+1:2n],k\in[n]} e^{(1+\lambda)B^2} + \sum_{j\in[n],k\in[n+1:2n]} e^{(1+\lambda)B^2} + \sum_{j,k\in[n+1:2n]} e^{3B^2/2},$$
$$\leq n^2 e^{3\lambda B^2} + 2n^2 e^{(1+\lambda)B^2} + n^2 e^{3B^2/2} \leq 2n^2 e^{3B^2/2},$$

for large enough $B$ when $\lambda$ is constant.

**Bounds on the numerator.** We analyze bounds on $[XD^{1,1}YD^{2,1}Z]_{i,i}$ when a positive certificate of $\mathsf{IP}\Delta$ contains $a_i$ versus when it does not.

**Case 1: $\mathsf{IP}\Delta$ present at $i$.** In this case, we have

$$[XD^{1,1}YD^{2,1}Z]_{i,i} = \sum_{j,k\in[\tilde{n}]} e^{(\langle Q_i^{(1)},Q_j^{(2)}\rangle+\langle Q_j^{(2)},Q_k^{(3)}\rangle+\langle Q_k^{(3)},Q_i^{(1)}\rangle)/\tilde{d}} V_{j,1}^{(1)} V_{k,2}^{(1)}$$
$$= \sum_{j,k\in[n]} e^{(\langle Q_i^{(1)},Q_j^{(2)}\rangle+\langle Q_j^{(2)},Q_k^{(3)}\rangle+\langle Q_k^{(3)},Q_i^{(1)}\rangle)/\tilde{d}} \quad \text{(Using values of } V^{(1)}, V^{(2)}\text{)},$$
$$= \sum_{j,k\in[n]} e^{(\langle a_i,b_j\rangle+\langle b_j,c_k\rangle+\langle c_k,a_i\rangle)B^2/\tilde{d}} > e^{3\lambda B^2} + (n^2-1)e^{3(1-\varepsilon)\lambda B^2} > e^{3\lambda B^2},$$

since we have some $j,k \in [n]$ such that $\langle a_i,b_j\rangle = \langle b_j,c_k\rangle = \langle c_k,a_i\rangle = m$.

Therefore,

$$\frac{[XD^{1,1}YD^{2,1}Z]_{i,i}}{[XYZ]_{i,i}} > \frac{e^{3\lambda B^2}}{2n^2 e^{3B^2/2}}. \tag{7}$$

**Case 2: $\mathsf{IP}\Delta$ not present in $i$.** Here, we have all $\langle a_i,b_j\rangle + \langle b_j,c_k\rangle + \langle c_k,a_i\rangle \leq (2M+(1-\varepsilon)M)\log n$ for all $j,k$ since otherwise it will contain a $\mathsf{IP}\Delta$. Therefore,

$$[XD^{1,1}YD^{2,1}Z]_{i,i} = \sum_{j,k\in[\tilde{n}]} e^{(\langle Q_i^{(1)},Q_j^{(2)}\rangle+\langle Q_j^{(2)},Q_k^{(3)}\rangle+\langle Q_k^{(3)},Q_i^{(1)}\rangle)/\tilde{d}} V_{j,1}^{(1)} V_{k,2}^{(2)}$$
$$= \sum_{j,k\in[n]} e^{(\langle Q_i^{(1)},Q_j^{(2)}\rangle+\langle Q_j^{(2)},Q_k^{(3)}\rangle+\langle Q_k^{(3)},Q_i^{(1)}\rangle)/\tilde{d}},$$
$$= \sum_{j,k\in[n]} e^{(\langle a_i,b_j\rangle+\langle b_j,c_k\rangle+\langle c_k,a_i\rangle)B^2/\tilde{d}} \leq \sum_{j,k\in[n]} e^{(3-\varepsilon)\lambda B^2} \leq n^2 e^{(3-\varepsilon)\lambda B^2}.$$

which implies

$$\frac{[XD^{1,1}YD^{2,1}Z]_{i,i}}{[XYZ]_{i,i}} < \frac{e^{(3-\varepsilon)\lambda B^2}}{e^{3B^2/2}}. \tag{8}$$

**Wrapping up.** Let $u_i$ be the value of the approximation of the $i$-th entry of the first row of the Strassen-attention matrix, i.e.,

$$\left| u_i - \frac{[XD^{1,1}YD^{2,1}Z]_{i,i}}{[XYZ]_{i,i}} \right| \leq \gamma.$$

We will show that $u_i$ is a distinguisher between the yes and no instances of $\mathsf{IP\Delta}$; in particular for appropriate settings of the parameters we will see that the value of $u_i$ in Case 1 (the yes case) is always greater than the value of $u_i$ in Case 2 (the no case).

In Case 1, using Equations 7, we have

$$u_i > \frac{[XD^{1,1}YD^{2,1}Z]_{i,i}}{[XYZ]_{i,i}} - \gamma > \frac{e^{3\lambda B^2}}{2n^2 e^{3B^2/2}} - \gamma,$$

and in Case 2, using Equation 8, we have

$$u_i < \frac{[XD^{1,1}YD^{2,1}Z]_{i,i}}{[XYZ]_{i,i}} + \gamma < \frac{e^{(3-\varepsilon)\lambda B^2}}{e^{3B^2/2}} - \gamma.$$

Thus it suffices to verify the following inequality:

$$\frac{e^{(3-\varepsilon)\lambda B^2}}{e^{3B^2/2}} + \gamma < \frac{e^{3\lambda B^2}}{2n^2 e^{3B^2/2}} - \gamma,$$

which is indeed satisfied for $\gamma < \frac{1}{n^{2+\Omega(1)}}$ and $e^{\varepsilon\lambda B^2} > n^2$. Therefore, $B^2 = \Omega(\log n)$ suffices.

Therefore, we have reduced $\mathsf{Gap\text{-}IP\Delta}$ to $\mathsf{APAC}^{(h_S)}$ where $\Gamma = B = \Omega(\sqrt{\log n})$, completing the proof of the lemma. $\qquad\square$

Therefore, if $\mathsf{APAC}^{(h_S)}$ could be solved in $O(n^{\omega-\delta})$ time, then that would imply that $\mathsf{IP\Delta}$ could be solved in $O(n^{\omega-\Omega(\delta)})$ time (Theorem C.8), which in turn would imply $\mathsf{Max\text{-}2SAT}$ could be solved in $2^{(\omega/3-\Omega(\delta))n}$ time (Lemma C.7), which can not be true for an absolute constant $\delta > 0$ (Hypothesis 3).

## D PROOFS OF SECTION 3.2: TREE-ATTENTION

In this section, we prove the first part of Theorem 3.5 by giving an algorithm to exactly compute the output of tree-attention. The second and third parts are computational complexities of special subcases of poly-attention, which has been proved in Section E.

Before giving an algorithm for the exact computation complexity of tree-attention, we show a property of branchings in the graphical representation. This happens when the underlying polynomial for the poly-attention is a variable separable polynomial.

**Lemma D.1** (Variable separability). *If $h(x_1, \ldots, x_t) = f(x_1, \ldots, x_i) + g(x_1, x_{i+1}, \ldots, x_t)$ for some $i \in [t-1]$ and some polynomials $f, g$ of minimum possible degrees, i.e., $h$ is variable separable (Definition A.5), then we have $Att^{(h)} = Att^{(f)} \odot Att^{(g)}$ and also the entrywise-approximation $\widehat{Att^{(h)}} = \widehat{Att^{(f)}} \odot \widehat{Att^{(g)}}$. If the (entrywise-approximations of) outputs of poly-attention, $Att^{(f)}$ and $Att^{(g)}$, can be computed in time $T^f(n)$ and $T^g(n)$ respectively, then computing the (entrywise-approximation of) output of poly-attention for $h$, $Att^{(h)}$, can be performed in time $O(\max\{T^f(n), T^g(n)\} + nd)$.*

*Proof.* For all $j \in [n], k \in [d]$, we have,

$$
\begin{aligned}
&Att_{j,k}^{(f)} \cdot Att_{j,k}^{(g)} \\
&= \frac{\sum_{\ell_2,\ldots,\ell_i} \exp(\frac{1}{d} f(Q_j^{(1)}, Q_{\ell_2}^{(2)}, \ldots, Q_{\ell_i}^{(i)})) V_{\ell_2,k}^{(2)} \cdots V_{\ell_i,k}^{(i)}}{\sum_{\ell_2,\ldots,\ell_i} \exp(\frac{1}{d} f(Q_j^{(1)}, Q_{\ell_2}^{(2)}, \ldots, Q_{\ell_i}^{(i)}))} \\
&\qquad \times \frac{\sum_{\ell_{i+1},\ldots,\ell_t} \exp(\frac{1}{d} g(Q_j^{(1)}, Q_{\ell_{i+1}}^{(i+1)}, \ldots, Q_{\ell_t}^{(t)})) V_{\ell_{i+1},k}^{(i+1)} \cdots V_{\ell_t,k}^{(t)}}{\sum_{\ell_{i+1},\ldots,\ell_t} \exp(\frac{1}{d} g(Q_j^{(1)}, Q_{\ell_{i+1}}^{(i+1)}, \ldots, Q_{\ell_t}^{(t)}))} \qquad (9) \\
&= \frac{\sum_{\ell_2,\ldots,\ell_t} \exp\left(\frac{1}{d}(f(Q_j^{(1)}, Q_{\ell_2}^{(2)}, \ldots, Q_{\ell_i}^{(i)}) + g(Q_j^{(1)}, Q_{\ell_{i+1}}^{(i+1)}, \ldots, Q_{\ell_t}^{(t)}))\right) V_{\ell_2,k}^{(2)} \cdots V_{\ell_t,k}^{(t)}}{\sum_{\ell_2,\ldots,\ell_t} \exp\left(\frac{1}{d}(f(Q_j^{(1)}, Q_{\ell_2}^{(2)}, \ldots, Q_{\ell_i}^{(i)}) + g(Q_j^{(1)}, Q_{\ell_{i+1}}^{(i+1)}, \ldots, Q_{\ell_t}^{(t)}))\right)} \\
&= \frac{\sum_{\ell_2,\ldots,\ell_t} \exp(\frac{1}{d} h(Q_j^{(1)}, Q_{\ell_2}^{(2)}, \ldots, Q_{\ell_t}^{(t)})) V_{\ell_2,k}^{(2)} \cdots V_{\ell_i,k}^{(t)}}{\sum_{\ell_2,\ldots,\ell_t} \exp(\frac{1}{d} h(Q_j^{(1)}, Q_{\ell_2}^{(2)}, \ldots, Q_{\ell_t}^{(t)}))} = Att_{j,k}^{(h)}.
\end{aligned}
$$

This implies $Att^{(f)} \odot Att^{(g)} = Att^{(h)}$, and if we obtain entrywise approximations $\widehat{Att^{(f)}}$ and $\widehat{Att^{(g)}}$ respectively with error $\gamma = \frac{1}{\text{poly}(n)}$, then $\widehat{Att^{(f)}} \odot \widehat{Att^{(g)}}$ will be an entrywise approximation of $Att^{(h)}$ with error $\gamma_0 = O(\gamma) = \frac{1}{\text{poly}(n)}$ as well.

Note that the polynomials might not even contain the variable $x_1$, in which case we all the rows of the output of the corresponding poly-attention matrix will be the same. □

Now, we prove that $Att^{(h)}$, where $h$ is a tree polynomial, can be computed in $O(n^2)$ time.

**Theorem D.2.** *If $h$ is a tree polynomial (graphical representation of $h$ is a tree or a forest), then we can compute $Att^{(h)}$ exactly in $\tilde{O}(n^2)$ time.*

*Proof.* Algorithm 2 gives a procedure for computing the output of tree-attention given query-key and value matrices as inputs. Indeed, if there were multiple forests, we could have computed the output of tree-attention for each of them separately, and composed them together using Lemma D.1.

**Overview.** We start with a tree rooted at $v_1$, and compute poly-attention on each of the subtrees (polynomials corresponding to the subtrees) where the query variable[3] is the root of the subtree. The main idea to compute this is, whenever we have a branching, we compute each of the subtrees separately, and compose them together using Hadamard product of Lemma D.1.

In Algorithm 2, we fix each of the columns $\ell \in [d]$ (Step 2), and compute $Att_{(1:n,\ell)}^{(h)}$, one at a time. The computation of proceeds as computing the numerator and the denominator terms separately, from the graphical representation $G^h$ (as in Equation 1). In this recursive formulation, we employ compute the values in a DFS fashion, first, we fix the root of the tree given by variable $x_1$ (vertex $v_1$ in the graph), having the corresponding query-key matrix $Q^{(1)}$, and proceed to computing the output of the poly-attention mechanism for its subtree polynomial.

**Each branch.** Without loss of generality, consider the root variable $v_1$, and for each branch from $v_1$, consider an edge given by $(v_1, v_{j_i})$, i.e., $v_1 — v_{j_i}$, for $i \in [p]$, where $p$ is the number of branches. When $v_{j_i}$ is a leaf, we compute the poly-attention $Att^{(x_1 x_{j_i})}$, and recursively pass it up the tree. The denominator and numerator of $Att^{(x_1 x_{j_i})}$ are defined in Step 12 of Algorithm 2 – two vectors in $\mathbb{R}^{n \times 1}$ which can be computed in $O(n^2)$ time and then their ratio is the poly-attention output for this branch (Step 13).

Next, when $v_{j_i}$ is not a leaf, i.e., the tree proceeds as $v_1 — v_{j_i} —$, let us assume the polynomial whose subtree rooted at $v_{j_i}$ is given by $g_i(x_{j_i}, \bar{x}^i)$ and that we have already computed $Att^{(g_i(x_{j_i}, \bar{x}^i))}$

---

[3]*Query variable* refers to the variable of the highest priority in the polynomial (priority of monomials and variables has been defined in Definition E.1). It is usually the variable $x_1$, and the indices of the corresponding query-key matrix in the softmax computation correspond to the rows of $Att^{(h)}$ (see Equation 1).

---

**Algorithm 2** Algorithm to compute tree attention $Att^{(h)}$

---

**Input:** A polynomial $h(x_1, \ldots, x_t)$ whose graphical representation is a tree, query-key matrices
  $Q^{(1)}, \ldots, Q^{(t)} \in \mathbb{R}^{n \times d}$ and value matrices $V^{(2)}, \ldots, V^{(t)} \in \mathbb{R}^{n \times d}$.
**Output:** $Att^{(h)} \in \mathbb{R}^{n \times d}$

1: Construct $G$ as the graphical representation of $h$, with vertices $v_1, \ldots, v_t$.
2: **for** $\ell \in [d]$ **do**
3:     Let $p$ be the number of children of $v_1$.
4:     **for** all child node $v_{j_i}$ of $v_1$, $i \in [p]$ **do**
5:         **if** $v_{j_i}$ is not a leaf **then**
6:             Let $g_i(x_{j_i}, \bar{x^i})$ be the polynomial of the subtree rooted at $v_{j_i}$.
7:             Compute $Att_{(1:n,\ell)}^{(g_i(x_{j_i}, \bar{x^i}))}$ recursively, where $v_{j_i}$ is the query variable, by computing
              the numerator term and the denominator term separately. Let the numerator term be
              $P^{(g_i(x_{j_i}, \bar{x^i}))} \in \mathbb{R}^{n \times 1}$ and the denominator term be $R^{(g_i(x_{j_i}, \bar{x^i}))} \in \mathbb{R}^{n \times 1}$.
8:             Define the numerator

$$P^{(x_1 x_{j_i} + g_i(x_{j_i}, \bar{x^i}))} := [Q^{(1)}(Q^{(j_i)})^T]^e D^{V^{(j_i)}} P^{(g_i(x_{j_i}, \bar{x^i}))},$$

              and the denominator,

$$R^{(x_1 x_{j_i} + g_i(x_{j_i}, \bar{x^i}))} := [Q^{(1)}(Q^{(j_i)})^T]^e R^{(g_i(x_{j_i}, \bar{x^i}))},$$

              where $D^{V^{(j_i)}} = diag(V_{(1:n,\ell)}^{(j_i)}) \in \mathbb{R}^{n \times n}$.
9:             Compute

$$Att_{(1:n,\ell)}^{(x_1 x_{j_i} + g_i(x_{j_i}, \bar{x^i}))} := \frac{P^{(x_1 x_{j_i} + g_i(x_{j_i}, \bar{x^i}))}}{R^{(x_1 x_{j_i} + g_i(x_{j_i}, \bar{x^i}))}}.$$

10:         **else**
11:             Here, $g_i(x_{j_i}, \bar{x^i}) = 0$ since there is no tree rooted at $v_{j_i}$.
12:             Define the numerator

$$P^{(x_1 x_{j_i})} := [Q^{(1)}(Q^{(j_i)})^T]^e V_{(1:n,\ell)}^{(j_i)},$$

              and the denominator,

$$R^{(x_1 x_{j_i})} := [Q^{(1)}(Q^{(j_i)})^T]^e \mathbf{1}_{n \times 1}.$$

13:             Compute

$$Att_{(1:n,\ell)}^{(x_1 x_{j_i})} := \frac{P^{(x_1 x_{j_i})}}{R^{(x_1 x_{j_i})}}.$$

14:         **end if**
15:     **end for**
16:     For composing the branches together, compute the final numerator

$$P^{(h)} := P^{(x_j x_{j_1} + g_1(x_{j_1}, \bar{x^1}))} \odot \ldots \odot P^{(x_j x_{j_p} + g_p(x_{j_p}, \bar{x^p}))},$$

        and the final denominator,

$$R^{(h)} := R^{(x_j x_{j_1} + g_1(x_{j_1}, \bar{x^1}))} \odot \ldots \odot R^{(x_j x_{j_p} + g_p(x_{j_p}, \bar{x^p}))},$$

        where $h = x_j x_{j_1} + g_1(x_{j_1}, \bar{x^1}) + \ldots + x_j x_{j_p} + g_p(x_{j_p}, \bar{x^p})$ (by definition).
17:     Define

$$Att_{(1:n,\ell)}^{(h)} := \frac{P^{(h)}}{R^{(h)}}.$$

18: **end for**
19: **return** $Att^{(h)}$.

---

(the numerator and the denominator are separately given to us as $P^{(g_i(x_{j_i}, \bar{x^i}))}$, $R^{(g_i(x_{j_i}, \bar{x^i}))} \in \mathbb{R}^{n \times 1}$ respectively). By $\bar{x^i}$, we simply denote the subset of variables other than $x_{j_i}$ that the subtree consists of. The output of tree-attention of the subtree rooted at $v_1$ is essentially $Att^{(x_1 x_{j_i} + g_i(x_{j_i}, \bar{x^i}))}$. For this, the numerator and the denominator can be computed as in Step 8 – both of these computations take $O(n^2)$ time. The final value of $Att^{(x_1 x_{j_i} + g_i(x_{j_i}, \bar{x^i}))}_{(1:n, \ell)}$ is given by Step 9, and we pass the numerator and denominator vectors up the tree recursively.

**Along a branching.** For conglomerating the branches, let us say that the children nodes of $v_1$ are $v_{j_1}, \ldots, v_{j_p}$, where the polynomials corresponding to their subtrees are $g_1(x_{j_1}, \bar{x^1}), \ldots, g_p(x_{j_p}, \bar{x^p})$ ($\bar{x^1}, \ldots, \bar{x^p}$ are disjoint subsets of variables which are precisely the ones present in each of the $p$ subtrees, respectively). We also assume that we have recursively computed the $\ell$-th columns of the poly-attention outputs $Att^{(g_1(x_{j_1}, \bar{x^1}))}_{(1:n, \ell)}, \ldots, Att^{(g_p(x_{j_p}, \bar{x^p}))}_{(1:n, \ell)}$, in terms of the numerators $P^{(g_1(x_{j_1}, \bar{x^1}))}, \ldots, P^{(g_p(x_{j_p}, \bar{x^p}))}$, and denominators $R^{(g_1(x_{j_1}, \bar{x^1}))}, \ldots, R^{(g_p(x_{j_p}, \bar{x^p}))}$ respectively. Now, the poly-attention output for the polynomial having the subtree rooted at $v_1$, which is

$$h(x_1, \bar{x^1}, \ldots, \bar{x^p}) := x_1 x_{j_1} + g_{j_1}(x_{j_1}, \bar{x^1}) + \ldots + x_1 x_{j_p} + g_{j_p}(x_{j_p}, \bar{x^p}),$$

is computed in Steps 16-17, and the correctness of this computation follows from Lemma D.1.

**Time complexity.** We show a quadratic time-complexity for Algorithm 2. Let us assume that recursively in a branch, the numerator and the denominator of $Att^{(g_i(x_{j_i}, \bar{x}))}$ can be computed in $\tilde{O}(n^2)$ time (Step 7). From this, extending the output matrix of poly-attention to the current vertex (Steps 8-9, 12-13, followed by 16-17) each require $\tilde{O}(n^2)$ time. The number of these sub-tree attention computations required is at most the size of the tree, $O(s)$, which is a constant. Therefore, this gives a DFS-style procedure to compute the $Att^{(h)}_{(1:n, \ell)}$ in time $\tilde{O}(n^2)$ since the graph is of constant size, and repeating for all $\ell \in [d]$, we will be able to find the entire matrix $Att^{(h)}$. $\qquad \square$

## E PROOFS OF SECTION 3.3: COMPUTATIONAL COMPLEXITIES OF POLY-ATTENTION

Throughout this paper, we will compute the numerator and the denominator in Equation 1 separately, where the *numerator term* is $\sum_{\ell_2, \ldots, \ell_t \in [n]} \exp\left(\frac{1}{d} h(Q^{(1)}_{\ell_1}, \ldots, Q^{(k)}_{\ell_k})\right) V^{(2)}_{\ell_2} \odot V^{(3)}_{\ell_3} \odot \ldots \odot V^{(t)}_{\ell_t}$, and the *denominator term* is $\sum_{\ell_2, \ldots, \ell_t \in [n]} \exp\left(\frac{1}{d} h(Q^{(1)}_{\ell_1}, \ldots, Q^{(k)}_{\ell_k})\right)$.

We also define a *monomial ordering*, which will help us proceed with the proofs of these theorems.

**Definition E.1** (Monomial ordering). *A monomial $m_1$ is said to be* higher preference *than another monomial $m_2$ if either of the following holds:*

- $\deg(m_1) > \deg(m_2)$, *or*

- $\deg(m_1) = \deg(m_2)$ *and $m_1$ comes lexicographically before $m_2$, i.e., if $i$ is the smallest index such that $x_i$ is present in exactly one of the monomials, then the monomial in which $x_i$ is present has higher preference.*

We will order the monomials of $h$ according to this order, and $m_i$ will denote the $i$-th monomial. Note that this definition can also be used with variables, where a variable $x_i$ has a higher preference than $x_j$ if and only if $i < j$.

The polynomial method Alman & Song (2023; 2024; 2025) can again be applied to poly-attention, by reducing $\mathsf{APAC}^{(h)}$ to the computation the output of a larger $t$-tensor attention, where the query-key vectors in tensor attention are of dimension $n \times (sd)$. However, the bound on the variables in this case of computing poly-attention will be $o((\log n)^{1/k})$ in contrast to that of tensor attention being $o((\log n)^{1/t})$ Alman & Song (2024).

For proving Theorem 3.6, we show the two parts, upper and lower bounds, separately. For upper bounds, we give a polynomial method algorithm if the entries of the query-key matrices are bounded

(Theorem E.2), and if the entries are large, we give hardness results for entry-wise approximation conditioned on fine-grained complexity conjectures (Theorem E.3).

**Theorem E.2** (Polynomial method on poly-attention). *Given an attention polynomial $h(x_1, \ldots, x_t)$ of degree $k$ having $s$ monomials, where $t, k, s$ are constants, there is an algorithm that solves $\mathsf{APAC}^{(h)}(n, d = O(\log n), \Gamma = o((\log n)^{1/k}), \gamma = 1/\mathrm{poly}(n))$ with query-key matrices $Q^{(1)}, \ldots, Q^{(t)} \in [-\Gamma, \Gamma]^{n \times d}$, and value matrices $V^{(2)}, \ldots, V^{(t)} \in \mathbb{R}^{n \times d}$ in time $O(n^{1+o(1)})$.*

**Theorem E.3** (Lower bound for poly-attention). *Given an attention polynomial $h(x_1, \ldots, x_t)$ of degree $k$ having $s$ monomials, where $t, k, s$ are constants, we are interested in computing an entry-wise $\gamma$-approximation $Att^{(h)}$ having query-key matrices $Q^{(1)}, \ldots, Q^{(t)} \in [-\Gamma, \Gamma]^{n \times d}$, and value matrices $V^{(2)}, \ldots, V^{(t)} \in \mathbb{R}^{n \times d}$, for $d = O(\log n), \gamma = 1/\mathrm{poly}(n)$. Then, depending on the structure of $h$,*

1. *If $k \geq 2$, then assuming $\mathsf{SETH}$ (Hypothesis 1), an entry-wise approximation of $Att^{(h)}$ can not be computed in time $O(n^{k-\Omega(1)})$ when $\Gamma = \Omega((\log n)^{1/k})$.*

2. *If $h$ contains an elementary symmetric polynomial $\binom{[t_0]}{k}$ for some $t_0 \leq t$, then assuming the $\mathsf{Max}\text{-}k\mathsf{SAT}$ conjecture (Hypothesis 2), an entry-wise approximation of $Att^{(h)}$ can not be computed in time $O(n^{k_0-\Omega(1)})$ when $\Gamma = \Omega((\log n)^{1/k})$.*

3. *If $k = 2$ and $h$ is not a tree polynomial, then assuming the $\mathsf{Max}\text{-}2\mathsf{SAT}$ conjecture (Hypothesis 3), an entry-wise approximation of $Att^{(h)}$ can not be computed in time $O(n^{\omega-\Omega(1)})$ when $\Gamma = \Omega((\log n)^{1/2})$.*

### E.1 POLYNOMIAL METHOD FOR POLY-ATTENTION

In this section, we prove Theorem E.2. We start with the polynomial $h$ as defined in Theorem E.2, and reduce the problem of computing an entry-wise approximation of $Att^{(h)} \in \mathbb{R}^{n \times d}$ to that of $Att^{(T)} \in \mathbf{R}^{n \times (sd)}$, by constructing query-key matrices $K^{(1)}, \ldots, K^{(t)} \in \mathbb{R}^{n \times (sd)}$ and value matrices $W^{(1)}, \ldots, W^{(t)} \in \mathbb{R}^{n \times (sd)}$, such that the row-softmax matrix of

$$\frac{1}{d} K^{(1)} \left( K^{(2)} \oslash K^{(3)} \oslash \ldots \oslash K^{(t)} \right)^T,$$

is same as the softmax matrix of $Att^{(h)}$, and $Att^{(h)}$ is exactly equal to $Att^{(T)}_{(1:n, 1:d)}$ using these inputs, and the remaining entries of $Att^{(T)}$ are zeros.

**Defining $K^{(j)}$.** We define $K^{(j)} \in \mathbb{R}^{n \times (sd)}$, for all $j \in [t]$, by dividing the columns into $s$ blocks, each having $d$ columns. These blocks are defined as, for $j \in [t]$:

- the $i$-th block, for $i \in [s]$, contains the matrix $Q^{(j)}$ if the $i$-th monomial of $h$ contains the variable $x_j$,
- otherwise, the $i$-th block, for $i \in [s]$, contains the all ones matrix $\mathbf{1}_{n \times d}$.

Roughly, the query-key matrices can be seen as:

$$K^{(j)} = \begin{bmatrix} \overbrace{1 \ \ldots \ 1}^{d} & \overbrace{Q^{(j)}_{1,1} \ \cdots \ Q^{(j)}_{1,d}}^{d} & \overbrace{1 \ \ldots \ 1}^{d} & \\ \vdots \quad \vdots & \vdots \qquad \vdots & \vdots \quad \vdots & \cdots \\ \underbrace{1 \ \ldots \ 1}_{x_j \text{ not in } m_1} & \underbrace{Q^{(j)}_{n,1} \ \cdots \ Q^{(j)}_{n,d}}_{x_j \text{ is in } m_2} & \underbrace{1 \ \ldots \ 1}_{x_j \text{ not in } m_3} & \end{bmatrix}_{n \times (sd)}.$$

Using these definitions, it can be verified that for this choice of $K^{(j)}$'s, we have

$$\langle K^{(1)}_{\ell_1}, K^{(2)}_{\ell_2}, \ldots, K^{(t)}_{\ell_t} \rangle = \sum_{i \in [s]} \langle K^{(1)}_{\ell_1, (i-1)d+1:id}, K^{(2)}_{\ell_2, (i-1)d+1:id}, \ldots, K^{(t)}_{\ell_t, (i-1)d+1:id} \rangle \quad (10)$$

$$= \sum_{i \in [s]} \langle Q^{(j_1)}_{\ell_{j_1}}, \ldots, Q^{(j_{k_i})}_{\ell_{j_{k_i}}} \rangle, \quad (11)$$

where the monomials of $h$ are defined as before (Definition 2.2).

**Defining $W^{(j)}$.** The value matrices for the $t$-tensor attention operation will be the same as that of poly-attention. In order to match the embedding dimensions of the query-key matrices and the value matrices of the $t$-tensor attention operation (as was used in Alman & Song (2024)), we can simply consider the new $n \times (sd)$ dimensional value matrices, $W^{(j)}$'s to contain the corresponding $n \times d$ dimensional value matrices $V^{(t)}$ in the first $d$-columns, and all the remaining entries of $W^{(j)}$ contain zero. More specifically,

$$W^{(j)} = \begin{bmatrix} V^{(j)} & \mathbf{0}_{n \times d} & \cdots & \mathbf{0}_{n \times d} \end{bmatrix}_{n \times (sd)}. \tag{12}$$

Now, in Equation 1, note that the poly-attention output can be written as

$$D^{-1} A W^{(2)} \oslash \ldots \oslash W^{(t)},$$

where $A \in \mathbb{R}^{n \times n^{t-1}}$ is defined as

$$A = [\frac{1}{d} K^{(1)} (K^{(2)} \oslash \ldots \oslash K^{(t)})^T]^e,$$

and $D$ is the $n \times n$ diagonal matrix

$$D = diag \left( [\frac{1}{d} K^{(1)} (K^{(2)} \oslash \ldots \oslash K^{(t)})^T]^e \underbrace{\mathbf{1}_{n \times 1} \oslash \ldots \oslash \mathbf{1}_{n \times 1}}_{(t-1) \text{ times}} \right).$$

This is precisely the form of a $t$-tensor attention mechanism. Next, in order to use the polynomial method on this matrix, we need the entries to be bounded.

**Lemma E.4** (Bounded entries). *Given $Q^{(j)} \in [-\Gamma, \Gamma]^{n \times d}$ and $h$ defined as above, we have*

$$e^{-s\Gamma^k} \leq \exp\left(\frac{1}{d} h(Q_{\ell_1}^{(1)}, \ldots, Q_{\ell_t}^{(t)})\right) \leq e^{s\Gamma^k},$$

*for all $\ell_1, \ldots, \ell_t \in [n]$. For $\Gamma = o(\frac{1}{s}(\log n)^{1/k}) = o((\log n)^{1/k})$, the entries $\exp\left(\frac{1}{d} h(Q_{\ell_1}^{(1)}, \ldots, Q_{\ell_t}^{(t)})\right)$ are sub-polynomial in $n$.*

*Proof.* Since $h$ is a degree $k$ polynomial with constant coefficients, for each monomial $m_i$ of $h$, $\frac{1}{d} m_i(Q_{\ell_1}^{(1)}, \ldots, Q_{\ell_t}^{(t)})$ in in the range of $[-\Gamma^k, \Gamma^k]$. There are $s$ monomials and the total value is bounded inside the interval $[-s\Gamma^k, s\Gamma^k]$, which gives the required result after exponentiation. $\qquad \square$

For completing the algorithm, we use results which follow from the proofs in (Alman & Song, 2024, Apx. E).

**Theorem E.5** (Alman & Song (2024)). *Given matrices $K^{(1)}, \ldots, K^{(t)} \in [-\Gamma, \Gamma]^{n \times d}$ and value matrices $W^{(2)}, \ldots, W^{(t)} \in \mathbb{R}^{n \times d}$, we can compute an entry-wise $\gamma$-approximation, for $\gamma = 1/\mathrm{poly}(n)$, of the following:*

1. *A matrix $\widehat{Att} \in \mathbb{R}^{n \times d}$ which is the entry-wise $\gamma$-approximation of the numerator matrix of tensor attention output*

$$Att = [\frac{1}{d} K^{(1)} (K^{(2)} \oslash \ldots \oslash K^{(t)})^T]^e W^{(2)} \oslash \ldots \oslash W^{(t)},$$

*that is, for all $i \in [n], j \in [d]$,*

$$|\widehat{Att}_{i,j} - Att_{i,j}| < \gamma.$$

2. *A diagonal matrix $\hat{D} \in \mathbb{R}^{n \times n}$ which is an entry-wise approximation of the diagonal matrix $D \in \mathbb{R}^{n \times n}$ given by*

$$D = diag\left([\frac{1}{d}K^{(1)}(K^{(2)} \oslash \ldots \oslash K^{(t)})^T]^e \mathbf{1}_{n \times 1} \oslash \ldots \oslash \mathbf{1}_{n \times 1}\right),$$

*that is, for all $i \in [n]$,*

$$|\hat{D}_{i,i} - D_{i,i}| < \gamma.$$

*Here, when the condition* $\max\left\{\frac{\log(1/\gamma)}{\log(\log(1/\gamma)/\Lambda)}, \Lambda\right\} = o(\log n)$ *is met (where $\Lambda = ||\frac{1}{d}K^{(1)}(K^{(2)} \oslash \ldots \oslash K^{(t)})^T||_\infty$), the time complexity for finding the matrices $\widehat{Att}$, $\hat{D}$, and hence an entry-wise $2\gamma$-approximation of $D^{-1}Att$, is $n^{1+o(1)}$.*

Using Lemma E.4, the value of $\Lambda$ in Theorem E.5 is $O(\Gamma^k)$, and for the choice of $\Gamma = o((\log n)^{\frac{1}{k}})$, the quantity $\max\left\{\frac{\log(1/\gamma)}{\log(\log(1/\gamma)/\Lambda)}, \Lambda\right\}$ is indeed $o(\log n)$, which gives our required almost-linear complexity for computing $Att^{(h)}$.

Summing up, the algorithm for computing entry-wise approximation of $Att^{(h)}$ is given as the following algorithm.

---

**Algorithm 3** Algorithm to compute an entry-wise approximation of $Att^{(h)}$

---

**Input:** An attention polynomial $h(x_1, \ldots, x_t)$ of degree $k$, matrices $Q^{(1)}, \ldots, Q^{(t)}, V^{(2)}, \ldots, V^{(t)} \in \mathbb{R}^{n \times d}, \gamma = \frac{1}{\text{poly}(n)}$

**Output:** Entry-wise $\gamma$-approximation $\widehat{Att^{(h)}} \in \mathbb{R}^{n \times d}$ of $Att^{(h)} \in \mathbb{R}^{n \times d}$.
1: Using $Q^{(1)}, \ldots, Q^{(t)}$ and $h$, compute $K^{(1)}, \ldots, K^{(t)} \in \mathbb{R}^{n \times (sd)}$ (Equation 10). $\quad \triangleright O(nd)$ time.
2: Compute $W^{(2)}, \ldots, W^{(t)} \in \mathbb{R}^{n \times (sd)}$ from $V^{(2)}, \ldots, V^{(t)}$ (Equation 12). $\quad \triangleright O(nd)$ time.
3: Compute entry-wise $\gamma$-approximation $\widehat{Att} \in \mathbb{R}^{n \times (sd)}$ of

$$Att = [\frac{1}{d}K^{(1)}(K^{(2)} \oslash \ldots \oslash K^{(t)})^T]^e W^{(2)} \oslash \ldots \oslash W^{(t)},$$

using Theorem E.2, Step 1. $\quad\quad\quad\quad\quad\quad\quad\quad\quad\quad\quad\quad \triangleright O(n^{1+o(1)}d)$ time.
4: Compute entry-wise $\gamma$-approximation $\hat{D} \in \mathbb{R}^{n \times n}$ of

$$D = diag\left([\frac{1}{d}K^{(1)}(K^{(2)} \oslash \ldots \oslash K^{(t)})^T]^e \mathbf{1}_{n \times 1} \oslash \ldots \oslash \mathbf{1}_{n \times 1}\right),$$

which is a diagonal matrix, using Theorem E.2, Step 2. $\quad\quad\quad\quad \triangleright O(n^{1+o(1)}d)$ time.
5: **Return** $\hat{D}^{-1}\widehat{Att}_{(1:n,1:d)}$. $\quad\quad\quad\quad\quad\quad\quad\quad\quad\quad\quad\quad\quad \triangleright O(nd)$ time.

---

This proves Theorem E.2.

## E.2 TIME LOWER BOUNDS FOR POLY-ATTENTION

We complete the main complexity result of this paper, either we can compute an entry-wise approximation of poly-attention in near-linear time, when the entries of the query-key matrices are bounded; or we require at least superquadratic time, unless the polynomial for poly-attention is a tree polynomial.

Our proofs for showing the hardness of entry-wise approximation of $Att^{(h)}$ consists of two reductions: (1) first we reduce from each of $k$IP, HypergraphIP, and IP$\Delta$ (which have popularly known hardness conjectures of SETH, Max-2SAT, Max-$k$SAT respectively) to $n^{o(1)}$ instances of their respective gap versions, and (2) secondly, we reduce each of those gap versions to an entry-wise approximation of poly-attention. These subcases and the starting complexity assumptions will be based on the structure of $h$ provided, as categorized in Theorem E.3.

For proving Step 1, when we prove the first case, we get hard instances of $\varepsilon$-Gap-$k$IP assuming SETH (Theorem E.7). For the second case, we assume Max-$k$SAT is true, reduce Max-$k$SAT using a known reduction (Lemma E.10) to $n^{o(1)}$ instances of HypergraphIP, and further reduce each of those instances to $n^{o(1)}$ instances of $\varepsilon$-Gap-HypergraphIP (Corollary E.12). For the third case, we start with Max-2SAT and reduce that to $n^{o(1)}$ instances of IP$\Delta$ (Lemma C.7), and then to $n^{o(1)}$ instances of $\varepsilon$-Gap-IP$\Delta$ (Theorem C.8).

We complete the reductions for Step 2 in each of the following subsections.

### E.2.1 Time lower bounds based on degree of polynomial using SETH

In this section, we prove the first part of Theorem E.3. We first start with an instance of $k$IP, which is SETH-hard, reduce it to $\varepsilon$-Gap-$k$IP (Definition E.6) using some previous works Rubinstein (2018); Alman & Song (2024), and then using the instances of $\varepsilon$-Gap-$k$IP, create query-key matrices for $Att^{(h)}$ such that an entry-wise $\gamma$-approximation of $Att^{(h)}$ would solve the instance of $\varepsilon$-Gap-$k$IP.

**Definition E.6** ($\varepsilon$-Gap-$k$IP). *For every $\varepsilon \in (0,1)$ and positive integers $k \geq 2$, given sets of vectors $A^1, \ldots, A^k \subseteq \{0,1\}^d$ with $|A^1| = \ldots = |A^k| = n$, a target inner product $m \in \{0, \ldots, d\}$, and the promise that for any $a_1 \in A^1, \ldots, a_k \in A^k$,*

- *either $\langle a_1, \ldots, a_k \rangle = m$,*

- *or, $\langle a_1, \ldots, a_k \rangle \leq (1-\varepsilon)m$,*

*the problem of $\varepsilon$-Gap-$k$IP$_{n,d}$ is to decide if there exist vectors $a_1 \in A^1, \ldots, a_k \in A^k$ such that $\langle a_1, \ldots, a_k \rangle = m$.*

Using Rubinstein (2018)-like techniques, conditional hardness of $\varepsilon$-Gap-$k$IP can be obtained.

**Theorem E.7** (Alman & Song (2024); Rubinstein (2018)). *For every $\delta > 0$ and every constant $\varepsilon \in (0,1)$, there exists a constant $c > 0$, such that $\varepsilon$-Gap-$k$IP$_{n,c\log n}$ for any target inner product $m \in \{0, \ldots, c\log n\}$, cannot be solved in time $O(n^{(1-\delta)k})$, unless SETH is false.*

Due to this result, we start with an instance of $\varepsilon$-Gap-$k$IP and reduce that to an entry-wise approximation of $Att^{(h)}$. If the entry-wise approximation of $Att^{(h)}$ can be computed in $n^{(1-\delta)k}$ time for a constant $\delta > 0$, then $\varepsilon$-Gap-$k$IP can be solved in $\tilde{O}(n^{(1-\delta)k})$ time, which would refute SETH.

**Lemma E.8** ($\varepsilon$-Gap-$k$IP to APAC$^{(h)}$). *For every constant $\varepsilon > 0$, every $\delta \in (0, 0.01)$, every $c, M > 0$, given an attention polynomial $h(x_1, \ldots, x_t)$ of degree $k \geq 2$ having $s$ monomials, where $t, k, s$ are constants, there exist constants $C_a > 0$ and $C_b > 0$ such that if APAC$^{(h)}(2n, (s+1)c\log n, \Gamma = C_b(\log n)^{1/k}, \gamma = n^{-C_a})$ (Definition A.6) with query-key matrices $Q^{(1)}, \ldots, Q^{(t)} \in [-\Gamma, \Gamma]^{2n \times (s+1)c\log n}$ and value matrices $V^{(2)}, \ldots, V^{(t)} \in \mathbb{R}^{2n \times (s+1)c\log n}$ can be solved in time $O(n^{k-\delta})$, then $\varepsilon$-Gap-$k$IP$_{n,c\log n}$ (Definition E.6) with target inner product $m = M\log n$ can also be solved in $O(n^{k-\delta})$ time for any constant $M$.*

*Proof.* Let us start with an instance of $\varepsilon$-Gap-$k$IP$_{n,d=c\log n}$ that we want to solve, with $k$ sets of vectors $A^1, \ldots, A^k \subseteq \{0,1\}^d$, consisting of $n$ vectors each. The vectors are $\{a_1^i, \ldots, a_n^i\} := A^i$ and the target inner product is $m = M\log n$, for a constant $M$, with the promise of the gap condition for an approximation factor $\varepsilon$. We also assume that there does not exist an all one's vector in $A^i$ for each $i \in [k]$, as that would violate the gap-property (as $m$ needs to be smaller than $d$ for hardness).

Using this instance of deciding $\varepsilon$-Gap-$k$IP, we reduce it to computing an entry-wise approximation of $Att^{(h)}$, with query-key matrices $Q^{(1)}, \ldots, Q^{(t)} \in [-\Gamma, \Gamma]^{\tilde{n} \times \tilde{d}}$, and value matrices $V^{(2)}, \ldots, V^{(t)} \in \mathbb{R}^{\tilde{n} \times \tilde{d}}$, for $\tilde{n} = 2n$, $\tilde{d} = (s+1)d = (s+1)c\log n$, and a $\Gamma$ that we will choose later.

Let us assume that the highest preference monomial of $h$, a monomial of degree $k$, is given by $x_{r_1} \ldots x_{r_k}$, where $r_1$ has the index of the highest preference that may or may not be 1.

We will construct the query-key matrices such that each matrix $Q^{(r_j)}$ will contain vectors from $A^j$ for $j \leq k$, zeros otherwise. Having the monomials ordered according to descending order of the monomial ordering (Definition E.1), each of these $Q^{(r_j)}$'s will consist of blocks of columns which

correspond to monomials– the $i$-th column block, containing $d$ columns from $(i-1)d+1$ to $i.d$, for $i \in [s]$, will correspond to the monomial $m_i$, and the last column block will be a normalizing block. The idea of the reduction is that only the degree $k$ term $x_{r_1} \ldots x_{r_k}$ of $h$ will contribute to computing the final inner product, the terms which are subsets of this degree $k$ term will cancel each other out, and all the other terms will be zero, thereby not contributing anything to $h(Q^{(1)}_{\ell_1}, Q^{(2)}_{\ell_2}, \ldots, Q^{(t)}_{\ell_t})$. More specifically, we want,

$$h(Q^{(1)}_{\ell_1}, \ldots, Q^{(t)}_{\ell_t}) = \Lambda \langle a^1_{\ell_{r_1}}, \ldots, a^k_{\ell_{r_k}} \rangle,$$

for $\ell_{r_1}, \ldots, \ell_{r_k} \in [n]$, and some scaling factor $\Lambda$ which we will see later.

**Construction of matrices.** Let us now define each block of $Q^{(j)}$, $j \in [t]$, which will have $2n$ rows and $(s+1)d$ columns. We will define them by defining each of the column-blocks using a scaling factor $B = \omega(1)$. Considering the set $T = \{r_1, \ldots, r_k\}$, we define:

1. For $Q^{(j)}$'s, if $j \notin T$, we just make the entire matrix zero $\mathbf{0}_{2n \times (s+1)d}$.

2. We now fix $j \in [k]$ and define $Q^{(r_j)}$ (i.e., some value of $r_j \in T$). We define first column block of $Q^{(r_j)}$ as:

$$Q^{(r_j)}_{(1:2n,1:d)} = B \begin{bmatrix} a^j_1 \\ a^j_2 \\ \vdots \\ a^j_n \\ \mathbf{0}_d \\ \mathbf{0}_d \\ \vdots \\ \mathbf{0}_d \end{bmatrix}_{2n \times d}.$$

For column blocks $i \in [s]$, if the monomial $m_i$ does not divide $x_{r_1} \ldots x_{r_k}$, we just make that block all zeros

$$Q^{(r_j)}_{(1:2n,(i-1)d+1:i.d)} = \mathbf{0}_{2n \times d}.$$

3. If monomial $i \in [s]$ does indeed divide $x_{r_1} \ldots x_{r_k}$, consider $j_1$ as the index of the highest preference variable present in $m_i = x_{r_{j_1}} \ldots x_{r_{j_{k_i}}}$, for $j_1, \ldots, j_{k_i} \in [k]$, $k_i < k$. Let $s_i$ be the negation of the integer which is the number of occurrences of this monomial $m_i$ along with coefficients, in each of the monomials ordered higher than $i$ and that divides $x_{r_1} \ldots x_{r_k}$ (these are the only non-zero monomials).

   More specifically, $s_i$ is the sum defined by adding:

   - $-1$ from the monomial $m_1$.
   - $-s_\ell$ whenever $1 < \ell < i$, the monomial $m_\ell$ divides $m_1$, the monomial $m_i$ divides $m_\ell$, and the highest preference variable of $m_\ell$ is also present in $m_i$.
   - $-1$ whenever $1 < \ell < i$, the monomial $m_\ell$ divides $m_1$ and $m_i$ divides $m_\ell$, but the highest preference variable of $m_\ell$ is not present in $m_i$.
   - $0$ in all other cases.

   If $x_{r_j}$ is not present in monomial $i$, we simply set

   $$Q^{(r_j)}_{(1:2n,(i-1)d+1:i.d)} := \mathbf{0}_{2n \times d},$$

otherwise:

$$Q^{(r_{j_1})}_{(1:2n,(i-1)d+1:i.d)} = B \begin{bmatrix} s_i a_1^{j_1} \\ s_i a_2^{j_1} \\ \vdots \\ s_i a_n^{j_1} \\ \mathbf{0}_d \\ \mathbf{0}_d \\ \vdots \\ \mathbf{0}_d \end{bmatrix}_{2n \times d},$$

where $x_{r_{j_1}}$ is the highest preference variable in $m_i$, and

$$Q^{(r_j)}_{(1:2n,(i-1)d+1:i.d)} = B \begin{bmatrix} a_1^j \\ a_2^j \\ \vdots \\ a_n^j \\ \mathbf{0}_d \\ \mathbf{0}_d \\ \vdots \\ \mathbf{0}_d \end{bmatrix}_{2n \times d},$$

for all other $j$'s such that $x_{r_j}$ is present in monomial $i$.

4. The last column block for $Q^{(r_1)}$ is the all ones matrix $\mathbf{1}_{n \times d}$ with a scaling factor, i.e.,

$$Q^{(r_1)}_{(1:2n,s.d+1:(s+1)d)} = B \cdot \mathbf{1}_{2n \times d},$$

and for $j \in [2:k]$, it is the matrix

$$Q^{(r_j)}_{(1:2n,s.d+1:(s+1)d)} = \begin{bmatrix} \mathbf{0}_{n \times d} \\ \mathbf{1}_{n \times d} \end{bmatrix}_{2n \times d}.$$

Roughly, the query-key matrices can be seen as:

$$Q^{(r_1)} = B \begin{bmatrix} \overbrace{a_1^1}^{d} & \overbrace{\mathbf{0}_d}^{d} & \overbrace{s_3 \cdot a_1^1}^{d} & \overbrace{\mathbf{1}_d}^{d} \\ \vdots & & \vdots & \\ a_n^1 & \vdots & s_3 \cdot a_n^1 & \vdots \\ \mathbf{0}_d & & \mathbf{0}_d & \cdots \\ \vdots & & \vdots & \\ \underbrace{\mathbf{0}_d}_{x_{r_1} \text{ not in } m_1} & \underbrace{\mathbf{0}_d}_{m_2 \text{ does not divide } m_1} & \underbrace{\mathbf{0}_d}_{m_3 \text{ divides } m_1} & \mathbf{1}_d \end{bmatrix}_{n \times ((s+1)d)},$$

and for all other $j \in [2:k]$,

$$Q^{(r_j)} = B \begin{bmatrix} \overbrace{a_1^j}^{d} & \overbrace{\mathbf{0}_d}^{d} & \overbrace{s_3 \cdot a_1^j}^{d} & \overbrace{\mathbf{0}_d}^{d} \\ \vdots & & \vdots & \vdots \\ a_n^j & \vdots & s_3 \cdot a_n^j & \mathbf{0}_d \\ \mathbf{0}_d & & \mathbf{0}_d & \cdots & \mathbf{1}_d \\ \vdots & & \vdots & \vdots \\ \underbrace{\mathbf{0}_d}_{x_{r_1} \text{ not in } m_1} & \underbrace{\mathbf{0}_d}_{m_2 \text{ does not divide } m_1} & \underbrace{\mathbf{0}_d}_{m_3 \text{ divides } m_1} & \mathbf{1}_d \end{bmatrix}_{n \times ((s+1)d)}.$$

For the value matrices $V^{(j)} \in \mathbb{R}^{(2n) \times (s+1)d}$, $j \in T \backslash \{1\}$, we define the first column as,

$$V^{(j)}_{(1:2n,1)} = \begin{bmatrix} \mathbf{1}_n^T \\ \mathbf{0}_n^T \end{bmatrix},$$

and for $j \in [2:t] \backslash T$, we define the first column as,

$$V^{(j)}_{(1:2n,1)} = \begin{bmatrix} \mathbf{1}_n^T \\ \mathbf{1}_n^T \end{bmatrix}.$$

All the other columns are completely zero $\mathbf{0}_{2n}^T$.

**Correctness of construction.** We now show that for $\ell_1, \ldots, \ell_t \in [n]$, $h(Q^{(1)}_{\ell_1}, Q^{(2)}_{\ell_2}, \ldots, Q^{(t)}_{\ell_t}) = B^k \langle a^1_{\ell_{r_1}}, \ldots, a^k_{\ell_{r_k}} \rangle$. By definition, $m_1 = x_{r_1} \ldots x_{r_k}$, and it is easy to note that $m_1(Q^{(1)}_{\ell_1}, \ldots, Q^{(t)}_{\ell_t}) = B^k \langle a^1_{\ell_{r_1}}, \ldots, a^k_{\ell_{r_k}} \rangle$. For all the other degree $k$ terms, the inner products from their corresponding blocks are all zeros as we had defined $Q^{(j)}$ as all zeros matrix for all $j \notin T$.

We want to show that for all other $i$'s, $m_i(Q^{(1)}_{\ell_1}, \ldots, Q^{(t)}_{\ell_t}) = 0$. When we compute $m_i(Q^{(1)}_{\ell_1}, \ldots, Q^{(t)}_{\ell_t})$, the $\hat{i}$-th column blocks for $\hat{i} < i$ have some contributions to the inner product $m_i$ if and only if $m_i$ divides $m_{\hat{i}}$ (otherwise $m_i(Q^{(1)}_{\ell_1,(\hat{i}-1)d+1:\hat{i}.d}, \ldots, Q^{(t)}_{\ell_t,(\hat{i}-1)d+1:\hat{i}.d}])$ is zero), and no $\hat{i}$ has a contribution for $\hat{i} > i$ due to the correctness of the monomial ordering. Now, from the choice of $s_i$ as above, it follows that $m_i(Q^{(1)}_{\ell_1}, \ldots, Q^{(t)}_{\ell_t}) = \sum_{\hat{i}=1}^{i} m_i(Q^{(1)}_{\ell_1,(\hat{i}-1)d+1:\hat{i}.d}, \ldots, Q^{(t)}_{\ell_t,(\hat{i}-1)d+1:\hat{i}.d}) = 0$.

For bounding the values of $s_i$'s, we use induction to prove $|s_i| < s^i$. The base case is obviously true. For the induction step, assuming $|s_i| < s^i$, for the $(i+1)$-th monomial, $s_{i+1}$ needs to cancel the contribution to the inner product corresponding to $m_{i+1}$ from each monomial $m_{\hat{i}}$ which is divisible by $m_{i+1}$. The contribution is at most $|s_{\hat{i}}| < s^{\hat{i}}$ (from the induction hypothesis), and hence $|s_{i+1}| < \sum_{\hat{i} \, : \, m_{i+1} | m_{\hat{i}}} |s_{\hat{i}}| < \sum_{\hat{i} \, : \, m_{i+1} | m_{\hat{i}}} s^{\hat{i}} < i.s^i < s^{i+1}$. Therefore, we have $|s_i| < s^s$, which implies $\Gamma = O(s^s B)$, and from the definitions, we obviously have $\Gamma \geq B$ as well. Since, $s = O(1)$, we have $\Gamma = \Theta(B)$.

Further, these query-key and value matrices can be computed in $O(n^{1+o(1)})$ time.

**Approximation yields gap property.** We assume an entry-wise approximation of the self-attention matrix, and the goal is to compute two values, the numerator and the denominator, for computing the softmax. The numerator, for $\ell_1 \in [2n]$, is given by

$$\bar{P}_{\ell_1} = \sum_{\ell_2, \ldots, \ell_t \in [2n]} \exp\left(\frac{1}{\tilde{d}} h(Q^{(1)}_{\ell_1}, Q^{(2)}_{\ell_2}, \ldots, Q^{(t)}_{\ell_t})\right) V^{(2)}_{\ell_2} \odot \ldots \odot V^{(t)}_{\ell_t},$$

and the denominator by

$$R_{\ell_1} = \sum_{\ell_2, \ldots, \ell_t \in [2n]} \exp\left(\frac{1}{\tilde{d}} h(Q^{(1)}_{\ell_1}, Q^{(2)}_{\ell_2}, \ldots, Q^{(t)}_{\ell_t})\right).$$

The $\ell_1$-th row of the $Att^{(h)}$ will be $\frac{\bar{P}_{\ell_1}}{R_{\ell_1}}$, and we want to find an entry-wise approximation. Since in our choice of the value matrices, the first coordinate of $\bar{P}_{\ell_1}$ is the only non-zero one, and its summation is only upto the top half of the value matrices, $\ell_j \in [n]$, for $j \in [2:k]$. The only non-zero part of the numerator, that we care about, is therefore given by

$$P_{\ell_1} = \sum_{\substack{\ell_i \in [n] \, : \, i \in T \\ \ell_j \in [2n] \, : \, j \notin T}} \exp\left(\frac{1}{\tilde{d}} h(Q^{(1)}_{\ell_1}, Q^{(2)}_{\ell_2}, \ldots, Q^{(t)}_{\ell_t})\right).$$

If we have an entry-wise $\gamma$-approximation of $Att^{(h)}$, let $x_{\ell_1}$-th be the approximation for the $(\ell_1, 1)$ entry of $Att^{(h)}$. By definition, we have

$$\left| x_{\ell_1} - \frac{P_{\ell_1}}{R_{\ell_1}} \right| < \gamma. \tag{13}$$

**Bounds on denominator.** Consider the summation $\sum_{\ell_2,\ldots,\ell_t \in [2n]} \exp\left(\frac{1}{\tilde{d}} h(Q^{(1)}_{\ell_1}, Q^{(2)}_{\ell_2}, \ldots, Q^{(t)}_{\ell_t})\right)$. Define $\lambda$ as the factor such that when only the $r_1, \ldots, r_k$ coordinates are $B.\mathbf{1}_d$ and the remaining are zeros, i.e.,

$$h(\mathbf{0}_d, \underbrace{B.\mathbf{1}_d, \ldots, B.\mathbf{1}_d}_{k}, \mathbf{0}_d, \ldots, \mathbf{0}_d) = \lambda d B^k.$$

It is easy to see that $\lambda = 1 + o(1)$, since the evaluation of $h$ at these values will give a $B^k$ from the first monomial, and the other $s - 1$ monomials will give at most $(s-1)B^{k-1} = o(B^k)$.

For the choice of $Q^{(j)}$'s, we have

$$R_{\ell_1} > \sum_{\substack{\ell_i \in [n+1:2n] \,:\, i \in T \\ \ell_j \in [2n] \,:\, j \notin T}} \exp(\frac{1}{\tilde{d}} h(Q^{(1)}_{\ell_1}, Q^{(2)}_{\ell_2}, \ldots, Q^{(t)}_{\ell_t}))$$

$$> \sum_{\ell_2,\ldots,\ell_t \in [n:2n]} \exp(\lambda d B^k / \tilde{d}) = n^{t-1} e^{(B^k \frac{\lambda}{s+1})},$$

since all the $Q^{(r_j)}_{\ell_{r_j}}$'s, for $j \in [k]$, have the zeros in the last column-block and $\mathbf{1}_d$ along with the scaling factor $B$, which makes all the monomials of $h$ give inner product $dB^k$.

For the upper bound on $R_{\ell_1}$, we have to use the maximum possible value of $h(Q^{(1)}_{\ell_1}, Q^{(2)}_{\ell_2}, \ldots, Q^{(t)}_{\ell_t})$, irrespective of whether $\ell_j$'s are in $[n]$ or $[n+1:2n]$. Let us consider a choice of $\ell_2, \ldots, \ell_t \in [2n]$. If all the $\ell_{r_j}$'s, for $j \in T \backslash \{1\}$, are in $[n+1:2n]$, then the value of $h(Q^{(1)}_{\ell_1}, Q^{(2)}_{\ell_2}, \ldots, Q^{(t)}_{\ell_t})$ obtained will be be $\lambda d B^k$. Otherwise, there are some (but not all) $\ell_j$'s in $[n]$ for $j \in [2 : k]$, where the monomial of degree $< k$ containing only those variables will be at most $s^s d B^{k-1}$, and the maximum value will be obtained from the first term, which can be at most $B^k \langle a^1_{\ell_{r_1}}, \ldots, a^k_{\ell_{r_k}} \rangle = (d-1)B^k$. Thus, in this case, the maximum value of $h(Q^{(1)}_{\ell_1}, Q^{(2)}_{\ell_2}, \ldots, Q^{(t)}_{\ell_t})$ would be $(d - 1 + o(1))B^k$ which is still less than $\lambda d B^k$.

Therefore,

$$R_{\ell_1} = \sum_{\ell_2,\ldots,\ell_t \in [2n]} \exp\left(\frac{1}{\tilde{d}} h(Q^{(1)}_{\ell_1}, Q^{(2)}_{\ell_2}, \ldots, Q^{(t)}_{\ell_t})\right)$$

$$\leq \sum_{\ell_2,\ldots,\ell_t \in [2n]} \exp\left(\frac{\lambda d B^k}{\tilde{d}}\right) = 2^{t-1} n^{t-1} e^{(B^k \frac{\lambda}{s+1})}.$$

Therefore,

$$n^{t-1} e^{(B^k \frac{\lambda}{s+1})} < R_{\ell_1} < 2^{t-1} n^{t-1} e^{(B^k \frac{\lambda}{s+1})}. \tag{14}$$

**Bounds on numerator.** Now, we will show that if a vector tuple exists with the proper target inner product (a positive certificate for $\gamma$-Gap-$k$IP), then $P_{\ell_1}$ is so large that $x_{\ell_1}$ (Equation 13) is larger than a fixed threshold. Here, we first show a lower bound on $P_{\ell_1}$. Otherwise, we will show that $x_{\ell_1}$ is small since every inner product will be scaled down by a gap due to the approximation promise.

Consider $\ell_1 \in [n]$ when there exists $\ell^0_{r_1}, \ldots, \ell^0_{r_k} \in [n]$ such that the inner product $\langle a^1_{\ell^0_{r_1}}, \ldots, a^k_{\ell^0_{r_k}} \rangle = M \log n$ (it is quite possible that $r_1 = 1$, in which case we will only consider $\ell_1 = \ell^0_1$). Then, we

have

$$P_{\ell_1} = \sum_{\substack{\ell_i \in [n] \,:\, i \in T \\ \ell_j \in [2n] \,:\, j \notin T}} \exp\left(\frac{1}{\tilde{d}} h(Q_{\ell_1}^{(1)}, Q_{\ell_2}^{(2)}, \ldots, Q_{\ell_t}^{(t)})\right)$$

$$> \sum_{\substack{\ell_i = \ell_i^0 \,:\, i \in T \\ \ell_j \in [2n] \,:\, j \notin T}} \exp\left(\frac{1}{\tilde{d}} h(Q_{\ell_1}^{(1)}, Q_{\ell_2}^{(2)}, \ldots, Q_{\ell_t}^{(t)})\right)$$

$$= (2n)^{t-k-1} \exp\left(\frac{1}{\tilde{d}} B^k \langle a_{\ell_{r_1}^0}^1, a_{\ell_{r_2}^0}^2, \ldots, a_{\ell_{r_t}^0}^t \rangle\right)$$

$$= (2n)^{t-k-1} \exp\left(B^k \frac{M}{(s+1)c}\right),$$

where the second equality follows from the construction of the $Q^{(j)}$'s. Using the upper bound of $R_{\ell_1}$ in Equation 14, we get,

$$\frac{P_{\ell_1}}{R_{\ell_1}} > \frac{(2n)^{t-k-1} e^{\left(B^k \frac{M}{(s+1)c}\right)}}{(2n)^{t-1} e^{\left(B^k \frac{\lambda}{s+1}\right)}} = \frac{e^{\left(B^k \left(\frac{M}{c} - \lambda\right)/(s+1)\right)}}{(2n)^{k-1}}.$$

Using $x_{\ell_1} > \frac{P_{\ell_1}}{R_{\ell_1}} - \gamma$ (Equation 13), we get

$$x_{\ell_1} > \frac{e^{\left(B^k \left(\frac{M}{c} - \lambda\right)/(s+1)\right)}}{(2n)^{k-1}} - \gamma. \tag{15}$$

Now, for finding an upper bound on $x_{\ell_1}$ when an exact inner product tuple does not exist, we use

$$P_{\ell_1} = \sum_{\substack{\ell_i \in [n] \,:\, i \in T \\ \ell_j \in [2n] \,:\, j \notin [T]}} \exp\left(\frac{1}{\tilde{d}} h(Q_{\ell_1}^{(1)}, Q_{\ell_2}^{(2)}, \ldots, Q_{\ell_t}^{(t)})\right)$$

$$= (2n)^{t-k-1} \sum_{\ell_i \in [n] \,:\, i \in T} \exp\left(\frac{1}{\tilde{d}} B^k \langle a_{\ell_{r_1}}^1, a_{\ell_{r_2}}^2, \ldots, a_{\ell_{r_k}}^k \rangle\right),$$

using the construction of $Q^{(r_j)}$'s. Now, using the gap property of inner products in our instance of $\varepsilon$-Gap-$k$IP, we have

$$P_{\ell_1} = (2n)^{t-k-1} \sum_{\ell_i \in [n] \,:\, i \in T} \exp\left(\frac{1}{\tilde{d}} B^k \langle a_{\ell_1}^1, a_{\ell_2}^2, \ldots, a_{\ell_k}^k \rangle\right)$$

$$< (2n)^{t-k-1} \sum_{\ell_i \in [n] \,:\, i \in T} \exp\left(\frac{B^k}{\tilde{d}} (1 - \varepsilon) M \log n\right)$$

$$\implies P_{\ell_1} < 2^{t-k-1} n^{t-1} e^{\left((1-\varepsilon) B^k \frac{M}{(s+1)c}\right)}.$$

Finally, using the lower bound of $R_{\ell_1}$ (Equation 14), we get

$$\frac{P_{\ell_1}}{R_{\ell_1}} < \frac{2^{t-k-1} n^{t-1} e^{\left((1-\varepsilon) B^k \frac{M}{(s+1)c}\right)}}{n^{t-1} e^{\left(B^k \frac{\lambda}{s+1}\right)}} = 2^{t-k-1} e^{\left(B^k \left(\frac{(1-\varepsilon)M}{c} - \lambda\right)/(s+1)\right)},$$

and the bound on $x_{\ell_1}$ from Equation 13 implies,

$$x_{\ell_1} < \frac{P_{\ell_1}}{R_{\ell_1}} + \gamma < 2^{t-k-1} e^{\left(B^k \left(\frac{(1-\varepsilon)M}{c} - \lambda\right)/(s+1)\right)} + \gamma. \tag{16}$$

**Wrapping up.** In order to differentiate between the cases, we must have the lower bound of $x_{\ell_1}$ when a positive instance for $\varepsilon$-Gap-$k$IP tuple exists, Equation 15, must be greater than the upper bound when such an instance does not exist, Equation 16:

$$\frac{1}{e^{\left(\varepsilon B^k \frac{M}{(s+1)c}\right)}} \frac{2^{t-k-1}}{e^{\left(B^k \left(\lambda - \frac{M}{c}\right)/(s+1)\right)}} + \gamma < \frac{1}{(2n)^{k-1} e^{\left(B^k \left(\lambda - \frac{M}{c}\right)/(s+1)\right)}} - \gamma,$$

which is true for the choice of $\gamma \leq n^{-C_a}$ and $B > C_b (\log n)^{1/k}$, for large enough constants $C_a, C_b > 0$. This would make $e^{(\varepsilon B^k \frac{M}{c})}$ large enough and $\gamma$ small enough, such that the inequality will be valid.

Now, since $s$ is constant, the maximum absolute value of the entries of the query-key matrices are $\Omega(B) = \Omega((\log n)^{1/k})$, which proves our result. Therefore, if we can find an algorithm for finding an entry-wise $\gamma$-approximation of $Att^{(h)}$ for $\mathsf{APAC}^{(h)}$ with these parameters, that runs in time $n^{k-\Omega(1)}$, then SETH will be refuted (Theorem E.7). $\qquad\square$

### E.2.2 Time Lower Bounds Based on Substructure of Polynomial Using Max-$k$SAT Conjecture

In the second part of Theorem E.3, we prove a stronger lower bound when the monomials of $h$ contains an elementary symmetric polynomial of degree $k$ in $t_0$ variables where $k < t_0 \leq t$. The underlying conjecture for this lower bound is the Max-$k$SAT. We first start with a problem called HypergraphIP (Definition E.9), which is at least as hard as Max-$k$SAT, show that its gap version, $\varepsilon$-Gap-HypergraphIP (Definition E.11), is also at least as hard as HypergraphIP using Rubinstein (2018); Abboud & Ron-Zewi (2025), and finally show that computing an entry-wise $\gamma$-approximation of $Att^{(h)}$ efficiently would solve $\varepsilon$-Gap-HypergraphIP faster, thereby refuting Max-$k$SAT conjecture.

**Definition E.9** (HypergraphIP$_{t,k}^{n,d}$). *For positive integers $t, k$, given $t$ sets of vectors $A^1, \ldots, A^t \in \{0,1\}^d$ with $|A^1| = \ldots = |A^t| = n$, and target inner products $m_1, \ldots, m_{\binom{t}{r}}$, the problem HypergraphIP$_{t,k}^{n,d}$ is to decide if there exist vectors $a_1 \in A^1, \ldots, a_t \in A^t$ such that for all subsets $S \in \binom{[t]}{k}$, we have $\langle a_{S[1]}, \ldots, a_{S[k]} \rangle = m_S$, where $m_S$ is the target inner product corresponding to the given $k$-sized subset among the $\binom{t}{k}$ choices.*

We will drop $n, d$ from the superscript and not include the target inner products as the parameters to make the problem definitions less cumbersome. This problem again has a hardness result, as follows.

**Lemma E.10** ((Alman & Vassilevska Williams, 2020, Theorem 23)). *Assuming the Max-kSAT conjecture (Hypothesis 2), for every $\delta > 0$ and every positive integer $t, k$, there exists a constant $c > 0$ and target inner products $m_1, \ldots, m_{\binom{t}{r}} \in \{0, \ldots, d\}$ such that HypergraphIP$_{t,k}^{n,c\log n}$ cannot be solved in time $O(n^{(1-\delta)t})$.*

We can again reduce HypergraphIP to its gap version Gap-HypergraphIP to show that this problem is hard as well.

**Definition E.11** ($\varepsilon$-Gap-HypergraphIP$_{t,k}^{n,d}$). *For every $\varepsilon \in (0,1)$ and positive integers $t, k$, given $t$ sets of vectors $A^1, \ldots, A^t \in \{0,1\}^d$ with $|A^1| = \ldots = |A^t| = n$, and target inner product $m \in \{0, \ldots, d\}$, along with the promise that for every $a_1 \in A^1, \ldots, a_t \in A^t$ and $\forall S \in \binom{[t]}{k}$,*

- *either, $\langle a_{S[1]}, \ldots, a_{S[k]} \rangle = m$,*

- *or, $\langle a_{S[1]}, \ldots, a_{S[k]} \rangle \leq (1-\varepsilon)m$,*

*the problem $\varepsilon$-Gap-HypergraphIP$_{t,k}^{n,d}$ is to decide if there exist vectors $a_1 \in A^1, \ldots, a_t \in A^t$ such that $\forall S \in \binom{[t]}{k}$, we have $\langle a_{S[1]}, \ldots, a_{S[k]} \rangle = m$.*

Again, similar to Gap-IP$\Delta$, for Gap-HypergraphIP, we consider the target inner products to be the same for all the subsets of inner products, since the Rubinstein (2018)-like reduction accommodates this, and we need this condition for reducing Gap-HypergraphIP to entry-wise approximation of $Att^{(h)}$.

The hardness of $\varepsilon$-Gap-HypergraphIP follows from a proof very similar to Theorem C.8, given by the following corollary.

**Corollary E.12.** *For positive integers $t, k$ with $k \geq 3$, and every $\delta > 0$, assuming the Max-kSAT conjecture, there exists a constant $c$ and target inner product $m \in \{0, \ldots, c\log n\}$, the problem $\varepsilon$-Gap-HypergraphIP$_{t,k}^{n,c\log n}$ cannot be solved in time $O(n^{(1-\delta)t})$.*

*Proof.* We can again use the reductions of Lemma C.9. We start with an instance of $\mathsf{HypergraphIP}_{t,k}^{n,d}$ having sets of vectors $A^1, \ldots, A^t \subseteq \{0,1\}^d$ containing $n$ vectors each, and reduce that to $n^{o(1)}$ instances of $\varepsilon\text{-}\mathsf{Gap\text{-}HypergraphIP}_{t,k}^{n,\tilde{d}}$ having sets of vectors $B^1, \ldots, B^k \subseteq \{0,1\}^{\tilde{d}}$ for $\tilde{d} = \Theta(\log n)$, where each $B^i$ contains $n$ vectors.

The proof goes as– for each $k$-tuple $(j_1, \ldots, j_k) \in \binom{[t]}{k}$, we reduce $A^{j_1}, \ldots, A^{j_k}$, an instance of $k\mathsf{IP}$, to $n^{o(1)}$ instances of $\varepsilon\text{-}\mathsf{Gap\text{-}}k\mathsf{IP}$ of dimension $d_0$ (using methods of Alman & Song (2024); Rubinstein (2018); Abboud & Ron-Zewi (2025)). Then, we combine each of the $\varepsilon\text{-}\mathsf{Gap\text{-}}k\mathsf{IP}$ instances for all $(j_1, \ldots, j_k) \in \binom{t}{k}$ by creating $\binom{t}{k}$ column blocks, each of dimension $d_0$, as done in the proof of Theorem C.8, where the block corresponding to $(j_1, \ldots, j_k)$ will contain vectors obtained from the above reduction, and the rest will be zero. The hardness result also holds true when the target inner product for every subset of $B^1, \ldots, B^k$ are equal. $\qquad\square$

Now, to show hardness of computing an entry-wise $\gamma$-approximation of $Att^{(h)}$ where $h$ satisfies the conditions of Part 2 of Theorem E.3, we reduce $\varepsilon\text{-}\mathsf{Gap\text{-}HypergraphIP}_{t_0,r}$ (which we know is at least as hard as $\mathsf{Max\text{-}}k\mathsf{SAT}$), to an entry-wise approximation of $Att^{(h)}$. Armed with Corollary E.12, we are now ready to prove the following lemma which completes the second part of Theorem E.3.

**Lemma E.13** ($\varepsilon\text{-}\mathsf{Gap\text{-}HypergraphIP}$ to $\mathsf{APAC}^{(h)}$)**.** *For every constant $\varepsilon > 0$, every $\delta \in (0, 0.01)$, every $c, M > 0$, given an attention polynomial $h(x_1, \ldots, x_t)$ of degree $k \geq 3$ having $s$ monomials, such that the set of monomials of $h$ contains as a subset all the monomials of the elementary symmetric polynomial in $t_0 < t$ variables of degree $k$, where $t, k, s, t_0$ are constants, there exist constants $C_a > 0$ and $C_b > 0$ such that if $\mathsf{APAC}^{(h)}(2n, (s + 1)c\log n, \Gamma = C_b(\log n)^{1/k}, \gamma = n^{-C_a})$ (Definition A.6) with query-key matrices $Q^{(1)}, \ldots, Q^{(t)} \in [-\Gamma, \Gamma]^{2n \times (s+1)c\log n}$ and value matrices $V^{(2)}, \ldots, V^{(t)} \in \mathbb{R}^{2n \times (s+1)c\log n}$ can be solved in time $O(n^{t_0 - \delta})$, then $\varepsilon\text{-}\mathsf{Gap\text{-}HypergraphIP}_{t_0,k}^{n,c\log n}$ (Definition E.11) with target inner product $m = M \log n$ can also be solved in $O(n^{t_0 - \delta})$ time for any constant $M$.*

*Proof.* First, we consider that the subset of the monomials of $h$, which constitute a symmetric polynomial in $t_0$ variables of degree $k$, is given by the set of subset of variables $x_{r_1}, \ldots, x_{r_{t_0}}$. Let us denote $T := \{r_1, \ldots, r_{t_0}\} \subseteq [t]$.

Let us start instance of $\varepsilon\text{-}\mathsf{Gap\text{-}HypergraphIP}_{t_0,k}^{n,d=c\log n}$ with $t_0$ sets of vectors be $A^1, \ldots, A^{t_0} \subseteq \{0,1\}^d$, having $n$ vectors each, and the target inner product being $m = M \log n$ with a promise of gap given with a constant approximation factor of $\varepsilon$. More specifically, we want to check if there exists $\ell_{r_1}, \ldots, \ell_{r_{t_0}} \in [n]$ such that for all $(j_1, \ldots, j_k) \in \binom{[t_0]}{k}$, we have $\langle a_{\ell_{j_1}}^{j_1}, \ldots, a_{\ell_{j_k}}^{j_1} \rangle = m$, i.e.,

$$\sum_{j_1, \ldots, j_k \in \binom{[t_0]}{k}} \langle a_{\ell_{r_{j_1}}}^{j_1}, \ldots, a_{\ell_{r_{j_k}}}^{j_k} \rangle = \binom{t_0}{k} m =: m_0,$$

where $m_0 = M_0 \log n$. We also have the promise that for every other tuple $\ell_{r_1}, \ldots, \ell_{r_{t_0}} \in [n]$ where $\mathsf{HypergraphIP}_{t_0,k}$ property is not satisfied,

$$\sum_{j_1, \ldots, j_k \in \binom{[t_0]}{k}} \langle a_{\ell_{r_{j_1}}}^{j_1}, \ldots, a_{\ell_{r_{j_k}}}^{j_k} \rangle < \left( \binom{t_0}{k} - 1 \right) m + (1 - \varepsilon)m =: (1 - \varepsilon_0)m_0,$$

for another constant $\varepsilon_0 = \varepsilon / \binom{t_0}{k}$.

**Constructing the matrices.** Now, we define the matrices $Q^{(j)}$'s, such that

$$h(Q_{\ell_1}^{(1)}, \ldots, Q_{\ell_t}^{(t)}) = \Lambda \sum_{j_1, \ldots, j_k \in \binom{[t_0]}{k}} \langle a_{\ell_{r_{j_1}}}^{j_1}, \ldots, a_{\ell_{r_{j_k}}}^{j_k} \rangle,$$

for a scaling factor $\Lambda$, in a construction quite similar to the proof of Lemma E.8. The query-key matrices will be $Q^{(1)}, \ldots, Q^{(t)} \in [-\Gamma, \Gamma]^{\tilde{n} \times \tilde{d}}$, for $\tilde{n} = 2n, \tilde{d} = (s + 1)d$, defined as follows using a scaling value $B = \omega(1)$ which we will choose later:

1. For $Q^{(j)}$'s, if $j \notin T$, we just make the entire matrix zero $\mathbf{0}_{2n \times (s+1)d}$.

2. For some $i \in [s]$, if $m_i$ is equal to some $x_{r_{j_1}} \ldots x_{r_{j_k}}$ for $j_1, \ldots, j_k \in \binom{[t_0]}{k}$, we define that block as:

$$
Q^{(r_{j_\ell})}_{(1:2n,(i-1)d+1:i.d)} = B \begin{bmatrix} a_1^{j_\ell} \\ a_2^{j_\ell} \\ \vdots \\ a_n^{j_\ell} \\ \mathbf{0}_d \\ \mathbf{0}_d \\ \vdots \\ \mathbf{0}_d \end{bmatrix}_{2n \times d},
$$

for all $\ell \in [k]$, and

$$
Q^{(j)}_{(1:2n,(i-1)d+1:i.d)} = \mathbf{0}_{2n \times d},
$$

for all other $j \in [t] \backslash \{r_{j_1}, \ldots, r_{j_k}\}$.

3. However, if for $i \in [s]$, monomial $i$ has degree $\leq k - 1$, let this be equal to $x_{r_{j_1}} \ldots x_{r_{j_{k_i}}}$, where $j_1, \ldots, j_{k_i} \in [t_0]$, $k_i < k$ is the degree (note that if the variables are anything outside $T$, we have defined the corresponding query-key matrices to be zeros anyway). Let $s_i$ be the integer which is the negation of the number of occurrences of this monomials in each of the monomials ordered higher preference than $i$.

   As before, $s_i$ is the sum defined by adding:

   - $-1$ whenever $\ell < i$ and $m_\ell$ is of degree $k$.
   - $-s_\ell$ whenever $\ell < i$, the monomial $m_\ell$ is of degree $\leq k$, $m_i$ divides $m_\ell$, and the highest preference variable of $m_\ell$ is also present in $m_i$.
   - $-1$ whenever $\ell < i$, the monomial $m_\ell$ is of degree $\leq k$ and $m_i$ divides $m_\ell$, but the highest preference variable of $m_\ell$ is also present in $m_i$.
   - $0$ in all other cases.

   If $x_{r_j}$ is not present in monomial $i$, we just set

$$
Q^{(r_j)}_{(1:2n,(i-1)d+1:i.d)} := \mathbf{0}_{2n \times d},
$$

   otherwise:

$$
Q^{(r_{j_1})}_{(1:2n,(i-1)d+1:i.d)} = B \begin{bmatrix} s_i a_1^{j_1} \\ s_i a_2^{j_1} \\ \vdots \\ s_i a_n^{j_1} \\ \mathbf{0}_d \\ \mathbf{0}_d \\ \vdots \\ \mathbf{0}_d \end{bmatrix}_{2n \times d},
$$

   where $x_{r_{j_1}}$ is the highest preference variable of $m_i$, and

$$
Q^{(r_j)}_{(1:2n,(i-1)d+1:i.d)} = B \begin{bmatrix} a_1^{j} \\ a_2^{j} \\ \vdots \\ a_n^{j} \\ \mathbf{0}_d \\ \mathbf{0}_d \\ \vdots \\ \mathbf{0}_d \end{bmatrix}_{2n \times d},
$$

   for all other $j$'s such that $x_{r_j}$ is present in monomial $i$.

4. The last column block for $Q^{(r_1)}$ is the all ones matrix $\mathbf{1}_{n \times d}$ with a scaling factor, i.e.,

$$Q^{(r_1)}_{(1:2n,s.d+1:(s+1)d)} = B \cdot \mathbf{1}_{2n \times d},$$

and for $j \in \{2, \ldots, t_0\}$, it is the all zeros matrix

$$Q^{(r_j)}_{(1:2n,s.d+1:(s+1)d)} = \begin{bmatrix} \mathbf{0}_{n \times d} \\ \mathbf{1}_{n \times d} \end{bmatrix}_{2n \times d}.$$

For the value matrices $V^{(j)} \in \mathbb{R}^{(2n) \times (s+1)d}$, $j \in T \backslash \{1\}$, we define the first column as,

$$V^{(j)}_{(1:2n,1)} = \begin{bmatrix} \mathbf{1}_n^T \\ \mathbf{0}_n^T \end{bmatrix},$$

and for $j \in [2:t] \backslash T$, we define the first column as,

$$V^{(j)}_{(1:2n,1)} = \begin{bmatrix} \mathbf{1}_n^T \\ \mathbf{1}_n^T \end{bmatrix},$$

with every other columns $\mathbf{0}_{2n}^T$.

**Correctness of construction.** Again, similar to the proof of Lemma E.8, we can prove that this construction does indeed give

$$h(Q^{(1)}_{\ell_1}, \ldots, Q^{(t)}_{\ell_t}) = B^k \sum_{j_1, \ldots, j_r \in S^T_{t_0}} \langle a^{j_1}_{\ell_{r_{j_1}}}, \ldots, a^{j_k}_{\ell_{r_{j_k}}} \rangle,$$

and the entries of the query-key matrices are in $[-\Gamma, \Gamma]$ for $B < \Gamma < O(s^s B)$.

Also, these query-key and value matrices can be computed in $O(n^{1+o(1)})$ time.

**Approximation yields gap property.** As before, let us assume there exists an entry-wise approximation $x_{\ell_1}$ of the $(\ell_1, 1)$-th element of $Att^{(h)}$ such that

$$\left| x_{\ell_1} - \frac{P_{\ell_1}}{R_{\ell_1}} \right| < \gamma,$$

where

$$P_{\ell_1} = \sum_{\substack{\ell_i \in [n] \,:\, i \in T \\ \ell_j \in [2n] \,:\, j \notin T}} \exp\left( \frac{1}{d} h(Q^{(1)}_{\ell_1}, Q^{(2)}_{\ell_2}, \ldots, Q^{(t)}_{\ell_t}) \right),$$

$$R_{\ell_1} = \sum_{\ell_2, \ldots, \ell_t \in [2n]} \exp\left( \frac{1}{d} h(Q^{(1)}_{\ell_1}, Q^{(2)}_{\ell_2}, \ldots, Q^{(t)}_{\ell_t}) \right),$$

and the $(\ell_1, 1)$-th element of $Att^{(h)}$ is $\frac{P_{\ell_1}}{R_{\ell_1}}$.

**Bounds on $P_{\ell_1}, R_{\ell_1}$.** Similar to before, we can prove

$$n^{t-1} e^{\left( \frac{\lambda B^k}{(s+1)} \right)} < R_{\ell_1} < 2^{t-1} n^{t-1} e^{\left( \frac{\lambda B^k}{(s+1)} \right)},$$

where $\lambda = 1 + o(1)$. For the numerator, we can show that when a positive certificate of $\varepsilon$-Gap-HypergraphIP does exist (if $r_1 \neq 1$, then this holds for all $\ell_1$'s, otherwise, there will be a fixed $\ell_1$ such that $a^1_{\ell_1}$ is included in the positive certificate),

$$P_{\ell_1} > (2n)^{t-k-1} e^{\left( B^k \frac{M_0}{(s+1)c} \right)},$$

which implies

$$x_{\ell_1} > \frac{P_{\ell_1}}{R_{\ell_1}} - \gamma > \frac{e^{\left( B^k \left( \frac{M_0}{c} - \lambda \right) / (s+1) \right)}}{(2n)^{k-1}} - \gamma. \tag{17}$$

Otherwise, if no positive certificate of $\varepsilon$-Gap-HypergraphIP exists when $r_1 \neq 1$, or when $r_1 = 1$, the positive certificate, if exists, does not contain the vector $a^1_{\ell_1}$, then

$$P_{\ell_1} < 2^{t-k-1} n^{t-1} e^{\left( (1-\varepsilon_0) B^k \frac{M_0}{(s+1)c} \right)},$$

and therefore,

$$x_{\ell_1} < \frac{P_{\ell_1}}{Q_{\ell_1}} + \gamma < e^{\left( B^k \left( \frac{(1-\varepsilon_0) M_0}{c} - \lambda \right) / (s+1) \right)} + \gamma. \tag{18}$$

**Wrapping up.** In order to maintain a gap between the cases of an HypergraphIP existing, we require the lower bound (Equation 18) must be less than the upper bound (Equation 17)

$$\frac{1}{e^{\left(\varepsilon_0 B^k \frac{M_0}{(s+1)c}\right)}} \frac{2^{t-k-1}}{e^{\left(B^k \left(\lambda - \frac{M_0}{c}\right)/(s+1)\right)}} + \gamma < \frac{1}{(2n)^{k-1} e^{\left(B^k \left(\lambda - \frac{M_0}{c}\right)/(s+1)\right)}} - \gamma.$$

Now, there exist large enough constants $C_a, C_b > 0$ such that this inequality is satisfied for $\gamma \leq n^{-C_a}$ and $B \geq C_b (\log n)^{1/k}$.

This proves that any algorithm for an entry-wise $\gamma$-approximation of $Att^{(h)}$ having maximum value of the entries $\Gamma = \Omega((\log n)^{1/k})$ requires time $\Omega(n^{t_0})$, assuming the Max-$k$SAT conjecture, since if APAC$^{(h)}$ could be solved in $O(n^{t_0 - \delta})$ time, then that would imply Max-$k$SAT could be solved in $2^{(1-\Omega(\delta))n}$ time (Corollary E.12), something that can not be true for an absolute constant $\delta > 0$ (Hypothesis 2). $\qquad \square$

**Remark 1.** *In Lemma E.13, for computing* APAC$^{(h)}$*, when $h$ is in $t$ variables, of degree $k$ and contains as a subpolynomial an elementary symmetric polynomial in $t_0 = t$ variables and degree $k$, the time-complexity is lower bounded by $\Omega(n^t)$. This is the strongest time complexity lower bound we can achieve, as the trivial algorithm for summing over the indices of all the query-key matrix also requires $O(n^t)$ time and we say that this is the best we can hope for!*

### E.2.3 TIME LOWER BOUNDS FOR DEGREE 2 POLYNOMIALS USING MAX-2SAT CONJECTURE

In this section, we prove the final part of Theorem E.3, where we show a lower bound for a certain subcase of $h$ when the degree is 2. For the remaining degree 2 cases, we have already shown in Sections 3.2 and D that they can be computed in $O(n^2)$ time, which is essentially tight from Part 1 of Theorem E.3.

Unlike using SETH which proves lower bounds which are integer powers of $n$, in order to prove lower bounds of the form $n^\omega$, we use the Max-2SAT conjecture (Hypothesis 3) by giving a reduction from $\varepsilon$-Gap-IP$\Delta$ (Theorem C.8) to entry-wise approximation of $Att^{(h)}$.

The reductions work as, we first use the reduction of Max-2SAT to IP$\Delta$, then reduction of IP$\Delta$ to a new problem IP-DIR-$r$CYC using Alman & Vassilevska Williams (2020), which then is reduced to its gap version containing $n^{o(1)}$ instances of $\varepsilon$-Gap-IP-DIR-$r$CYC. Finally, we reduce $\varepsilon$-Gap-IP-DIR-$r$CYC to computing an entry-wise approximation of $Att^{(h)}$.

For these sets of reductions, we first define the new problem of IP-DIR-$r$CYC, which was introduced in Alman & Vassilevska Williams (2020).

**Definition E.14** (IP-DIR-$r$CYC). *For a positive integer $r$, given $r$ sets of vectors $A^1, \ldots, A^r \subseteq \{0,1\}^d$ with $|A^1| = \ldots = |A^r| = n$, and target inner products $m_1, \ldots, m_r \in \{0, \ldots, d\}$, the problem IP-DIR-$r$CYC$_{n,d}$ is to decide if there exist vectors $a_1 \in A^1, \ldots, a_r \in A^r$ such that simultaneously $\langle a_1, a_2 \rangle = m_1, \langle a_2, a_3 \rangle = m_2, \ldots, \langle a_{r-1}, a_r \rangle = m_{r-1}, \langle a_r, a_1 \rangle = m_r$.*

Naturally, to prove hardness of entry-wise approximation of poly-attention, we will again require the hardness of the gap version of this problem, $\varepsilon$-Gap-IP-DIR-$r$CYC.

**Definition E.15** ($\varepsilon$-Gap-IP-DIR-$r$CYC). *For every $\varepsilon > 0$ and positive integer $r$, given $r$ sets of vectors $A^1, \ldots, A^r \in \{0,1\}^d$ with $|A^1| = \ldots = |A^r| = n$, and a target inner product $m \in \{0, \ldots, d\}$ along with the promise that for all $i \in [r]$, for all vectors $a_i \in A^i$, and $a_{i+1 \bmod r} \in A^{i+1 \bmod r}$,*

- *either $\langle a_i, a_{(i+1) \bmod r} \rangle = m$,*

- *or $\langle a_i, a_{(i+1) \bmod r} \rangle \leq (1 - \varepsilon)m$,*

*the problem of $\varepsilon$-Gap-IP-DIR-$r$CYC$_{n,d}$ is to decide if there exist vectors $a_1 \in A^1, \ldots, a_r \in A^r$ such that simultaneously $\langle a_1, a_2 \rangle = \langle a_2, a_3 \rangle = \ldots = \langle a_{r-1}, a_r \rangle = \langle a_r, a_1 \rangle = m$.*

Now, we know that IP-DIR-$r$CYC is at least as hard as IP$\Delta$, which in turn is at least as hard as Max-2SAT (Lemma C.7), given by the following lemma. An OV version of this lemma was proved

in (Alman & Vassilevska Williams, 2020, Lemma 21), i.e., when the target inner products are zero, by reducing OV$\Delta$ to OV-DIR-$r$CYC, but all the proofs work similarly for reducing IP$\Delta$ to IP-DIR-$r$CYC as well.

**Lemma E.16** (IP$\Delta$ to IP-DIR-$r$CYC Alman & Vassilevska Williams (2020)). *For every $\delta > 0$ and positive integer $r \geq 3$, if IP-DIR-$r$CYC$_{n,d}$, can be computed in $O(n^{\omega-\delta})$ time, then IP$\Delta_{n,d}$ can also be computed in time $O(n^{\omega-\delta})$.*

Again, the $\varepsilon$-Gap-IP-DIR-$r$CYC is at least as hard as IP-DIR-$r$CYC using proofs very similar to Theorem C.8.

**Corollary E.17.** *For every $\delta > 0$, positive integer $r \geq 3$ and every constant $\varepsilon > 0$, assuming the Max2SAT conjecture, there exists a constant $c > 0$ and target inner product $m \in \{0, \ldots, d\}$, such that $\varepsilon$-Gap-IP-DIR-$r$CYC$_{n,c\log n}$ cannot be solved in time $O(n^{\omega-\delta})$.*

*Proof.* We prove the hardness of $\varepsilon$-Gap-IP-DIR-$k$CYC by starting with a hard instance of IP-DIR-$r$CYC containing sets vectors $A^1, \ldots, A^r \subseteq \{0,1\}^d$, where $n = |A^i|$ and $d = c\log n$.

Following the technique of the proof of Theorem C.8, we consider $A^i, A^{i+1 \bmod r}$ for each $i \in [r]$ as a 2IP instance, and reduce it to $n^{o(1)}$ many instances of $\varepsilon$-GapIP having two sets $n$ vectors of dimension $d_0$. For the final instance of $\varepsilon$-Gap-IP-DIR-$r$CYC, we create vectors having $r$ blocks, each block having the dimension $d_0$. The $((i-1) \bmod r)$-th block and the $i$-th block in the final instances of the reduction will contain vectors from each of the instances of $\varepsilon$-GapIP obtained from the instances of 2IP from $A^{(i+1) \bmod r}, A^i$ and $A^i, A^{(i+1) \bmod r}$ respectively, while the other blocks will be zero, exactly similar to the proof of Theorem C.8. This hardness result is also true when all the target inner products are the same. $\square$

Therefore, for proving the hardness of the entry-wise approximation of poly-attention based on Max-2SAT conjecture, it is sufficient to start with a hard instance of $\varepsilon$-Gap-IP-DIR-$r$CYC. Further, we prove the lower bound for poly-attention for all polynomials that are not tree polynomials (since we already know that tree polynomials have exact computational complexity $O(n^2)$). If a polynomial is not a tree polynomial, the graphical representation must contain at least one cycle.

**Lemma E.18** ($\varepsilon$-Gap-IP-DIR-$r$CYC to APAC$^{(h)}$). *For every constant $\varepsilon > 0$, every $\delta \in (0, 0.01)$, every $c, M > 0$, given an attention polynomial $h(x_1, \ldots, x_t)$ of degree $2$ having $s$ monomials, such that its graphical representation contains a cycle of size $r$, where $t, s, r$ are constants, there exist constants $C_a > 0$ and $C_b > 0$ such that if APAC$^{(h)}(2n, (r+1)c\log n, \Gamma = C_b\sqrt{\log n}, \gamma = n^{-C_a})$ (Definition A.6) with query-key matrices $Q^{(1)}, \ldots, Q^{(t)} \in [-\Gamma, \Gamma]^{2n \times (s+1)c\log n}$ and value matrices $V^{(2)}, \ldots, V^{(t)} \in \mathbb{R}^{2n \times (s+1)c\log n}$ can be solved in time $\tilde{O}(n^{\omega-\delta})$, then $\varepsilon$-Gap-IP-DIR-$r$CYC$_{n,c\log n}$ (Definition E.15) with target inner product $m = M\log n$ can also be solved in $O(n^{\omega-\delta})$ time for any constant $M$.*

*Proof.* In our final part of Theorem E.3, we reduce Max-2SAT to entry-wise approximate computation of poly-attention. We start with an instance of $\varepsilon$-Gap-IP-DIR-$r$CYC$_{n,d=c\log n}$, since we know that this is at least as hard as Max-2SAT (Theorem E.17, Lemma E.16), consisting of sets of vectors $A^1, \ldots, A^r \subseteq \{0,1\}^d$, where $A^i$ for all $i \in [t]$ has $n$ vectors $\{a_1^i, \ldots, a_n^i\}$ each. The target inner product is $M\log n$, and the constant approximation factor is $\varepsilon$ for the gap condition. This is equivalent to checking if there exists $a_{j_1}^1 \in A^1, a_{j_2}^2 \in A^2, \ldots, a_{j_r}^r \in A^r$ such that $\langle a_{j_1}^1, a_{j_2}^2 \rangle + \langle a_{j_2}^2, a_{j_3}^3 \rangle + \cdots + \langle a_{j_{r-1}}^{r-1}, a_{j_r}^r \rangle + \langle a_{j_{r-1}}^{r-1}, a_{j_1}^1 \rangle = M_0\log n$, or, due to the promise, if $\langle a_{j_1}^1, a_{j_2}^2 \rangle + \langle a_{j_2}^2, a_{j_3}^3 \rangle + \cdots + \langle a_{j_{r-1}}^{r-1}, a_{j_r}^r \rangle + \langle a_{j_{r-1}}^{r-1}, a_{j_1}^1 \rangle \leq (1 - \varepsilon_0)M_0\log n$, where $M_0 = \Theta(M), \varepsilon_0 = \Theta(\varepsilon)$.

For the graph $G$ of the polynomial, we consider a vertex $v_{t_0}$ where the cycle of length $r$ starts. If there are multiple cycles, we consider any one.

Let the cycle be of length $r$ be given by $(v_{t_0}, v_{t_0+1}), (v_{t_0+1}, v_{t_0+2}), \ldots, (v_{t_0+r-1}, v_{t_0})$, without loss of generality. When we construct the matrices $Q^{(j)}$'s, the idea is to construct the instance of $\varepsilon$-Gap-IP-DIR-$r$CYC from $v_{t_0}$ (i.e., from $Q^{(t_0)}$), and make every other query-key matrix corresponding to variables outside the cycle to be zero.

Similar to as before, we construct query-key matrices such that for all $\ell_1, \ldots, \ell_t \in [n]$,

$$h(Q^{(1)}_{\ell_1}, \ldots, Q^{(t)}_{\ell_t}) = \Lambda(\langle a^1_{\ell_{t_0}}, a^2_{\ell_{t_0+1}} \rangle + \ldots + \langle a^{r-1}_{\ell_{t_0+r-1}}, a^r_{\ell_{t_0+r}} \rangle + \langle a^r_{\ell_{t_0+r}}, a^1_{\ell_{t_0}} \rangle), \qquad (19)$$

for a scaling factor $\Lambda$.

**Constructing the matrices.** We form the matrices $Q^{(1)}, \ldots, Q^{(t)} \in [-\Gamma, \Gamma]^{\tilde{n} \times \tilde{d}}$, $\tilde{n} = 2n$, $\tilde{d} = (r+1)d$ as follows, using a scaling factor $B = \omega(1)$:

1. For $Q^{(j)}$'s, if $j < t_0$ or $j > r + t_0 - 1$, we just make the entire matrix zero $\mathbf{0}_{2n \times (r+1)d}$.

2. For defining $Q^{(t_0)}$, we define the first column block (starting of the cycle) as,

$$Q^{(t_0)}_{(1:2n,1:d)} = B \begin{bmatrix} a^1_1 \\ a^1_2 \\ \vdots \\ a^1_n \\ \mathbf{0}_d \\ \mathbf{0}_d \\ \vdots \\ \mathbf{0}_d \end{bmatrix}_{2n \times d},$$

the $r$-th column block (end of the cycle) as,

$$Q^{(t_0)}_{(1:2n,(r-1)d+1:r.d)} = B \begin{bmatrix} a^1_1 \\ a^1_2 \\ \vdots \\ a^1_n \\ \mathbf{0}_d \\ \mathbf{0}_d \\ \vdots \\ \mathbf{0}_d \end{bmatrix}_{2n \times d},$$

the final block that balances the inner product as

$$Q^{(t_0)}_{(1:2n,r.d+1:(r+1)d)} = B\mathbf{1}_{2n \times d},$$

and all the other remaining blocks as $\mathbf{0}_{2n \times d}$.

3. Now, for the matrices inside the cycle, i.e., $j \in [t_0 + 1, t_0 + r - 1]$, we define $Q^{(j)}$ as follows. For $i = j - 1, j$ (which is the traversal inside the cycle from $v_{j-1}$ to $v_j$, and $v_j$ to $v_{j-1}$ respectively), we define that block as,

$$Q^{(j)}_{(1:2n,(i-1)d+1:i.d)} = B \begin{bmatrix} a^{j-(t_0-1)}_1 \\ a^{j-(t_0-1)}_2 \\ \vdots \\ a^{j-(t_0-1)}_n \\ \mathbf{0}_d \\ \mathbf{0}_d \\ \vdots \\ \mathbf{0}_d \end{bmatrix}_{2n \times d},$$

the final block as,

$$Q^{(j)}_{(1:2n,r.d+1:(r+1)d)} = \begin{bmatrix} \mathbf{0}_{n \times d} \\ \mathbf{1}_{n \times d} \end{bmatrix}_{2n \times d},$$

and all other blocks as $\mathbf{0}_{2n \times d}$.

For the value matrices $V^{(j)} \in \mathbb{R}^{(2n) \times (r+1)d}$, $j \in [t_0, t_0 + r - 1]$, we define the first column as,

$$V^{(j)}_{(1:2n,1)} = \begin{bmatrix} \mathbf{1}_n^T \\ \mathbf{0}_n^T \end{bmatrix},$$

and for all other $j$'s, we define the first column as,

$$V^{(j)}_{(1:2n,1)} = \begin{bmatrix} \mathbf{1}_n^T \\ \mathbf{1}_n^T \end{bmatrix},$$

with every other columns $\mathbf{0}_{2n}^T$.

**Correctness of construction.** We prove that indeed Equation 19 is satisfied when $\ell_1, \ldots, \ell_t \in [n]$. When we consider $h$, all the monomials containing variables $x_j$ for $j < t_0$ or $j > t_0 + r - 1$ vanish since $Q^{(j)}$'s are zero. Whenever we have a monomial of the form $x_j x_{j+1}$, $j \in [t_0, t_0 + r - 1]$, it survives and gives $\langle a_{\ell_{(j-t_0+1)}}^{j-t_0+1}, a_{\ell_{((j-t_0+2) \bmod r)}}^{(j-t_0+2) \bmod r} \rangle$.

These query-key and value matrices can be computed in $O(n^{1+o(1)})$ time.

**Approximation yields gap property.** We again consider the entry-wise approximation of $Att^{(h)}_{\ell_1,1}$ as $x_{\ell_1}$, and we have

$$|x_{\ell_1} - \frac{P_{\ell_1}}{R_{\ell_1}}| < \gamma,$$

for

$$P_{\ell_1} = \sum_{\ell_2,\ldots,\ell_t \in [n]} \exp\left( \frac{1}{d} h(Q^{(1)}_{\ell_1}, Q^{(2)}_{\ell_2}, \ldots, Q^{(t)}_{\ell_t}) \right),$$

$$R_{\ell_1} = \sum_{\ell_2,\ldots,\ell_t \in [2n]} \exp\left( \frac{1}{d} h(Q^{(1)}_{\ell_1}, Q^{(2)}_{\ell_2}, \ldots, Q^{(t)}_{\ell_t}) \right),$$

when the $(\ell_1, 1)$-th element of $Att^{(h)}$ is $\frac{P_{\ell_1}}{R_{\ell_1}}$.

**Bounds on $P_{\ell_1}, R_{\ell_1}$.** For the lower bound on $R_{\ell_1}$, using a calculation exactly similar to that of the proof of Lemma E.8 gives us

$$n^{t_0-1} e^{\left( \frac{rB^2}{(r+1)} \right)} < R_{\ell_1} < 2^{t_0-1} n^{t_0-1} e^{\left( \frac{rB^2}{(r+1)} \right)}$$

When a positive certificate for the given $\varepsilon$-Gap-IP-DIR-$r$CYC exists, we will have some $\ell_{t_0}^0, \ell_{t_0+1}^0, \ldots, \ell_{t_0+r-1}^0 \in [n]$ for which $\langle a_{\ell_{t_0}^0}^1, a_{\ell_{t_0+1}^0}^2 \rangle + \ldots + \langle a_{\ell_{t_0+r-1}^0}^{r-1}, a_{\ell_{t_0+r}^0}^r \rangle + \langle a_{\ell_{t_0+r}^0}^r, a_{\ell_{t_0}^0}^1 \rangle = M_0 \log n$. This would give

$$P_{\ell_1} > (2n)^{t_0-r-1} e^{\left( \frac{M_0}{(r+1)c} B^2 \right)},$$

which implies

$$x_{\ell_1} > \frac{P_{\ell_1}}{R_{\ell_1}} - \gamma > \frac{e^{\left( B^2 \left( \frac{M_0}{c} - r \right)/(r+1) \right)}}{(2n)^r} - \gamma. \tag{20}$$

Otherwise, if no positive certificate for IP-Dir-$r$CYCLE exists, then

$$P_{\ell_1} < 2^{t_0-r-1} n^{t_0-1} e^{\left( (1-\varepsilon_0) B^2 \frac{M_0}{(r+1)c} \right)},$$

and therefore,

$$x_{\ell_1} < \frac{P_{\ell_1}}{R_{\ell_1}} + \gamma < 2^{t_0-r-1} e^{\left( B^2 \left( \frac{(1-\varepsilon_0) M_0}{c} - r \right)/(r+1) \right)} + \gamma. \tag{21}$$

Note that if a positive instance of $\varepsilon$-Gap-IP-DIR-$r$CYC exists, then $x_{\ell_1}$ is the greater than the lower bound (it is greater for all $\ell_1$ if $t_0 \neq 1$, otherwise we choose only that $\ell_1$ for which the $\varepsilon$-Gap-IP-DIR-$r$CYC instance contains $a_{\ell_1}^1$), otherwise always lesser than the lower bound.

**Wrapping up.** In order to maintain a gap between the cases of a positive instance of $\varepsilon$-Gap-IP-DIR-$r$CYC existing, we require the lower bound (Equation 21) must be less than the upper bound (Equation 20)

$$\frac{1}{e^{\left(\varepsilon_0 B^2 \frac{M_0}{(r+1)c}\right)}} \frac{2^{t_0-r-1}}{e^{\left(B^2\left(r-\frac{M_0}{c}\right)/(r+1)\right)}} + \gamma < \frac{1}{(2n)^r e^{\left(B^2\left(r-\frac{M_0}{c}\right)/(r+1)\right)}} - \gamma.$$

Now, there exist large enough constants $C_a, C_b > 0$ such that this inequality is satisfied for $\gamma \leq n^{-C_a}$ and $B \geq C_b\sqrt{\log n}$.

This proves that any algorithm for an entry-wise $\gamma$-approximation of $Att^{(h)}$ having maximum value of the entries $\Gamma = B = \Omega(\sqrt{\log n})$ requires time $\Omega(n^\omega)$, assuming the Max-2SAT conjecture, since if $\mathsf{APAC}^{(h)}$ could be solved in $O(n^{\omega-\delta})$ time, then that would imply that would imply Max-2SAT could be solved in $2^{(\omega/3-\Omega(\delta))n}$ time (Corollary E.12), which can not be true for an absolute constant $\delta > 0$ (Hypothesis 3). $\qquad\square$

# F  PROOFS OF SECTION 3.1: FUNCTION COMPOSITION

In this section, we describe a poly-attention mechanism whose one attention head can simulate $t$-fold function composition. In order to study the representational powers, it is important to also consider the number of bits stored for each entry for the matrices, denoted as *precision*, $p$. Since the entries are usually considered to be polynomial in $n$, it is safe to assume $p = n^{o(1)}$. Furthermore, as usual, we consider the embedding dimension $d = O(\log n)$.

Before showing the representational strength of poly-attention, we first show that Strassen-attention and 3-tensor attention cannot simulate 3-fold function composition. For this limitation result, we require a communication lower bound proved in a previous work of Chakrabarti (2007) on *myopic pointer jumping*.

**Definition F.1** (Myopic pointer jumping). *For every $t \geq 2$, myopic pointer jumping can be seen as similar to function composition, where we are interested in computing $t$-fold function composition, for inputs as functions $f_1, \ldots, f_t : [n] \to [n]$ and a value $x \in [n]$. There are $t$ players and a coordinator $C$, such that:*

- *Player 1 has as inputs $x$ and $f_2$,*

- *Player $i$ for $i \in [2 : t-1]$ have inputs $x$ and $f_1, \ldots, f_{i-1}, f_{i+1}$,*

- *Player $t$ has inputs $x$ and $f_1, \ldots, f_{t-1}$.*

*The Players $i \in [t]$ can only send messages to $C$, and the goal of the protocol is for $C$ to compute the value of $f_t(f_{t-1} \ldots f_1(x))$.*

Now, the lower bound due to Chakrabarti (2007) for myopic pointer jumping is given as below.

**Lemma F.2** ((Chakrabarti, 2007, Theorem 1)). *To solve the myopic pointer jumping problem, the players need to send at least $\Omega(n/t)$ bits to $C$ in order for $C$ to compute $f_t(f_{t-1} \ldots f_1(x))$.*

We want to study the representational strengths and limitations in terms of function composition. We say that an attention mechanism *simulates* $t$-fold function composition, if, given the input $X \in \mathbb{R}^{(tn+1)\times d}$ containing descriptions of $f_1, \ldots, f_t$ and an $x \in [n]$, the attention mechanism is able to output the value of $f_t(f_{t-1} \ldots f_1(x))$. As before, the input function $f_i$ will be given as the $i$-th block of $X$, in $X_{((i-1)n+1:i.n)}$ for all $i \in [t]$, and $x$ will be given in $X_{tn+1}$, and we want the attention mechanism to output the value of $f_t(f_{t-1} \ldots f_1(x)$ in the $(tn + 1)$-th entry of the output.

The first limitation result, Strassen-attention can not simulate 3-fold function composition is given by:

**Theorem F.3.** *One layer of Strassen-attention requires at least $H > n^{1-o(1)}$ heads to simulate 3-fold function composition.*

*Proof.* Let us consider an instance of 3-fold function composition where, given $f_1, f_2, f_3 : [n] \to [n]$, and $x \in [n]$, we want to compute $f_3(f_2(f_1(x)))$. As usual, the input $X$ contains $N = 3n + 1$ rows

of embedding dimension $d = O(\log n)$, where $X_{(1:n)}$ corresponds to the values of $f_1(1), \ldots, f_1(n)$, $X_{(n+1:2n)}$ corresponds to the values of $f_2(1), \ldots, f_2(n)$, $X_{(2n+1:3n)}$ corresponds to the values of $f_3(1), \ldots, f_3(n)$ and finally $X_{3n+1}$ corresponds to $x$.

The main idea for proving this lower bound is by assuming that Strassen-attention can simulate 3-fold function composition using $H$ heads. We are given the query-key and value matrices for $H$ Strassen-attention heads such that the output of mechanism contains the value of $f_3(f_2(f_1(x)))$. Using these, we define a communication problem which will use computations required for outputting the matrix $Att^{(S)}$, that gives the value of $f_3(f_2(f_1(x)))$. Next, we will use existing lower bounds (Lemma F.2) to contradict this statement, which would give a lower bound on the minimum number of heads of Strassen-attention required to compute $f_3(f_2(f_1(x)))$.

We now define the communication problem to capture this setting. Consider 3 players with inputs,

- Player 1 has $x, f_2$,

- Player 2 has $x, f_1, f_3$,

- Player 3 has $x, f_1, f_2$,

and a coordinator $C$. The communication channel is such that only the 3 players can send messages to the coordinator. The communication complexity is the total number of bits sent by the players to the coordinator such that the coordinator can compute the value of $f_3(f_2(f_1(x)))$.

As defined before, this communication setting is an instance of myopic pointer jumping for $t = 3$, and the lower bound from Lemma F.2 implies that at least $\Omega(n)$ bits are need to be communicated.

Now, let us assume that there exists a Strassen-attention mechanism that computes 3-fold function composition using $H$ heads, where we will denote the index of the head as a superscript $u \in [H]$. The weight matrices for query-key are $W_{Q^{(1)}}^u, W_{Q^{(2)}}^u, W_{Q^{(3)}}^u \in \mathbb{R}^{d \times d}$ and the value weights are $W_{V^{(2)}}^u, W_{V^{(3)}}^u \in \mathbb{R}^{d \times d}$ for the attention head $u \in [H]$. Let the precision of the values be $p$. These matrices and the functions computed by the first and last MLP layers are known to all the 3 players and the coordinator. Assuming that Strassen-attention can simulate 3-fold function composition, we devise a communication protocol for the above problem using the value of $Att^{(S)}$ to obtain lower bounds on $H$ using a proof inspired by works of Peng et al. (2024); Sanford et al. (2024b).

The output matrix of the $u$-th head of Strassen-attention, $Att^{(S)u}$, for $u \in [H]$, is given as

$$Att_N^{(S)u} = \frac{\sum_{j,k \in [N]} r_{j,k}^{N\,u} (X_j \, W_{V^{(2)}})^u \odot (X_k \, W_{V^{(3)}}^u)}{\sum_{j,k \in [N]} r_{j,k}^{N\,u}}, \tag{22}$$

where we have $N = 3n + 1$, which is the row of $Att^{(S)}$ where we want the value of $f_3(f_2(f_1(x)))$, and

$$r_{j,k}^{N\,u} = \exp\left(\frac{1}{d}(X_{3n+1} \, W_{Q^{(1)}}^u (W_{Q^{(2)}}^u)^T X_j^T + X_j \, W_{Q^{(2)}}^u (W_{Q^{(3)}}^u)^T X_k^T \right.$$
$$\left. + X_k \, W_{Q^{(3)}}^u (W_{Q^{(1)}}^u)^T X_{3n+1}^T)\right),$$

for all heads $u \in [H]$. The players have parts of $X$, i.e., for $f_1$ they have $X_{(1:n)}$, for $f_2$ they have $X_{(n+1:2n)}$, for $f_3$ they have $X_{(2n+1:3n)}$ and for $x$ they have $X_{3n+1}$.

The communication protocol proceeds as follows, where the player sends the values for each Strassen-attention head $u \in [H]$:

1. Player 1 sends $\widehat{L_1}^u$ and $\widehat{L_1'}^u$, where $\widehat{L_1}^u$ is an $O(p \log \log n)$-bit approximation of the binary expression of $L_1^u$, and $\widehat{L_1'}^u$ is an $O(p \log \log n)$-bit approximation of the binary expression of $L_1'^u$, where

$$L_1^u := \sum_{\substack{j \in S_1, k \in S_2 \\ S_1, S_2 \in \{\{3n+1\}, [n+1:2n]\}}} r_{j,k}^{N\,u},$$

and

$$L_1'^u := \frac{1}{L_1^u}\Bigg(\sum_{\substack{j\in S_1, k\in S_2 \\ S_1,S_2\in\{\{3n+1\},[n+1:2n]\}}} r_{j,k}^{N\,u}(X_j W_{V^{(2)}}{}^u)\odot(X_k W_{V^{(3)}}{}^u)\Bigg),$$

for all $u\in[H]$, to $C$.

2. Player 2 sends $\widehat{L_2}^u$ and $\widehat{L_2'}^u$, where $\widehat{L_2}^u$ is an $O(p\log\log n)$-bit approximation of the binary expression of $L_2^u$, and $\widehat{L_2'}^u$ is an $O(p\log\log n)$-bit approximation of the binary expression of $L_2'^u$, where

$$L_2^u := \sum_{\substack{j\in S_1, k\in S_2 \\ S_1,S_2\in\{\{3n+1\},[n],[2n+1:3n]\} \\ (S_1,S_2)\neq(\{3n+1\},\{3n+1\})}} r_{j,k}^{N\,u},$$

and

$$L_2'^u := \frac{1}{L_2^u}\Bigg(\sum_{\substack{j\in S_1, k\in S_2 \\ S_1,S_2\in\{\{3n+1\},[n],[2n+1:3n]\} \\ (S_1,S_2)\neq(\{3n+1\},\{3n+1\})}} r_{j,k}^{N\,u}(X_j W_{V^{(2)}}{}^u)\odot(X_k W_{V^{(3)}}{}^u)\Bigg),$$

for all $u\in[H]$, to $C$.

3. Player 3 sends $\widehat{L_3}^u$ and $\widehat{L_3'}^u$, where $\widehat{L_3}^u$ is an $O(p\log\log n)$-bit approximation of the binary expression of $L_3^u$, and $\widehat{L_3'}^u$ is an $O(p\log\log n)$-bit approximation of the binary expression of $L_3'^u$, where

$$L_3^u := \sum_{\substack{j\in S_1, k\in S_2 \\ S_1,S_2\in\{[n],[n+1:2n]\} \\ S_1\neq S_2}} r_{j,k}^{N\,u},$$

and

$$L_3'^u := \frac{1}{L_3^u}\Bigg(\sum_{\substack{j\in S_1, k\in S_2 \\ S_1,S_2\in\{[n],[n+1:2n]\} \\ S_1\neq S_2}} r_{j,k}^{N\,u}(X_j W_{V^{(2)}}{}^u)\odot(X_k W_{V^{(3)}}{}^u)\Bigg),$$

for all $u\in[H]$, to $C$.

4. $C$ computes

$$\frac{\sum_{i\in[3]}\widehat{L_i'}^u.\widehat{L_i}^u}{\sum_{i\in[3]}\widehat{L_i}^u}\in\mathbb{R}^d, \tag{23}$$

as the $N$-th row of the $Att^{(S)u}$ matrix.

Note that Equation 23 is the correct value of the approximation of $Att_N^{(S)u}$, for all $u\in[H]$, since the values of $L_N^u, L_N'^u$ are simply the partial sums, all of which amount to Equation 22 with the given bounds on each of the summations. Sanford et al. (2024b) showed that using $O(p\log\log n)$ bits of precision is sufficient in this approximation, and this gives us the correct value of $f_3(f_2(f_1(x)))$ upto $p$ bits of precision. The number of bits communicated is equal to $O(dpH\log\log n)$, and using the lower bound from Lemma F.2, we must have $dpH > \Omega(n/(\log\log n))$. Since we usually choose $d = O(\log n), p = n^{o(1)}$, we must have, the number of heads, $H > n^{1-o(1)}$. $\qquad\square$

**Corollary F.4.** *One layer of 3-tensor attention requires at least $n^{1-o(1)}$ heads to simulate 3-fold function composition.*

*Proof.* The proof is very similar to that of Theorem F.3, where again we have 3 players and a coordinator in a myopic pointer jumping instance. Using the construction of 3-tensor attention, we can again infer that the communication complexity will be $O(dpH\log\log N)$, which needs to be greater than $\Omega(n)$ from Lemma F.2. This gives our result. $\qquad\square$

In fact, we can show a stronger result.

**Theorem F.5.** *If $h$ can be written as a variable separable polynomial, where each branch (see Definition A.5) has $\leq t_0$ variables, then one layer of poly-attention for $h$ requires at least $H > n^{1-o(1)}$ heads to solve $t_0$-fold function composition.*

*Proof.* We use the same proof as of Theorem F.3, by constructing a communication protocol for $t_0$-fold function composition if poly-attention for $h$ can solve it, and using the lower bound result of Lemma F.2. The input $X$ contains $N = t_0 n + 1$ tokens, and we want the output to be in the last row of $Att^{(h)u}$ for each head $u \in [H]$.

We define a communication problem again as that of myopic pointer jumping, with $t_0$ players and a coordinator $C$ who wants to compute $f_{t_0}(f_{t_0-1} \ldots f_1(x))$ (Definition F.1). Since $t_0$ is constant, Lemma F.2 states that this requires $\Omega(n)$ bits of communication.

Now, we develop a communication protocol for function composition using the $Att^{(h)u}$ matrices, $\forall u \in [H]$, which will have a communication complexity of $O(Hdp \log \log N)$. In computing the output of the poly-attention mechanism at the last row of $Att^{(h)u}$, we have the numerator term as

$$\sum_{\ell_2,\ldots,\ell_{t_0}\in[N]} \exp(h(Q_N^{(1)u}, Q_{\ell_2}^{(2)u}, \ldots, Q_{\ell_{t_0}}^{(t_0)u}))V_{\ell_2}^{(2)u} \odot \ldots \odot V_{\ell_{t_0}}^{(t_0)u},$$

and the denominator term as

$$\sum_{\ell_2,\ldots,\ell_{t_0}\in[N]} \exp(h(Q_N^{(1)u}, Q_{\ell_2}^{(2)u}, \ldots, Q_{\ell_{t_0}}^{(t_0)u})).$$

If the polynomial $h$ is variable separable and has $r$ branches, where each branch is given by the polynomial $g_i(x_1, \bar{x}^i)$ having $\leq t_0$ variables each, i.e., $h(x_1, \ldots, x_t) = \sum_{i\in[r]} g_i(x_1, \bar{x}^i)$, then players devise a protocol to separately compute the $(t_0 n + 1)$-th row of $Att^{(g_i)}$ for all $i \in [r]$. Similar to the proof of Theorem F.3, the summation of $\ell_2, \ldots, \ell_{t_0} \in [N]$ will be broken down to partial summations, which correspond to computations performed from the inputs of each player.

In computing the poly-attention output of each branch (both numerator and denominator as in the proof of Theorem F.3), let the corresponding variables of that branch be $x_{r_1}, \ldots, x_{r_{t_0}}$. Now, Player 1 would send the summations of $\ell_{r_1}, \ldots, \ell_{r_{t_0}} \in [n+1:2n] \cup \{t_0 n + 1\}$, Player 2 would send the summations over $\ell_{r_1}, \ldots, \ell_{r_{t_0}} \in [n] \cup [2n+1:3n] \cup \{t_0 n + 1\}$ except the tuples that have already been sent, and so on until Player $i$ would send the summations over $\ell_{r_1}, \ldots, \ell_{r_{t_0}} \in [(i-1)n+1] \cup [i.n+1:(i+1)n] \cup \{t_0 n + 1\}$ except the tuples that have already been sent. Since there are $t_0 - 1$ variables that are not fixed ($\ell_1$ is fixed to $N$) and all the $t_0$ players with their given inputs completely cover the summation required in the softmax computation of $Att^{(h)}$.

In this way, the players can communicate $O(Hdp \log \log N)$ bits as before to compute the value of $Att^{(g_i)u}_N$ for all $i \in [r]$ and $u \in [H]$, and given the poly-attention outputs for all these branching polynomials, the coordinator can compute the value of $Att^{(h)}$ using Lemma D.1.

Therefore, with a total of $O(Hdp \log \log N)$ bits (since the number of branches, $r$, of the polynomial $h$ is constant), the coordinator will be able to solve $t_0$-fold function composition. By Lemma F.2, $Hdp \log \log N \geq \Omega(n)$, and considering $d = O(\log n), p = n^{o(1)}$, we require $H > n^{1-o(1)}$. $\qquad \square$

Next we prove that a certain class of tree-attention, given by polynomials of the form $h_t(x_1, \ldots, x_{t+1}) = x_1 x_2 + x_2 x_3 + \ldots + x_t x_{t+1}$ can simulate $t$-fold function composition. This proves Theorem 3.4, which is also the generalization of Theorem 3.1.

**Theorem F.6.** *For every integer $t \geq 2$, poly-attention for the polynomial*

$$h_t(x_1, \ldots, x_t) = x_1 x_2 + x_2 x_3 + \ldots + x_t x_{t+1}$$

*can simulate $t$-fold function composition using one poly-attention head.*

*Proof.* For solving the problem of $t$-fold function composition, we consider the $t$ functions $f_1, \ldots, f_t : [n] \to [n]$. The input (before the first MLP layer) is a sequence of numbers $\phi(1), \ldots, \phi(tn+1) \in [n]$,

such that for $\ell \in [n]$, $j \in [t]$, we have $\phi(\ell + (j-1)t) = f_j(\ell)$, and finally $\phi(3n+1) = x$. Our task is to compute the value of $f_t(f_{t-1} \ldots f_1(x))$, and we give a construction of the MLPs, the query-key weights and the value weights of poly-attention for $h_t$, such that this Transformer layer can compute the same using only one head. We adopt the construction of Kozachinskiy et al. (2025) due to its simplicity, and use it to define the parameters of poly-attention.

We define the first MLP layer such that its output, i.e., the positional encoding of the $i$-th entry of the input to poly-attention, is given by:

$$X_i = \begin{bmatrix} 1 & i & i^2 & \phi(i) & (\phi(i))^2 & \mathbf{0}_{3k-5} \end{bmatrix}_{1 \times 3k},$$

for $i \in [tn+1]$. Here, a precision of $p = \Theta(\log n)$ can be used. Next, we construct the weight matrices $W_{Q^{(1)}}, \ldots, W_{Q^{(t)}}$.

Our goal is to create a them such that

$$h_t(Q_{\ell_1}^{(1)}, \ldots, Q_{\ell_{t+1}}^{(t+1)}) = -A^2 \log n \Big( (\phi(\ell_1) - \ell_2)^2 + (\ell_3 - n - \phi(\ell_2))^2$$

$$+ (\ell_4 - 2n - \phi(\ell_3))^2 + \ldots + (\ell_{t+1} - (t-1)n - \phi(\ell_t))^2 \Big),$$
(24)

for a constant $A > 1$. For $\ell_1 = tn + 1$, this is maximized when

$$\ell_2 = \phi(\ell_1) = \phi(tn+1) = x,$$
$$\ell_3 = n + \phi(\ell_2) = n + \phi(x) = n + f_1(x) = f_2(f_1(x)),$$
$$\vdots$$
$$\ell_{t+1} = (t-1)n + \phi(\ell_{t-1}) = (t-1)n + f_{t-1}(f_{t-2} \ldots f_1(x)) = f_t(f_{t-1} \ldots f_1(x)),$$

which is precisely our required value.

For constructing $Q^{(j)} \in \mathbb{R}^{n \times 3t}$ for $j \in [t+1]$, with such properties, we can define each row as:

1. for $i = 1$:

$$Q_\ell^{(1)} = A\sqrt{\log n} \begin{bmatrix} \phi(\ell)^2 \\ \phi(\ell) \\ 1 \\ \mathbf{0}_{3t-3}^T \end{bmatrix}_{3k \times 1}^T,$$

2. for $j \geq 2$:

$$Q_\ell^{(j)} = A\sqrt{\log n} \begin{bmatrix} \mathbf{0}_{3(j-2)} \\ -1 \\ 2(\ell - (j-2)n) \\ -(\ell - (j-2)n)^2 \\ \phi(\ell)^2 \\ \phi(\ell) \\ 1 \\ \mathbf{0}_{3(t-j)}^T \end{bmatrix}_{3k \times 1}^T,$$

for all $\ell \in [n]$.

Note that, for any $j \in [t]$,

$$\langle Q_{\ell_j}^{(j)}, Q_{\ell_{j+1}}^{(j+1)} \rangle = -A^2 \log n (\ell_{j+1} - n - \phi(\ell_j))^2,$$

which is consistent with Equation 24. While computing the softmax entries for $\ell_1 = tn + 1$, the value of $h_t(Q_{tn+1}^{(1)}, Q_{\ell_2}^{(2)}, \ldots, Q_{\ell_{t+1}}^{(t+1)})$ for all $\ell_2, \ldots, \ell_{t+1}$ that do not maximize this value, will be a factor of $n^{-A}$ less than the maximum value. Since while computing softmax, we take a sum over all $\ell_2, \ldots, \ell_{t+1} \in [tn+1]$, as long as we choose $A > \Omega(\sqrt{t})$, the maximum value will be obtained in the correct setting of $\ell_j$'s.

For outputting the value, we set the first column of all the $V^{(j)}$'s, $j \in [2:t]$, as ones, and the rest as zeros; and for $V^{(t+1)}$, we define the first column as $V^{(t+1)}_{\ell,1} = \ell$, for all $\ell \in [tn+1]$, and the rest as zeros. The error in the final output will be $n^{t-A^2}$, and as long as this is less than the number of bits of precision, we have the correct output. $\qquad\square$

As we will see, even though poly-attention for $h_t$ will be able to solve $t$-fold function composition, the previous theorem, Theorem F.5, shows that not only poly-attention for $h_{t-1}$ can not simulate $t$-fold function composition, but neither can the poly-attention for the polynomial $h(x_1, \ldots, x_{t+2}) = x_1 x_2 + x_2 x_3 + \ldots x_{t-1} x_t + x_1 x_{t+1} x_{t+2}$, which is a polynomial in $t+2$ variables!

**Remark 2.** *From Theorem F.6, we saw that poly-attention for $h_2(x_1, x_2, x_3) = x_1 x_2 + x_2 x_3$ can simulate 3-fold function composition just as Strassen-attention. Again, Strassen-attention is poly-attention for the polynomial $h(x_1, x_2, x_3) = x_1 x_2 + x_2 x_3 + x_3 x_1$, which is just one monomial different from $h_2$. However, even though they might seem similar, the cost of this one monomial is huge– $Att^{(h_2)}$ can be computed in $\tilde{O}(n^2)$ time, while computing $Att^{(S)}$ requires at least $\Omega(n^\omega)$ time.*

# G   PROOFS OF SECTION 3.4: POLYNOMIAL ROOT-FINDING

In this final section of the proofs, we prove the strong characterization of representational strength of poly-attention introduced in Section 3.4. We show this by giving a construction of the weight matrices of a poly-attention mechanism which solves polynomial root-finding (Theorem 3.7).

In this problem of polynomial root-finding, for a fixed polynomial $p(x_1, \ldots, x_t)$ and given as input a set $S \subseteq \mathbb{R}^n$, we are interested in finding if there are elements $y_1, \ldots, y_t \in S$ such that $p(y_1, \ldots, y_t) = 0$. For the output, if $y_1^0, \ldots, y_t^0$ is a root of $p$ and $S[j] = y_1^0$, then in the row $j$ of the output, we want to output an encoding of that root.

**Theorem G.1** (Polynomial root-finding). *For a polynomial $p(x_1, \ldots, x_t)$ of degree $k_0$, and given an input $S \subseteq \mathbb{R}^n$, for any integers $k, s$ if a polynomial $h(x_1, \ldots, x_t)$ of degree $k$ and sparsity $s$ is such that all the monomials of the polynomial $p^2$ divide at least some degree $k$ monomial of $h$, then poly-attention for $h$ with 2 attention heads can perform polynomial root-finding for $p$ with the input.*

*Proof.* We give a construction of the MLP layers, query-key weights and the value weights such that the Transformer can find a root of the polynomial from $S^t$, and output it. First, given $S$, considering $s_0$ as the sparsity of $p^2$, we set the embedding dimension as $d = s_0.s$. For the input $X \in \mathbb{R}^{n \times (s_0.s)}$, let the embedding of $X_i$ after the first MLP layer be

$$X_i = \begin{bmatrix} 1 & y_i & y_i^2 & \ldots & y_i^{2k_0} & \mathbf{0}_{s_0.s-2k_0-1} \end{bmatrix}_{1 \times (s_0.s)},$$

where we require $s_0.s > 2k_0 + 1$.

**Construction of first head.**   Now, our goal is to define the weight matrices such that after computing the query-key matrices $Q^{(1)}, \ldots, Q^{(t)}$, the value of $h(Q^{(1)}_{\ell_1}, \ldots, Q^{(t)}_{\ell_t})$ will yield $-n^2 p(y_{\ell_1}, \ldots, y_{\ell_t})^2$ where $y_i = S[i]$, and $\ell_1, \ldots, \ell_t \in [n]$.

Choose a $h(x_1, \ldots, x_t)$ of degree $k$ (where $k$ is a number greater than the maximum number of variables in each monomial of $p^2$), and is of any sparsity $s$ (satisfying $s_0.s > 2k_0$), where each monomial of $p^2$ divides at least some degree $k$ monomial of $h$. We assign each of these monomials of $p^2$ to exactly one degree $k$ monomial $m_i$ of $h$ for $i \in [s]$, and we associate a set $T_i$ which stores all the monomials of $p^2$ that are assigned to this monomial $m_i$ of $h$.

Now, define $Q^{(1)}, \ldots, Q^{(t)} \in \mathbb{R}^{n \times (s_0.s)}$, where each column block is of size $s_0$, as:

1. For the $i$-th column block, where for each column $j \in [s_0]$ of the block, we consider the exponents of the variables of $p$ such that $h(Q^{(1)}_{\ell_1}, \ldots, Q^{(t)}_{\ell_t})$ will give evaluations of the $j$-th monomial of $-p^2$ at $(y_{\ell_1}, \ldots, y_{\ell_t})$, for all $\ell_1, \ldots, \ell_t \in [n]$. For these values of $i, j$, we will simply denote these terms as the monomial corresponding to this column $(i-1)s_0 + j$.

(a) If the $j$-th monomial of $p^2$, for $j \in [s_0]$, consists of $k_j$ variables, and is in $T_i$ for some $i \in [s]$, let $C_j x_{r_1}^{d_{r_1}} \dots x_{r_{k_j}}^{d_{r_{k_j}}}$ be this monomial where $x_{r_1}$ is the highest preference variable. Then, we define the $j$-th column of the $i$-th column block of $Q^{(r_1)}$ as

$$Q^{(r_1)}_{1:n,(i-1)s_0+j} := n \begin{bmatrix} -C_j y_1^{d_{r_1}} \\ \vdots \\ -C_j y_n^{d_{r_1}} \end{bmatrix},$$

and for $1 < q \le k_j$,

$$Q^{(r_q)}_{1:n,(i-1)s_0+j} := n \begin{bmatrix} y_1^{d_{r_q}} \\ \vdots \\ y_n^{d_{r_q}} \end{bmatrix}.$$

For all $r \in [t]$ such that $x_r$ is a variable of $m_i$ and the $j$-th monomial of $p^2$ does not contain $x_r$ but is present in $T_i$, we define

$$Q^{(r)}_{1:n,(i-1)s_0+j} := n.\mathbf{1}_n,$$

and otherwise, if $x_r$ is not present in $m_i$

$$Q^{(r)}_{1:n,(i-1)s_0+j} := \mathbf{0}_n.$$

(b) If the $j$-th monomial of $p^2$, for $j \in [s_0]$, is not in $T_i$, then we define

$$Q^{(r)}_{(1:n,(i-1)s_0+j)} = \mathbf{0}_n,$$

for all $r \in [t]$.

2. Fixing an $i$ such that $m_i$ is of degree $\le k$, we define the query-key matrices as before, to cancel out the terms which were defined in the degree $k$. Each degree $k$ term had $s_0$ terms which could lead to non-zero values, and now for the block $i$, corresponding to the monomial $i$, the $r$-th column in that block will cancel out the $j$-th columns of each block obtained from the degree $k$-terms, for $j \in [s_0]$.

Let $s_i^j$ be the integer which is the number of occurrences of $j$-th monomial of $p^2$ while computing the monomial containing variables $x_1, \dots, x_t$ corresponding to $m_i(Q_{\ell_1}^{(1)}, \dots, Q_{\ell_t}^{(t)})$, when we consider each of the monomials $m_\ell$ ordered higher preference than $i$, (i.e., $\ell < i$), and is divisible by $m_i$.

As before, $s_i^j$ is the sum defined by adding:

- $-C_j$ whenever $\ell < i$, $m_i$ divides $m_\ell$, degree of $m_\ell$ is exactly $k$, and the highest priority variable of $m_\ell$ is present in $m_i$.
- $-s_\ell^j$ whenever $\ell < i$, $m_i$ divides $m_\ell$, degree of $m_\ell$ is less than $k$, and the highest priority variable of $m_\ell$ is also present in $m_i$.
- $-1$ otherwise when the above conditions are not met but $m_\ell$ divides $m_i$.
- $0$ in all other cases.

For every $j \in [s_0]$, if $x_r$ is not present in monomial $j$ of $p^2$ for $r \in [t]$, we just set

$$Q^{(r)}_{(1:n,(i-1)s_0+j)} := \mathbf{0}_n,$$

otherwise, for the highest preference $x_{r_1}$ variable of the $j$-th monomial of $p^2$, we define:

$$Q^{(r_1)}_{(1:n,(i-1)s_0+j)} = n \begin{bmatrix} s_i^j y_1^{d_{r_1}} \\ s_i^j y_2^{d_{r_1}} \\ \vdots \\ s_i y_n^{d_{r_1}} \end{bmatrix}_{n \times s_0.s},$$

and for all other $r$ such that $x_r$ divides this monomial,

$$Q^{(r)}_{(1:n,(i-1)s_0+j)} = n \begin{bmatrix} y_1^{d_r} \\ y_2^{d_r} \\ \vdots \\ y_n^{d_r} \end{bmatrix}_{n \times s_0.s}.$$

Notice that in these constructions, we have only used linear combinations of $y_r^q$'s for $r \in [t]$ and $q \in [2k_0]$. Therefore, weight matrices $W_{Q^{(r)}} \in \mathbb{R}^{(s_0.s) \times (s_0.s)}$ exist for every fixed polynomial $p$ such that

$$\underbrace{\begin{bmatrix} 1 & y_1^1 & \cdots & y_1^{2k_0} & 0 & \cdots & 0 \\ 1 & y_2^1 & \cdots & y_2^{2k_0} & 0 & \cdots & 0 \\ \vdots & & \vdots & & & \vdots \\ 1 & y_n^1 & \cdots & y_n^{2k_0} & 0 & \cdots & 0 \end{bmatrix}}_{s_0.s}^T W_{Q^{(r)}}$$

yield the required $Q^{(r)}$'s. For defining the value matrices, for the first $t$ coordinates, the $r$-th coordinate of $V^{(r)}$, $r \in [2:t]$ stores the corresponding value of $x_r$, and all the other entries are of the coordinates in $[2:t] \backslash \{r\}$ are one, and the first coordinate is zero. More specifically, we define

$$V^{(r)} = \begin{bmatrix} 0 & 1 & \cdots & 1 & y_1 & 1 & \cdots & 1 & 0 & \cdots & 0 \\ 0 & 1 & \cdots & 1 & y_2 & 1 & \cdots & 1 & 0 & \cdots & 0 \\ \vdots & & & & \vdots & & & & & & \vdots \\ 0 & 1 & \cdots & 1 & y_n & 1 & \cdots & 1 & 0 & \cdots & 0 \end{bmatrix},$$

where the $r$-th column has the values of the $y_i$'s.

Using the construction defined above, we have $h(Q^{(1)}_{\ell_1}, \ldots, Q^{(t)}_{\ell_t}) = -n^k p^2(y_{\ell_1}, \ldots, y_{\ell_t})$ since the degree $k$ monomials of $h$ are what contribute to $-p^2(y_{\ell_1}, \ldots, y_{\ell_t})$ from the corresponding column blocks. Inside each of these column blocks corresponding to degree $k$ monomials of $h$, the $j$-th column for $j \in [s_0]$ gives the value of the $j$-th monomial of $-p^2$ at $(y_{\ell_1}, \ldots, y_{\ell_t})$. Due to our construction, all the values of $m_i(Q^{(1)}_{\ell_1}, \ldots, Q^{(t)}_{\ell_t})$ are zeros when $m_i$'s are of degree $< k$, which finally gives us the required result.

Now, for each fixed $\ell_1$, the value of $h(Q^{(1)}_{\ell_1}, \ldots, Q^{(t)}_{\ell_t}) = -p^2(y_{\ell_1}, \ldots, y_{\ell_t})$ which is maximized for some indices $\ell_2^0, \ldots, \ell_t^0$, is at least $e^{n^2}$ factor larger than all the other values in the summation $\sum_{\ell_2, \ldots, \ell_t \in [n]} e^{h(Q^{(1)}_{\ell_1}, \ldots, Q^{(t)}_{\ell_t})}$. With the given construction of $V^{(r)}$'s, the values of $y_{\ell_2^0}, \ldots, y_{\ell_t^0}$ for which $-p^2(y_{\ell_1}, *)$ is maximized, will be present in the first $t$ coordinates of the output $Att^{(h)}_{\ell_i}$.

**Construction of second head.** Finally, we need to verify that if there exists some $\ell_1^0$ such that the values of $x_2, \ldots, x_t$ encoded in $Att^{(h)}_{\ell_1^0}$ indeed is a root of the polynomial. For this, we need the value of $y_{\ell_1}$'s for each of the $\ell_1$-th coordinate, and we incorporate this by using a second attention-head, whose output matrix contains the vector $[y_1 \quad \cdots \quad y_n]^T$ in the first column and all zeros elsewhere.

Therefore, when we add the two attention heads, the $\ell_1$-th row will contain the values of $(y_{\ell_1}, \ldots, y_{\ell_t})$ which maximizes the value of $-p^2(y_{\ell_1}, *)$. Finally, we can check using the output MLP layer if indeed the value is a root of the polynomial. $\qquad \square$

# H   EXPERIMENTAL DETAILS

## H.1   FUNCTION COMPOSITION

In this section, we explain the experimental setup behind Figure 2. We train a transformers that uses self-attention for one layer, a transformer that uses self-attention for two layers, as well as a transformer that uses tree-attention for one layer, for the attention polynomial $h(x_1, x_2, x_3) =$

$x_1 x_2 + x_2 x_3$. (This is the polynomial from Theorem 3.1 above.) We infer from the experimental findings that tree-attention is better: it is faster, more learnable, and uses less space compared to its representational counterpart, the two layer self-attention.

In the remainder of this subsection, we first explain the details behind Figure 2, which shows that despite having less trainable parameters than two layer self-attention tree-attention is more learnable. Note that two layer self-attention requires two query matrices, two key matrices, and two MLP layers, while tree-attention requires only three query-key matrices and one MLP layer. Second, we show that the time to compute tree-attention is comparable to the runtime to compute two-layer self-attention.

**Problem set-up.** We solve the task of function composition, where, for an integer $n$, given two functions $f_1, f_2 : [n] \rightarrow [n]$ and a value $x \in [n]$, we are interested in computing the value of $f_1(f_2(x))$.

We know that a two layer transformer using self-attention can solve function composition but one layer can not Peng et al. (2024), and we further proved in Theorem 3.1 that tree attention can solve it as well. We show that these theoretical results are in line with practice, where transformers with two layer self-attention as well as transformers with one layer tree-attention can both solve 0-function composition for $n = 25$ (which means the number of tokens is $2n + 1 = 51$).

**Input generation.** As described above, we train the transformers to learn $f_1(f_2(x))$ where $f_1, f_2 : [n] \rightarrow [n]$, and $x \in [n]$, for $n = 25$. The functions $f_1, f_2$, and the value $x$, are generated uniformly at random from the set $[n]$ for each batch in each epoch.

With these $f_1, f_2$ and $x$, the input tokens to the transformers are given by a sequence of $2n + 1$ tuples, $(i, f_1(i))$ for $i \in [n]$, followed by $(i, f_2(i - n))$ for $i \in [n + 1 : 2n]$ and finally $(2n + 1, x)$.

We define separate embedding tables for each of the index (first element of the tuple) and the value of the function (second element of the tuple), given by embedding matrices $\mathbb{E}_1 \in \mathbb{R}^{(2n+1) \times d}$ and $\mathbb{E}_2 \in \mathbb{R}^{n \times d}$, where $d$ is the embedding dimension. The input to the self-attention for the token $(i, j)$ is then given as $\mathbb{E}_1(i) + \mathbb{E}_2(j)$.

**Architecture details.** We choose a sequence length of 51. The transformer has an embedding dimension $d = 32$, number of heads $H = 4$, followed by an MLP layer which uses ReLU activation with one hidden layer of size 128. We also use the standard sinusoidal positional encoding from Vaswani et al. (2017), given by

$$PE_{i,2j} = \sin\left(\frac{i}{10000^{2j/d}}\right),$$
$$PE_{i,2j+1} = \cos\left(\frac{i}{10000^{2j/d}}\right),$$

, for $i \in [n]$, $j \in \{0, \ldots, d/2\}$, which is added to the $i$-th token.

**Training details.** For learning, we use a batch size of 64, a learning rate of 0.001 and train the model using an Adam optimizer. The model is trained for $100,000$ epochs on a 2024 Apple Macbook Air with an M3 Chip, and the evaluations have been shown in Figure 2 and Figure 3.

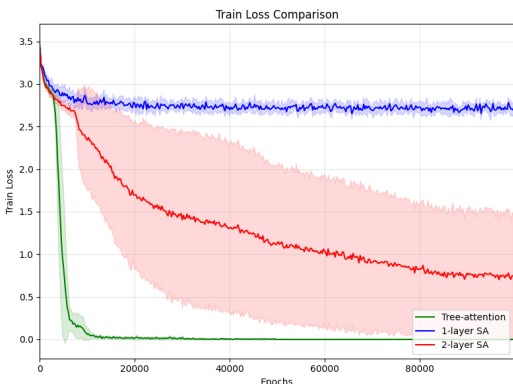

Figure 3: Training loss per epoch, averaged over 10 seeds, for learning $f_1(f_2(x))$ for sequence length 51, on a single layer of tree-attention, one layer self-attention and two layer self-attention. Tree-attention learns faster and has less fluctuations.

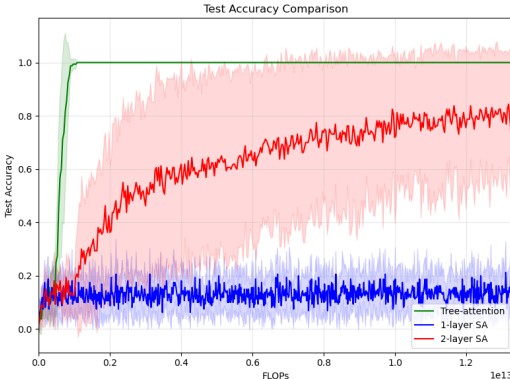

Figure 4: Accuracy per FLOP, averaged over 10 seeds, for tree-attention, 1-layer self-attention and 2-layer self-attention for learning function composition. Notice that tree-attention learns more efficiently and the learning is stable.

**Observed inference running time.** We plot the running time of computing various attention schemes for sequence lengths in $\{20, 50, 100\}$. We use vocabulary size $v = 32$, embedding dimension $d = 64$, number of heads $H = 4$, and hidden layer width 256 for the transformers, and evaluate it on a batch of size $B = 64$.

With this architecture, we randomly choose query, key and value weights in $\mathbb{R}^{d \times d}$, and random weights and biases for the MLP layer. Then we randomly generate 1000 sets of inputs $X \in \mathbb{R}^{B \times n \times v}$ and compute the running time of the attention mechanisms. The average running time has been depicted in the following table.

| Seq len | 1-layer SA (ms) | 2-layer SA (ms) | 1-layer tree (ms) | 1-layer 3-tensor (ms) | 1-layer Strassen (ms) |
|---|---|---|---|---|---|
| 20 | $1.076 \pm 0.057$ | $1.775 \pm 0.085$ | $1.367 \pm 0.057$ | $1.442 \pm 0.062$ | $1.593 \pm 0.086$ |
| 50 | $1.079 \pm 0.048$ | $1.757 \pm 0.060$ | $1.363 \pm 0.055$ | $2.911 \pm 0.044$ | $1.594 \pm 0.088$ |
| 100 | $1.080 \pm 0.048$ | $1.781 \pm 0.097$ | $1.374 \pm 0.060$ | $13.813 \pm 0.051$ | $3.395 \pm 0.081$ |

Figure 5: Average running time of various attention schemes implemented on NVIDIA A100 GPU. Tree-attention performs as fast as self-attention, implying that hidden constants in the time complexity computations are small.

**Discussion.** We obtain the following conclusion about tree-attention from these experiments.

- One layer tree-attention can successfully learn function composition, despite having only three query-key matrices and only one MLP layer (compared to two-layer self-attention that has two query matrices, two key matrices and two MLP layers).

- One layer tree-attention exhibits better learnability for function composition than two layer self-attention as in Figure 2, since accuracy increases faster for tree-attention.

- Tree-attention has an efficient inference time. From Table 5, we infer that it has a running time comparable to self-attention, and in our cases, even outperforms two layer self-attention.

## H.2 COGS DATASET

To give further evidence of the practical advantage of tree-attention, we evaluate two simple models (one with self-attention and one with tree-attention) on a benchmark NLP task, the COGS dataset Kim & Linzen (2020). We compare the two models, which differ only in which attention mechanism they use, and evaluate the difference.

COGS is a dataset which challenges the model to perform a composition based task, in which it must parse sentences into fragments, and understand the relationships of the different fragments throughout the sentence. In our experiment, the words of the sentences, along with special characters, are input to the transformer after encoding by a pre-trained tokenizer, and the error is computed on the output which is expected to be a semantically parsed sentence.

**Example 1.** *Input: A melon was given to a girl by the guard .*
**Target:** *\* guard ( x _ 9 ) ; melon ( x _ 1 ) AND give . theme ( x _ 3 , x _ 1 ) AND give . recipient ( x _ 3 , x _ 6 ) AND give . agent ( x _ 3 , x _ 9 ) AND girl ( x _ 6 )*

*From the above example, the transformer is supposed to figure out that 'guard' is the subject, and is present at position 9 of the sentence, where indexing starts from 0. The other nouns are 'melon' and 'girl', in positions 1 and 6 respectively. The verb 'give', present at position 3, has logical forms given by a theme 'melon' (position 1), recipient 'girl' (position 6), and agent 'guard' (position 9).*

We trained both tree-attention and self-attention on the training set of COGS, and tested them on in-distribution test set as well as a generalization test set. Including a generalization test set is important to make sure the model is not just memorizing the distribution. We compare the exact token match accuracy for both, and infer that tree-attention performs better in the generalization set than self-attention. This gives strong evidence of the inherent ability of tree-attention to solve composition-related tasks.

Figure 6 shows the table for the final accuracy results. The learning plots have been described in Figure 7, where tree-attention almost always out-performs self-attention. The training was performed over 10 randomly chosen seeds, and we see in Figure 8, that tree-attention achieves considerable performance of around 30-50% accuracy on almost half of the seeds.

**Implementation details.** We use simple 3 layer encoder-only transformers, having embedding dimension 64 having 4 heads, and an MLP with a hidden layer of size 256. Both transformer models (using tree-attention and self-attention), were trained for 200 epochs with a batch size of 32 (755 batches per epoch) with a learning rate of 0.001, and tested on the in-distribution test set and the generalization test set. The results for in-distribution test token accuracy were similar, both giving an exact match of around 97.5%. The generalization set accuracies have been plotted as follows.

|  | Tree-attention | Self-attention |
|---|---|---|
| Generalization token accuracy | $\mathbf{0.727691 \pm 0.013486}$ | $0.723993 \pm 0.008649$ |
| Generalization exact match | $\mathbf{0.264919 \pm 0.127609}$ | $0.239024 \pm 0.087350$ |

Figure 6: Table for mean accuracies and standard deviation over 10 random seeds.

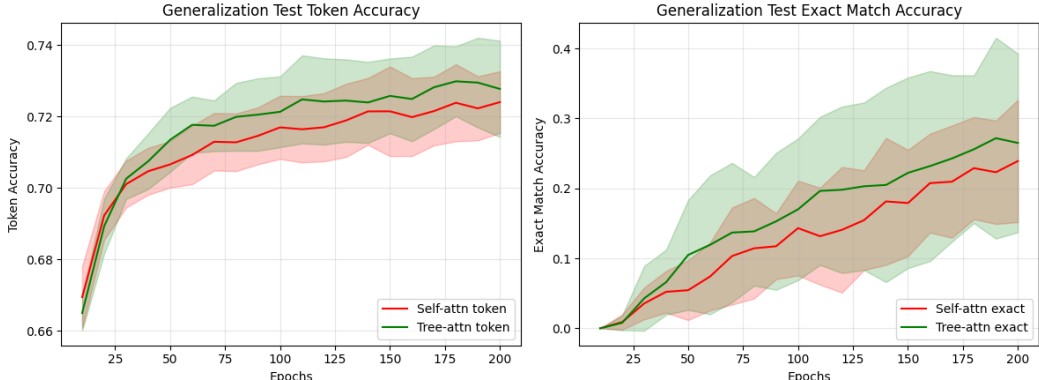

Figure 7: Plots for mean $\pm$ one standard deviation over 10 random seeds for token accuracy and exact match accuracy on the generalization set. Tree-attention has higher accuracy than self-attention.

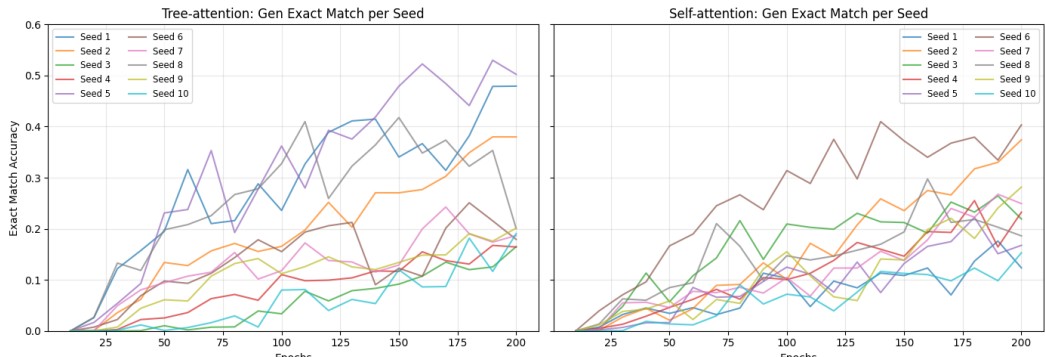

Figure 8: Exact match accuracies with each seed on the generalization set for tree-attention and self-attention. Tree-attention reaches $\sim 40\%$ accuracy for 4 out of 10 random seeds.

**Conclusion.** From the learning experiments, we infer that tree-attention is more expressive when it is used to solving composition based task. As can be inferred in Figures 7 and 8, tree-attention noticeable performance benefits for several seeds, which calls for future work to explore learning heuristics for further strengthening the results.

# I    THE USE OF LARGE LANGUAGE MODELS (LLMS)

Large language models have been used to find related works, and to polish the code for experiments.

