# OpenReview forum: "Poly-attention: a general scheme for higher-order self-attention"
_ICLR.cc/2026/Conference — ICLR 2026 Poster_

### Official Review · Reviewer_9YpC · 2025-10-28

**Soundness:** 3
**Presentation:** 2
**Contribution:** 3
**Rating:** 6
**Confidence:** 3

**Summary:**

This paper introduces **Poly-Attention**, a unifying mathematical framework that generalizes standard self-attention into a family of *higher-order attention mechanisms*. Each variant is parameterized by an “attention polynomial” defining the interaction structure among tokens. The framework subsumes existing mechanisms such as **self-attention**, **tensor-attention**, and **Strassen-attention** as special cases. The authors analyze both **expressive power** and **computational complexity**。 Overall, the work bridges representational theory and fine-grained complexity, positioning Poly-Attention as a fundamental generalization of Transformer attention mechanisms.

**Strengths:**

* Introduces a **novel formal framework** — “attention polynomials” — that unifies multiple strands of higher-order attention research under a single algebraic abstraction.
* The theoretical results are carefully stated with proofs in the appendix, covering both upper- and lower-bound analyses.
* Experimental validation, though limited, correctly demonstrates the empirical viability of Tree-Attention.

**Weaknesses:**

* Experiments are confined to small synthetic tasks (function composition on sequences of length 25). While appropriate for illustrating theory, they do not demonstrate real-world scalability or integration into large models, since transformers are always stacked in a lot of layers. So I think a multi-layer experiments in real-world scenerio (like perplexity in NLP tasks) will make great contribution to the paper.

* The paper claims tree-attention has similar runtime to self-attention, but only reports per-epoch times qualitatively. Quantitative benchmarks on memory, throughput, or gradient stability would strengthen the practical argument.

**Questions:**

See weakness

---

> ### Author Response · Authors · 2025-11-21
> **Experiment on NLP dataset**
>
> Thank you for the thoughtful review. We respond to the weaknesses below.
>
> W1. In order to show practical benefits of tree-attention, we have implemented tree-attention and self-attention on the COGS dataset [1]. COGS is a prominent NLP dataset which tests whether the model can perform compositional tasks (similar to function composition), and tree-attention performs better than self-attention on this (see Figure 7). For both tree-attention and self-attention, we use 3-layer transformers having encoding dimension = 64, heads = 4, and train it with a batch size = 32, learning rate = 0.001, dropout = 0.05 for 200 epochs. We repeat this experiment for 10 random seeds and report the mean and variance of exact match accuracy on the generalization set. The results are as follows:
>
> Tree-attention: Mean 26.5, Std dev: 12.8
>
> Self-attention: Mean: 23.9, Std dev: 8.7
>
> Furthermore, on almost half of the seeds, the tree-attention performs even better, achieving an accuracy of 30-50% (see Figure 8). We believe that the accuracy can be further boosted on such tasks with further empirical techniques, but believe that this would attract future works in this area. We have included this in the Section H.2 of the revised version of our paper.
>
> W2. We’re not sure what you mean when you say that our paper “only reports per-epoch times qualitatively”. In Figure 4 in appendix H, we give a table of the precise experimental running times of the different models we compare. More generally, much of our paper focuses on quantifying the computational complexities of poly-attention mechanisms.
>
> At any rate, we’ve also added some additional experimental results which we hope will further address your concern. Please see Section H.
>
>
> 1] Kim, N., & Linzen, T. (2020). COGS: A compositional generalization challenge based on semantic interpretation. EMNLP 2020

---

> > ### Comment · Reviewer_9YpC · 2025-11-21
> >
> > Thank you for your responses. My concerns have been addressed and I will keep my score.

---

### Official Review · Reviewer_Henj · 2025-10-30

**Soundness:** 2
**Presentation:** 2
**Contribution:** 1
**Rating:** 4
**Confidence:** 3

**Summary:**

This paper presents a significant and elegant theoretical contribution to the understanding of attention mechanisms. It introduces "poly-attention," a general framework parameterized by an "attention polynomial" $h$ that generalizes standard self-attention and other recent higher-order proposals. The authors provide a comprehensive theoretical analysis of this framework, charting the trade-offs between computational complexity and representational power. While the theoretical work seems solid, the paper is held back by a critical lack of clarity in its core definition and insufficient experimental validation to support the breadth of its claims.

**Strengths:**

1. **Generalization of attention mechanism:** The poly-attention framework itself is an elegant generalization that unifies standard self-attention, tensor attention, and Strassen attention under a single, intuitive parameter: the attention polynomial $h$.

2. **Rigorous theoretical analysis:** The paper's main strength is its rigorous analysis. The authors use tools from computational complexity theory to provide a tight characterization of the running time. The finding that tree polynomials are the only class of poly-attention computable in quadratic time is an excellent and complete theoretical result.

**Weaknesses:**

1. **Clarification on core definitions:** The paper's central contribution, the poly-attention mechanism, is presented in Definition 2.2. This definition is inconsistent, making the paper's subsequent claims impossible to verify. Please correct me if I am wrong. The attention polynomial $h$ is defined over **$t$** variables ($x_1, ..., x_t$). However, the exponent in Definition 2.2 is written as $h(Q_{l_{1}}^{(1)},...,Q_{l_{k}}^{(k)})$, incorrectly using the polynomial's *degree* **$k$** as the number of arguments.

2. **Overstatement of representational power:** The claim that poly-attention can "solve polynomial root-finding" (Theorem 3.7) is a severe overstatement. However, the proof sketch in Appendix G does not describe any learnability result. Instead, it details a complex, static encoding where the polynomial $p^2$ is explicitly hard-coded into the query-key weights. This is just construction of a specific circuit, not evidence that a model can *learn* to solve this task.

3. **Meaningless Runtime Baselines:** The paper claims tree-attention's $O(n^2)$ complexity is a practical advantage over superquadratic methods (Strassen, 3-tensor). However, the runtime table (Figure 4) omits these mechanisms entirely, comparing only against 1- and 2-layer self-attention. Without these baselines, the practical benefit of the $O(n^2)$ complexity is completely unsubstantiated.

4. **Insufficient experiments on real world datasets :** A theoretical paper can be valuable. But when a paper makes explicit claims about practicality, efficiency, and solving tasks, it has a duty to validate them empirically. This paper's experiments are so weak, and they cannot demonstrate whether the proposed method is really useful.

**Questions:**

See above.

**Details Of Ethics Concerns:**

Not need

---

> ### Author Response · Authors · 2025-11-21
> **Clarification on proofs and improved experiments**
>
> We thank the reviewer for appreciating the analysis presented in the paper to provide a general framework for higher order self-attention. Here are some clarifications:
>
> W1: Thank you for pointing out this typo. We have corrected the k to t in Definition 2.2.
>
> W2: We’re confused by your claim; all our theorem statements are fully proved in the appendix. The Theorem 3.7 you mention is a result about representational strength. It is in a section titled “Representational Strength of Poly-Attention”. Our proof in Appendix G proves the representational strength as we claim. More generally, the focus of our paper is on introducing new attention mechanisms with representational strength guarantees and which can be computed quickly.
>
> You’re right that we don’t prove learnability. In fact, more generally, there are very few learnability theorems of any kind in the theory of machine learning, and proving this appears to be far beyond the current techniques in our field. In order to address this gap, we included the experiments (briefly described in section 5, and elaborated on in appendix H) that show that tree attention is able to learn the weights for function composition, even in many fewer epochs than a comparable (vanilla) attention network with more layers!
>
> W3: We already give a comparison of the running times of the different mechanisms from this work and prior work in Figure 1 on page 3. In particular, two of our results show that tree-attention can be computed in quadratic time, and that Strassen attention cannot be computed in quadratic time. (Prior work proved this for 3-tensor attention already.) In other words, we have already mathematically proved that tree-attention can be computed faster than the other mechanisms you mention.
>
> Figure 4 which you reference is in a different part of our paper and serves a different purpose. Its goal is to show that, in our experiments comparing tree-attention with (vanilla) attention, the clock running time of tree-attention is comparable with vanilla attention. In fact, it’s much faster than two layers of vanilla attention (tree-attention needs only one layer for function composition, whereas vanilla attention needs two layers). In particular, one objective of this figure is to show that the quadratic running time of our algorithm for tree-attention isn’t hiding anything impractical in its big-O notation; it can really be implemented to run in time which compares well with vanilla attention.
>
> W4: We’re, again, confused about what you’re referring to here. Could you please point to a specific example of a claim that you think we make but don’t justify?
>
> The goal of our paper is to introduce the new family of poly-attention mechanisms, and prove theorems about their representational strength and computational complexity. In particular, we highlight the tree-attention mechanism, which is the first mechanism which can solve function composition and can be computed in quadratic time. These are both results which we prove.
>
> As we explain in section H, there could in principle be two “gaps” that are not addressed by our theorems and which we aim to address by simple experiments: (1) empirical learnability (i.e., whether one can learn the weights to solve function composition), and (2) practical efficiency (i.e., whether there are large constants hidden by our big-O notation which precludes it from being practical). Our experiment shows that neither of these is a problem: (1) one head and one layer of tree-attention learns function composition in less than half as many epochs as two layers of (vanilla) attention (see Figure 2), and (2) the actual clock running time is approximately same as that for the one-layer (vanilla) attention (see Figure 5).
>
> In fact, in response to some reviews, we’ve prepared further experiments.
>
> In order to show practical benefits of tree-attention, we have implemented tree-attention and self-attention on the COGS dataset [1]. COGS is a prominent NLP dataset which tests whether the model can perform compositional tasks (similar to function composition), and tree-attention performs better than self-attention on this (see Figure 7). For both tree-attention and self-attention, we use 3-layer transformers having encoding dimension = 64, heads = 4, and train it with a batch size = 32, learning rate = 0.001, dropout = 0.05 for 200 epochs. We repeat this experiment for 10 random seeds and report the mean and variance of exact match accuracy on the generalization set. The results are as follows:
>
> Tree-attention: Mean 26.5, Std dev: 12.8
>
> Self-attention: Mean: 23.9, Std dev: 8.7
>
> Furthermore, on almost half of the seeds, the tree-attention performs even better, achieving an accuracy of 30-50% (see Figure 8). We believe that the accuracy can be further boosted on such tasks with further empirical techniques, but believe that this would attract future works in this area. We have included this in the Section H.2 of the revised version of our paper.

---

### Official Review · Reviewer_8kRi · 2025-10-30

**Soundness:** 3
**Presentation:** 3
**Contribution:** 2
**Rating:** 4
**Confidence:** 3

**Summary:**

The authors unified some of the existing attention variants into a new attention variant (poly-attention). In addition they showed that (with methematical theorems and proofs) under some conditions poly-attention can "efficiently" represent some functional forms that self attention, higher-order attention and strassen attention cannot represent "efficiently". One such example is the function composition problem. They showed that 2-fold function composition can be represented by poly-attention with $O(n^2)$ computational complexity while exiting attention variants require "superquadratic" time complexity. At the end they also provide an experiment which shows that poly attention learns 2-fold composition problem faster (in the sense of epochs) compared to 2-layer self attention, in the same experiment thay showed that 1 layer self attention cannot learn 2-fold composition problem.

**Strengths:**

The paper proposes a "unified poly-attentin" framework that organizes several high order attention machanisms. That seems useful especially for follow up theory work. I like the the represantation ability comparisons without ignoring the computational complexities. I didnt check every proof and the proofs in detail, but the upper/lower bounds story is believable and it explains why other attention variants blow up to superquadratic. Overall exposition is quite clear, mapping the special cases early helps both theory minded and experimental readers.

**Weaknesses:**

W1)About the motivation (I feel this is crucial):
Although it can possibly achieve other functions in the paper the main motivation/achievement of poly-attention is stated around the functional composition problem (even your "punchline" is around it). However, you should explain more clearly why this particular functional form matters. What makes it a good choice? In principle, we can always cook up an unusual functional form and then design a neural network that works well for it. Because of that, we need some justification that this is not just an arbitrary example but a form that’s actually important or likely to appear in practice.

W2)About the experiments:
The experimental "validation" seems strange. All of the paper is about representational abilities of some models but the experiments are about the learning dynamics. Thus the experiments are not specifically about "validating" the theories. Seeing that the main thing about the experiment is comparing 2 layer self attention and poly-attention learning speed, they are just experiments themselves. However, that does not mean that these experiments are not relevant. Seeing that the paper does not have any theory about the learning dynamics that makes the provided experiments even more important, so I feel authors should improve them:
1)The learning curves in the figure 2 and the appendix figures seem unusual. The accuracy slightly increases at the begining stays constant for some time then abruptly increases again and the epoch number it abruptly increases the second time seems "random" at first look. Therefore I strongly suggest repeating the same experiment with multiple random seeds (this experiment should not even require much computation the networks are very small so I am sure it can be repeated very easily), and eventually provide final plot with confidence bounds.
2)Also, seeing that the main discussion in the paper is around the computational efficiency, the figure 2 should be replaced with another version in which the horizontal axis is the number of flops used instead of the epochs.

as small note, in the experimental validations instead of using words like "roughly" they can easily provide the ratio of the time it takes (even a short footnote may be okay, instead of referring to appendix).

W3) About the literature review: the hyper(feature)attention paper available at “https://arxiv.org/abs/2506.06179” appears to be closely related and should probably be discussed. That’s the first paper that comes to my mind, but there are probably other, closely related ones (with very similar self-attention variants). I just wanted to check whether you’ve already reviewed the literature and how confident you are in your coverage.

In addition to these, I discuss a fundamental possible weakness in the questions section.

**Questions:**

I feel that every poly-attention head defined in the paper can be written as a (single) t-order attention head, if we allow the tensor head to use a feature dimension that is $s$ times larger, where s is the number of monomials in the attention polynomial h. Here is a rough proof sketch

Let
$$
h(x_1,\dots,x_t) = \sum_{i=1}^s m_i(x_1,\dots,x_t)
$$
be the attention polynomial used by the poly-attention, where each monomial
$m_i$ uses some subset of the variables, e.g.
$m_i(x_1,\dots,x_t) = x_{j^i_1}\cdots x_{j^i_{k_i}}$.
For the input $X$ the poly-attention first forms the usual projections
$Q^{(j)} = X W_Q^{(j)}$ and $V^{(j)} = X W_V^{(j)}$.

Now consider a $t$-tensor attention head whose key vectors have dimension
sd (that is: s blocks, each of size d).
For each slot $j=1,\dots,t$ we build its key as follows:
for monomial $m_i$, if $x_j$ appears in $m_i$ we put the row
$Q^{(j)}_{\ell,:}$ in block $i$; if $x_j$ does not appear in $m_i$ we put
the all-ones vector in block $i$ (which can be acquired by adding biasses to the Q, K projections).
So every key is just $s$ copies of either “the real query” or “all ones.”

When the tensor attention takes the order-$t$ inner product of these keys,
the product over block $i$ multiplies exactly the variables that $m_i$
wanted, and multiplies $1$ for the variables that $m_i$ does not use.
Therefore the tensor score is

$$\sum_{i=1}^s m_i\bigl(Q^{(1)}_{\ell_1},\dots,Q^{(t)}_{\ell_t}\bigr)
= h\bigl(Q^{(1)}_{\ell_1},\dots,Q^{(t)}_{\ell_t}\bigr),$$
i.e. the same score as the poly-attention.

For the values do the same trick. Hence the tensor attention output, restricted to the first $d$ coordinates, coincides with the poly-attention output, for every input X. This shows that, up to a factor s increase in feature dimension, poly-attention does not define a strictly larger class of functions than standard t-tensor (higher-order) attention. At first sight this increase in feature dimension might look like an artificial blow-up. However, throughout the paper the authors do not argue that d must stay small; d is simply the embedding size chosen by the model designer. In that regime, allowing a constant (or even moderate) factor increase in d in order to simulate poly-attention with a standard $t$-tensor attention is not a strong penalty. On top of that, in practice we almost never need exact equality of attention scores. Since the model is trained end-to-end, it can approximate the desired score function well enough with fewer parameters than the exact tensor construction. So the sd-dimensional tensor above should be read as a worst-case. A trained model would likely get close with a smaller dimension. That makes the “poly-attention is strictly more expressive” claim weaker in a practical sense.

Shortly, for any given poly-attention head, there are parameter settings of a higher-order/tensor attention head that can reproduce it. So poly-attention is not a strict superset of all higher-order variants; it’s another way to specify the same class of interactions. Thus my question is did you carefully consider this possibility while writing the paper?

Another caveat is practical: in real Transformer setups we usually don’t know the right interaction structure in advance, so fixing a specific polynomial might be brittle. From that viewpoint the current architecture looks more theoretical, and it would be good to follow up with experiments to show when/if poly-attention actually helps in some general situations, which brings us to the weakness W2 I mentioned.

---

> ### Author Response · Authors · 2025-11-21
> **Function Composition Motivation, Additional NLP Experiments**
>
> Thank you for your feedback. Indeed, the purpose of this paper is to describe a theoretical framework for higher-order self-attention that will help both theoreticians and practitioners to develop such attention schemes.
>
> W1: We focus on function composition because of its prominence in the literature on the representational strength of attention mechanisms. (See lines 68-75 of our paper where we give some references.) For instance, the previous works (on tensor-attention and Strassen-attention) both introduced mechanisms and focused on their ability to perform function composition.
>
> In short, function composition” is basically the tiniest possible toy problem that already captures multi-step reasoning, modularity, and systematic generalization, which are all very important for a large language model. It is needed for behaviors like following instructions (“Sort the list, then remove duplicates”), multi-hop reasoning: (“From A, infer B; from B, infer C”), combining the meanings of two different parts of an input, etc. If an architecture has a fundamental limitation on representing compositions of even very simple functions, that suggests a limitation on exactly these kinds of sequential, step-by-step behavior.
>
> As another example, the prominent COGS dataset from the NLP literature exactly tests for this kind of “function composition” reasoning, as we discuss next.
>
> W2: Thank you for the suggestions; we have implemented them (as we discuss in a moment).
>
> We first want to clarify the goal of our experiments. The goal of our paper is to introduce the new family of poly-attention mechanisms, and prove theorems about their representational strength and computational complexity. In particular, we highlight the tree-attention mechanism, which is the first mechanism which can solve function composition and can be computed in quadratic time. These are both results which we prove.
>
> As we explain in section H, there could in principle be two “gaps” that are not addressed by our theorems and which we aim to address by simple experiments: (1) empirical learnability (i.e., whether one can learn the weights to solve function composition), and (2) practical efficiency (i.e., whether there are large constants hidden by our big-O notation which precludes it from being practical). Our experiment shows that neither of these is a problem: (1) one head and one layer of tree-attention learns function composition in less than half as many epochs as two layers of (vanilla) attention (see Figure 2), and (2) the actual clock running time is less than half that for the two-layer (vanilla) attention (see Figure 4).
>
> (We emphasize that these two points must be validated through an experiment rather than a theorem. In particular, regarding empirical learnability, there are very few learnability theorems of any kind in the theory of machine learning, and proving this appears to be far beyond the current techniques in our field.)
>
> We’ve also prepared further experiments.
>
> In order to show practical benefits of tree-attention, we have implemented tree-attention and self-attention on the COGS dataset [1]. COGS is a prominent NLP dataset which tests whether the model can perform compositional tasks (similar to function composition), and tree-attention performs better than self-attention on this (see Figure 7). For both tree-attention and self-attention, we use 3-layer transformers having encoding dimension = 64, heads = 4, and train it with a batch size = 32, learning rate = 0.001, dropout = 0.05 for 200 epochs. We repeat this experiment for 10 random seeds and report the mean and variance of exact match accuracy on the generalization set. The results are as follows:
>
> Tree-attention: Mean 26.5, Std dev: 12.8
>
> Self-attention: Mean: 23.9, Std dev: 8.7
>
> Furthermore, on almost half of the seeds, the tree-attention performs even better, achieving an accuracy of 30-50% (see Figure 8). We believe that the accuracy can be further boosted on such tasks with further empirical techniques, but believe that this would attract future works in this area. We have included this in the Section H.2 of the revised version of our paper.
>
> W3. Thank you for sharing the reference to hyper-feature attention, we will add a reference to it in our paper. That said, we want to emphasize that it appears to be studying a much weaker class of mechanisms based on linear attention, i.e., without softmax or any other nonlinear activation. Since there are many more papers on attention mechanisms in the literature than we had space to reference in our paper, we focused on the ones we felt were most relevant, in which the paper aims to design more expressive mechanisms (rather than less expressive ones without softmax).

---

> ### Author Response · Authors · 2025-11-21
> **Clarification of the question**
>
> Q: This question appears to stem from a misunderstanding.
>
> First and foremost, the argument you give here appears already in our paper, in Section E.1. More generally, if you have any two polynomials f and g, where the monomials of f “cover” the monomials of g (in the sense that each monomial of g contains a subset of the variables of a monomial of f), then poly-attention for f is able to simulate poly-attention for g. The argument you give outlines the special case of this where f is the single monomial of degree t (corresponding to order-t tensor attention), which naturally covers any other polynomial in t variables.
>
> The fact is, doing this simulation is almost always a bad idea, since it increases the computational complexity of the resulting attention mechanism. Consider, for example, the tree-attention we discuss in Theorem 3.1. This is a new mechanism we introduce, which is the first known mechanism which can solve function composition and be computed in O(n^2) time. If one applied your simulation to it, the result would be order-3 tensor attention, which requires Omega(n^3) time to compute! It’s true that order-3 tensor attention can simulate tree-attention and solve function composition, but this misses the point: it cannot be computed as quickly (regardless of any details about blowing up d by constant factors).
>
> Here’s another example to drive home the point: by the argument you give, vanilla self-attention can be simulated by order-1000 tensor attention. However, this doesn’t mean one shouldn’t use self-attention; it can be computed much faster (O(n^2) time) than order-1000 tensor attention (which requires n^1000 time!!).
>
> Our paper introduces a large class of poly-attention mechanisms, and there is one mechanism for each polynomial. This generalizes Strassen-attention and tensor-attention because each corresponds to poly-attention for a specific polynomial. Poly-attention is a strict superset because there are many other polynomials which do not correspond to any previously-studied mechanism.
>
> In general there is a natural tradeoff: picking a more complicated polynomial increases the expressive power, but also increases the computational complexity. The argument you give in your question does just this, and so it’s moving to a more computationally-complex place in the tradeoff. One wouldn’t want to do this unless it came with a corresponding increase in the expressive power.
>
> One new example of a poly-attention mechanism which we highlight is tree-attention, which has not previously been studied, which can be computed in quadratic time, and which can solve function composition. We are the first to introduce a mechanism with these properties, which we believe gets an exciting new place in the tradeoff.

---

> > ### Comment · Reviewer_8kRi · 2025-11-21
> > **About the question and some final comments**
> >
> > Thanks again for the response. After reading your reply, I want to just clarify that my original point did not stem from a misunderstanding. I wasn’t talking about efficiency but about what the phrase “strict superset” is intended to mean. Since any poly-attention head can be simulated by tensor attention with a wider feature dimension, the two are expressively equivalent unless you fix a particular computational budget. If the claim is meant specifically within the class of mechanisms that remain quadratic-time, that is reasonable, but the paper does not phrase it that way.
> >
> > With that in mind, I would suggest adjusting the wording slightly so that it is clear that the “strict superset” claim refers to efficiency within a fixed computational regime rather than expressive power in the unrestricted sense.
> >
> > On the practical side, I still have doubts about how much this adds beyond existing higher-order variants like tensor attention or Strassen attention (or very similar other attention variants in the literature). In real settings we do not know the correct interaction structure in advance, and most practical systems learn actual interactions **approximately** instead of **exactly**. Thus remembering the approximate or linear-attention mechanisms rather than exact higher-order ones, the worst-case complexities emphasized in the paper are not usually the bottleneck.
> >
> > The absence of experimental comparisons, especially against multi-head and 2 or 3 layer versions of these existing attention mechanisms, also strengthens my impression that the main contribution here is a theoretical organization, instead of suggesting a practical improvements. It seems to offer a clean and unified framework that generalizes previous mechanisms, but not necessarily something that would significantly improve practical models.

---

> > > ### Author Response · Authors · 2025-12-03
> > >
> > > **More exposition on function composition**: Thank you for the suggestion. We were a bit limited for space in the original version, but we will expand on this motivation in the camera-ready version like you suggest.
> > >
> > > **Experiments**: Thank you, we’re glad that the new experiments addressed your concerns.
> > >
> > > **Hyperattention**: Thank you for mentioning this reference, we will include it in the camera ready version as well. The architectural definitions given in that paper are quite similar to tensor-attention, and since tensor-attention is a subset of poly-attention, poly-attention should be able to capture hyper-attention with minor modifications as well.
> > >
> > > **“Strict superset”**: We agree that it would be important to qualify the phrase “strict superset” if we had used it in our paper. However, that phrase doesn’t appear in our paper (even just the word “superset” doesn’t appear), nor do we believe we made an ambiguous claim like this using different wording, so we are unsure what you’re quoting or referring to.
> > >
> > > We aimed to be very careful about describing how poly-attention improves on prior work, and we don’t believe we made any false or misleading claims. For example, in the abstract, we wrote:
> > >
> > > “Our mechanisms can incorporate arbitrary higher-order (tensor) computations as well as arbitrary relationship structures between the input tokens, and they include the aforementioned alternatives as special cases.”
> > >
> > > This is entirely accurate and aligns with what you are saying (since tensor-attention and Strassen-attention correspond to poly-attention for particular polynomials).
> > >
> > > We’ve slightly adjusted the wording in a couple of places to emphasize this (which we’ve marked in red in the updated version of the paper). Unfortunately there is no longer a possibility of conversation with the reviewers, but we hope this addresses your concern here.
> > >
> > >
> > > **“On the practical side, I still have doubts”**: For practical advantages, we have now added Section H.2 where we show that 3-layer tree-attention performs significantly better than 3-layer self-attention on the COGS dataset (a well-known NLP dataset). We have included comparison only with self-attention as the tensor-attention and strassen-attention, with their cubic running times, are too slow on hardware (see Figure 5 where we quantify this in implementations) even though they are more expressive than self-attention and tree-attention.
> > >
> > >
> > > **“the worst-case complexities emphasized in the paper are not usually the bottleneck.”**-
> > > We’re unfortunately not completely sure what you meant by this, and since a back-and-forth conversation is no longer possible, here are two different interpretations we had and our responses:
> > > **If you think the lower bounds are worst case and more efficient algorithms can be constructed**: The inference running times (from brute force computations) and matching lower bounds are given in this paper, which will always be the same unless some heuristics are applied. Studying speed-up heuristics should be future work given that this paper is only for proposing a framework to build upon. But as you emphasize, approximations are often sufficient, and in our paper we give much faster, almost-linear-time approximation algorithms for all poly-attention mechanisms.
> > > **If you meant self-attention works well already in practice and we do not need higher-order ones**: We feel like this point is orthogonal to the purpose of this paper. Self-attention, despite working well in practice, has a lot of weaknesses that have been immensely studied. One way to mitigate this is to use stronger attention structures which gives rise to the importance of higher-order attention. Furthermore, we have experimentally shown that tree-attention learns function composition with much less data and much less FLOPs (Figure 4), which is an argument towards why tree-attention might be better.
> > >
> > > **“​​also strengthens my impression that the main contribution here is a theoretical organization”**- Indeed the purpose of this paper is to provide a framework for developing higher-order attention schemes. We feel it is unfair to say that this might not improve practical models as we have indeed observed better accuracy on real NLP tasks like the COGS dataset.
> > >
> > > These are baseline experiments and they look promising for a simple tree-attention. However, given the length of the paper and the theoretical results presented, we feel that more poly-attention and tree-attention schemes need to be studied in future work to see if they improve on other datasets. The goal is to give theoretical justification to really understand the tension between expressivity and speed. We do not claim to have a correct answer, but are looking for a sweet spot between expressivity and efficiency.

---

> ### Comment · Reviewer_8kRi · 2025-11-21
>
> Thank you for the replies.
> W1) I am personally aware of the place of "function composition" in the literature. I was just suggesting sparing more space on it in the paper since it is the main motivation. However, either way is totally fine.
>
> W2) Yes, I had not asked for convergence proof. However, I checked new version of figure 2, now it seems more scientific than the old version I remember. Thank you for updating it. I also believe that the further experiments improved the paper.
>
> W3) The mentioned paper is not just particularly about linear attention variants. Please be careful, it is more relevant than you guess. It has exactly the same architectural definitions.

---

### Official Review · Reviewer_Pxmp · 2025-11-02

**Soundness:** 3
**Presentation:** 3
**Contribution:** 3
**Rating:** 6
**Confidence:** 4

**Summary:**

This paper introduces poly-attention, a generalized class of self-attention mechanisms that incorporate higher-order (tensor) computations and arbitrary relational structures between input tokens. The authors show that existing higher-order attention mechanisms, such as tensor attention and Strassen attention, are special cases of poly-attention. They provide a systematic analysis of the computational complexity and expressive power of these mechanisms, including new algorithms and matching lower bounds. A key contribution is the introduction of tree-attention, a subclass of poly-attention that can be computed in quadratic time and can perform multi-fold function composition—a task beyond the reach of standard self-attention and prior higher-order variants under similar time constraints.

**Strengths:**

Originality: The paper introduces a unified framework (poly-attention) that generalizes several recent higher-order attention mechanisms. The introduction of tree-attention and its theoretical and empirical validation is novel and meaningful.

Quality: The technical contributions are substantial. The authors provide rigorous complexity analyses, including both upper and lower bounds, and support their claims with proofs and experiments. The connection to fine-grained complexity (e.g., SETH, Max-2SAT) is well-executed.

Clarity: The paper is generally well-written, with clear definitions and a structured presentation. The use of graphical representations for tree-attention is intuitive.

Significance: The work addresses a key limitation of transformers—their inability to capture higher-order dependencies efficiently—and offers a practical path toward more expressive and scalable attention mechanisms. The results could influence both theoretical and applied research in efficient transformer architectures.

**Weaknesses:**

Experimental Scope: While the paper includes an experimental validation of tree-attention, the evaluation is limited to a single task (function composition). More diverse benchmarks (e.g., on standard NLP or reasoning tasks) would strengthen the practical relevance of the proposed mechanisms.

Presentation of Lower Bounds: The lower-bound proofs, especially those based on fine-grained complexity, are highly technical and may be difficult to follow for a general audience. A more intuitive explanation or summary of the proof techniques would improve accessibility.

**Questions:**

1. Could the authors evaluate tree-attention on more complex or realistic tasks, such as logical reasoning, mathematical problem-solving, or long-range dependency tasks, to better demonstrate its practical advantage?

2. How does tree-attention scale with sequence length in practice, especially when n is large (e.g., >10K tokens)? Are there any memory or numerical stability issues?

3. Could the authors provide more intuition or a high-level overview of the lower-bound proofs, especially for readers less familiar with fine-grained complexity?

---

> ### Author Response · Authors · 2025-11-21
> **Experiments on NLP datasets and other clarifications**
>
> We thank the reviewer for the thoughtful review. We answer the questions below:
>
> W1. We’ve performed additional experiments, including on a popular NLP dataset; see the response to Q1 below.
>
> That said, we also want to clarify the goal of our experiments. The goal of our paper is to introduce the new family of poly-attention mechanisms, and prove theorems about their representational strength and computational complexity. In particular, we highlight the tree-attention mechanism, which is the first mechanism which can solve function composition and can be computed in quadratic time. These are both results which we prove.
>
> As we explain in section H, there could in principle be two “gaps” that are not addressed by our theorems and which we aim to address by simple experiments: (1) empirical learnability (i.e., whether one can learn the weights to solve function composition), and (2) practical efficiency (i.e., whether there are large constants hidden by our big-O notation which precludes it from being practical). Our experiment shows that neither of these is a problem: (1) one head and one layer of tree-attention learns function composition in less than half as many epochs as two layers of (vanilla) attention (see Figure 2), and (2) the actual clock running time is approximately same as that for the one-layer (vanilla) attention (see Figure 5).
>
> (We emphasize that these two points must be validated through an experiment rather than a theorem. In particular, regarding empirical learnability, there are very few learnability theorems of any kind in the theory of machine learning, and proving this appears to be far beyond the current techniques in our field.)
>
>
> W2. Our proofs are indeed quite involved. We aimed to give a very high-level overview in section 4 (“technique overview”), but we were quite constrained for space there. We then give more detailed proof overviews before each proof. For example, we begin section C.2 with an outline of the steps of the proof, then we break the proof of each lemma into sections with bold paragraph titles.
>
>
> Q1. In order to show practical benefits of tree-attention, we have implemented tree-attention and self-attention on the COGS dataset [1]. COGS is a prominent NLP dataset which tests whether the model can perform compositional tasks (similar to function composition), and tree-attention performs better than self-attention on this (see Figure 7).
> For both tree-attention and self-attention, we use 3-layer transformers having encoding dimension = 64, heads = 4, and train it with a batch size = 32, learning rate = 0.001, dropout = 0.05 for 200 epochs. We repeat this experiment for 10 random seeds and report the mean and variance of exact match accuracy on the generalization set. The results are as follows:
>
>
> Tree-attention: Mean  26.5, Std dev:   12.8
>
> Self-attention: Mean:  23.9, Std dev:  8.7
>
> Furthermore, on almost half of the seeds, the tree-attention performs even better, achieving an accuracy of 30-50% (see Figure 8). We believe that the accuracy can be further boosted on such tasks with further empirical techniques, but believe that this would attract future works in this area. We have included this in the Section H.2 of the revised version of our paper.
>
> Q2. The functioning of tree-attention is quite similar to that of self-attention, and we believe that it should behave in a similar way.
>
> Q3. The fine-grained complexity results used have been stated in Section A.2
>
>
> Please let us know if you have any further comments or suggestions.
>
>
>
>
> [1] Kim, N., & Linzen, T. (2020). COGS: A compositional generalization challenge based on semantic interpretation. EMNLP 2020

---

### Author Response · Authors · 2025-12-03

**Note to AC**: Since we weren’t able to conclude the discussion with the reviewers, we give a summary here of the reviews and our responses.

**SUMMARY OF REVIEWERS COMMENTS**

All of the reviewers appreciated the novelty of tree-attention (our key contribution), and the strength of our technical results and theorems. (Reviewer Henj: The finding of tree attention as the only (strong attn model) computable in quadratic time is an excellent and complete theoretical result; Reviewer Pxmp: the technical contributions are substantial; Review 8kRi: appreciated the unified framework of poly attention and commented its usefulness for future theoretical work; Review Henj: significant and elegant theoretical contribution to the understanding of attn mechanism. Main strength is its rigorous analysis.)
All reviewers said that the paper was well written with clear definitions and structured presentation and all reviewers appreciated the technical merits of the paper.

The main weakness raised by all reviewers concerns the strength of our empirical studies. The reviewers pointed out that our empirical results are limited to a single task (function composition) and that the performance we were getting was not convincing (e.g., stability of the results under initial conditions;  the experiments were at too small of a scale to demonstrate real-world applicability.) Two reviewers suggested further experiments on other benchmark NLP datasets, and several reviewers  suggested additional evaluation statistics (e.g., memory, throughput, stability).

---

> ### Author Response · Authors · 2025-12-03
>
> **SUMMARY OF OUR RESPONSE**
>
> We addressed the main criticism (weakness of our experimental results) with substantially improved empirical results.
>
> ​​First, we have obtained significantly improved empirical results on the task of function composition. (See Section H.1 in the revised version of our paper.) In our new experiments, we trained transformers of the following types: (i) one layer self attention, (ii) two layer self-attention, and (iii) one layer tree-attention. To summarize, one layer tree-attention can successfully learn function composition, despite having only three query-key matrices and only one MLP layer (compared to two-layer self-attention that has two query matrices, two key matrices and two MLP layers).
>
> Furthermore, our experiments demonstrate that tree-attention is faster to learn and has less fluctuations (See Figure 3), achieves better accuracy (See Figure 4), and uses less space compared to one and two-layer self attention. We also plot (Figure 5) the inference running time of the three attention mechanisms, and show that tree-attention has runtimes comparable to self attention.
>
>
>
>
>
> Secondly, we have implemented tree-attention and self-attention on the COGS dataset [1], a prominent NLP dataset which tests whether the model can perform compositional tasks. We show that 3-layer tree-attention performs significantly better than 3-layer self attention on this dataset. (See Section H.2 in the revised version of our paper.) We have included comparison only with self-attention since other related attention mechanisms (tensor attention and Strassen attention) have cubic runtimes and were too slow to implement. (See Figure 5 where we give quantitative results on these implementations.)
> For both tree-attention and self-attention, we use 3-layer transformers having encoding dimension = 64, heads = 4, and train it with a batch size = 32, learning rate = 0.001, dropout = 0.05 for 200 epochs. We reported the mean and variance of exact match accuracy (over random choice of seeds) on the generalization set. The results are summarized as follows:
>
> Tree-attention: Mean 26.5, Std dev: 12.8 Self-attention: Mean: 23.9, Std dev: 8.7
> Only reviewer 8kRi was able to reply to our response, and they wrote that they “believe that the further experiments improved the paper.”

---

> > ### Author Response · Authors · 2025-12-03
> >
> > We also addressed specific additional concerns raised that we feel were largely misunderstandings:
> >
> > - Reviewer Henj had concerns about the wording of our representational strength results. We have a number of theorems stating that there exist weights for certain poly-attention mechanisms which allow them to solve certain tasks (like function composition). The reviewer believed we were making the stronger claim that these weights could be efficiently learned. Although we tried to be very clear about this already in the paper, we have further clarified the wording around these results to eliminate any possible confusion. (More generally, there are very few learnability theorems of any kind in the theory of machine learning, and proving this appears to be far beyond the current techniques in our field. In order to address this gap, we included the experiments that show that tree attention is able to learn the weights for function composition, even in many fewer epochs than a comparable (vanilla) attention network with more layers.)
> >
> > - Reviewer Henj also mentioned some other weaknesses that we did not fully understand. For example, their weakness 3 says that our running time table did not compare with superquadratic mechanisms, but it referred to the wrong figure (Fig 4) in our paper. Our actual running time table (Fig 1) already did this comparison they asked about. We asked some clarifying questions, which unfortunately they did not reply to. We nonetheless gave some explanations in response to their review which we hope addresses their confusion.
> >
> > - Reviewer 8kRi had a misunderstanding about the sense in which poly-attention generalizes prior t-order tensor attention. They gave an argument (in their question section) showing that any poly-attention can be simulated by t-order attention for a sufficiently large t, and asked whether we had considered this. In fact we had, and the same argument given by the reviewer appeared already in our paper in section E.1. As we discussed in our response, this type of simulation is ill-advised, since it increases the runtime complexity of the attention mechanism without giving any benefits. In general, when picking a polynomial to use in poly-attention, there is a tradeoff between the runtime complexity of the attention mechanism and its representational strength. It doesn’t make sense to do a simulation with a more computationally expensive mechanism unless one gets more expressive power in exchange (which this argument does not). And indeed, a main draw of our tree-attention is that it is the first mechanism with only a quadratic runtime complexity which is strong enough to represent function composition. In order to help with this type of misunderstanding, we clarified the wording in a couple places in our paper where we compare different mechanisms, to make it clear that both computational complexity and representational strength must be simultaneously considered.

---

> > > ### Author Response · Authors · 2025-12-03
> > >
> > > **OVERALL SUMMARY**
> > >
> > > This is primarily a theoretical paper, where our goal was to unify existing higher-order attention mechanisms through a new general framework that we call poly-attention. A main contribution of our work is the introduction of tree-attention, which is a subclass of poly-attention. We prove  that tree-attention is the first attention mechanism that can be computed in quadratic time, and that can solve function composition. Additionally, we give a comprehensive theoretical analysis of the runtime complexity and representational strength of poly-attention and its subclasses.
> > >
> > > All the reviewers appreciated the novelty of our framework, the clear exposition, and the technical depth of our theoretical results. They have mentioned this as an “elegant generalization” of higher-order attention schemes, and liked the rigour and clarity in the proofs of both computational complexities and representational strengths.
> > >
> > > We emphasize that no weakness regarding the theoretical contributions were mentioned, and ICLR is a conference which welcomes both theoretical and experimental works. Considering that, we have significantly improved the experimental results to address the requests from the reviews, both on synthetic as well as on more realistic NLP datasets, we believe that we have demonstrated the power and potential of tree-attention in terms of expressivity and computational speed.
> > >
> > > We emphasize that this is primarily a theoretical paper, and all reviewers were very positive with respect to our theoretical contributions. As this is the first paper studying this new class of highly expressive attention mechanisms, it should not be expected, nor do we claim, that we have demonstrated that our new mechanisms are better than existing ones from a practical perspective.  We are claiming to have introduced a novel family of attention mechanisms, together with a thorough investigation on the power of these new mechanisms relative to previously studied ones. The goal is to give theoretical justification to really understand the tension between expressivity and speed, since prior higher-order mechanisms all involved worse running times.

---

### Meta-Review · Area_Chair_5Jk8 · 2026-01-12

**Summary:**

Reviewers appreciate the theoretical framework introduced in the paper to study the existing higher order attention variants, "a clean and unified framework that generalizes previous mechanisms." They find the theoretical analysis int he paper useful and written well. The main concerns were around practical application of the results and clarification about proof structure. Authors addressed the questions in their response. Given this is primarily a theoretical paper, limited experiments included are sufficient and I suggest acceptance.

**Reviewer Concerns:**

The main concerns were around practical application of the results and clarification about proof structure.

**Reviewer Scores:**

8kRi 4 -> 6
Henj 4-> 6

---

### Decision · Program_Chairs · 2026-01-26

Accept (Poster)